



# Seasonal variation and origins of volatile organic compounds observed during two years at a western Mediterranean remote background site (Ersa, Cape Corsica)

Cécile Debevec[1], Stéphane Sauvage[1], Valérie Gros[2], Thérèse Salameh[1], Jean Sciare[2,3], François Dulac[2], Nadine Locoge[1].

[1]IMT Lille Douai, Univ. Lille, SAGE - Département Sciences de l'Atmosphère et Génie de l'Environnement, 59000 Lille, France

[2] Laboratoire des Sciences du Climat et de l'Environnement (LSCE), Unité Mixte CEA-CNRS-UVSQ, IPSL, Univ. Paris-Saclay, Gif-sur-Yvette, 91190, France

[3] Climate and Atmosphere Research Centre, the Cyprus Institute (CyI), Nicosia, 2121, Cyprus

*Correspondence to*: Stéphane Sauvage (stephane.sauvage@imt-lille-douai.fr) – Cecile Debevec (cecile.debevec@imt-lille-douai.fr)

**Abstract.** An original time series of about 300 atmospheric measurements of a wide range of volatile organic compounds (VOCs) has been obtained at a remote Mediterranean station on the northern tip of Corsica Island (Ersa, France) over 25 months from June 2012 to June 2014. This study presents the seasonal variabilities of 25 selected VOCs, and their various associated sources. The VOC speciation was largely dominated by oxygenated VOCs (OVOCs) along with primary anthropogenic VOCs having a long lifetime in the atmosphere. VOC temporal variations are then examined. Primarily of local origin, biogenic VOCs exhibited notable seasonal and interannual variations, related to temperature and solar radiation ones. Anthropogenic compounds have shown an increasing concentration trend in winter (JFM months) followed by a decrease in spring/summer (AMJ/JAS months), and different concentration levels in winter periods of 2013 and 2014. OVOC concentrations were generally higher in summertime, mainly due to secondary and biogenic sources, whereas their concentrations during fall and winter were potentially more influenced by anthropogenic primary/secondary sources. Moreover, an apportionment factorial analysis was applied to a database comprising a selection of 14 primary individual or grouped VOCs by means of the positive matrix factorization (PMF) technique. A PMF solution composed of 5 factors was taken on. It includes a biogenic factor (which contributed 4% to the total VOC mass), three anthropogenic factors (namely short-lived anthropogenic sources, evaporative sources, and long-lived combustion sources; which together accounted for 57%), originating from either nearby or more distant emission areas (such as Italy and south of France); and a remaining one (39%) connected to the regional background pollution. Variations in these main sources impacting VOC concentrations observed at the receptor site are also investigated at seasonal and interannual scales. In spring and summer, VOC concentrations observed at Ersa were the lowest in the 2-yr period, despite higher biogenic source contributions and since anthropogenic sources advected to Ersa were largely influenced by chemical transformations and vertical dispersion phenomena and were mainly of regional origins. During fall and winter, anthropogenic sources showed higher accumulated contributions when





European air masses were advected to Ersa and could be associated to potential emission areas located in Italy and possibly more distant ones in central Europe. Higher VOC concentrations during winter 2013 compared to winter 2014 ones could be related to anthropogenic source contribution variations probably governed by emission strength of the main anthropogenic sources identified in this study together with external parameters, i.e. weaker dispersion phenomena and pollutant depletion.

High frequency observations collected during several intensive field campaigns conducted at Ersa during the three summers 2012-2014 confirmed findings from bi-weekly samples in terms of summer concentration levels and source apportionment. However, they suggest that higher sampling frequency and temporal resolution, in particular to observe VOC concentrations variation during the daily cycle, are needed to confirm the deconvolution of the different anthropogenic sources identified by PMF approach. Finally, comparisons of the 25 months of observations at Ersa with VOC measurements conducted at 17 other

European monitoring stations highlight the representativeness of the Ersa background station for monitoring seasonal variations in VOC regional pollution impacting continental Europe. Nevertheless, winter VOC concentration levels can significantly vary between sites, pointing out spatial variations in anthropogenic source contributions. As a result, Ersa concentration variations in winter were more representative of VOC regional pollution impacting central Europe. Interannual and spatial VOC concentration variations in winter were also significantly impacted by synoptic phenomena influencing

meteorological conditions observed in continental Europe, suggesting that short observation periods may reflect the variability of the identified parameters under the specific meteorological conditions of the studied period.

## 1 Introduction

The main trace pollutants in the atmosphere encompass a multitude of volatile organic compounds (VOCs), with lifetimes varying from minutes to months (e.g., Atkinson, 2000). Their distribution is principally owing to (i) multiple natural and

anthropogenic sources, which release VOCs directly to the atmosphere. At a global scale, natural emissions are quantitatively larger than anthropogenic ones (Guenther et al., 2000) and the largest natural source is considered to be the vegetation (Finlayson-Pitts & Pitts, 2000; Guenther et al., 2000, 2006). In urban areas, numerous anthropogenic sources can abundantly emit various VOCs (Friedrich and Obermeier, 1999). Once in the atmosphere, VOC temporal and spatial variabilities are notably influenced by (ii) mixing processes along with (iii) removal processes or chemical transformations (Atkinson and Arey,

2003; Atkinson, 2000). Accordingly, with a view to extensively characterize VOC sources, it is meaningful to examine their chemical composition, in addition to identify the factors controlling their variations at different time scales.

VOC regional distributions are eminently changing as a result of various confounding factors, namely emission strength of numerous potential sources, diverse atmospheric lifetimes and removal mechanisms, transport process and fluctuating meteorological conditions. Therefore, these elements underline the necessity to carry out long-term VOC

measurements. In Europe, studies essentially focus on urban and suburban locations (e.g., Derwent et al., 2014 and von Schneidemesser et al., 2010 in United Kingdom; Salameh et al., 2019 and Waked et al., 2016 in France; Roemer et al., 1999 in the Netherlands; Fanizza et al., 2014 in Italy), reflecting concerns about the role of VOCs in urban air quality control,



efficiency assessment of national VOC emission regulation implementations, and population exposure. European VOC observations in the background atmosphere are still dedicated largely to process studies and short-term research missions. However, there are growing efforts now to carry out European background measurements over several seasons (e.g., Seco et al., 2011), one year (such as Helmig et al., 2008; Legreid et al., 2008) and even several years (Solberg et al., 1996, 2001 and

Tørseth et al., 2012 at several European sites; Hakola et al., 2006 and Hellén et al., 2015 in Scandinavia; Dollard et al., 2007; Grant et al., 2011 and Malley et al., 2015 in United Kingdom; Borbon et al., 2004; Sauvage et al., 2009 and Waked et al., 2016 in France; Plass-Dülmer et al., 2002 in Germany; Navazo et al., 2008 in Iberian Peninsula; Lo Vullo et al., 2016 in Italy). These multi-year studies were conducted ensuing the increasing demand for high quality VOC data, and long-term monitoring have led to international programs like the European Research Infrastructure for the observation of Aerosol, Clouds and Trace

gases (ACTRIS - https://www.actris.eu/; last access: 03/04/2020), the European Monitoring and Evaluation Program (EMEP - http://www.emep.int/; last access: 03/04/2020 - Tørseth et al., 2012), and the Global Atmosphere Watch of the World Meteorological Organization (WMO-GAW - http://www.wmo.int/pages/prog/arep/gaw/gaw_home_en.html; last access: 03/04/2020). Regarding VOCs, these research studies principally explored emission regulation efficiency and links between tropospheric ozone production and changes in VOC concentrations, and assessed seasonal variations and regional distributions

in VOC concentrations. Nonetheless, investigations on principal factors governing temporal and spatial variations in VOC concentration levels in the European background atmosphere remain scarce. However, the consideration of the influence of (i) source emission strength variations (built upon a factorial analysis – e.g., Lanz et al., 2009), (ii) long-range transport of pollution (e.g., by the examination of air mass trajectories combined with measured concentrations at a study site; Sauvage et al., 2009) and (iii) fluctuations in meteorological conditions (which are prone to disperse the pollutants on a regional or long-

range scale through convective and advective transport) can supply relevant information to deal more in depth with the evaluation of seasonal variations and regional distribution of VOC concentrations in the European background atmosphere.

Particulate and gaseous pollutants detrimentally affect the Mediterranean atmosphere. Accordingly, they are prone to increase aerosol and/or ozone concentrations in the Mediterranean, regularly higher compared to most regions of continental Europe, and primarily during summer (Doche et al., 2014; Nabat et al., 2013; Safieddine et al., 2014). The Mediterranean

region is known to be a noteworthy climate change "hot spot", which is expected to go through severe warming and drying in the 21st century (Giorgi, 2006; Kopf, 2010; Lelieveld et al., 2014). As a consequence, this can have serious consequences on the release of VOCs from biogenic and anthropogenic sources along with their fate in the atmosphere, with uncertain predicted impacts (Colette et al., 2012, 2013; Jaidan et al., 2018). Actually, the examination of air composition, concentration levels and trends in the Mediterranean region persists to be challenging, primarily on account of the lack of extensive in-situ observations.

In order to improve our actual comprehension of the complexity of the Mediterranean atmosphere, it is essential to increase the atmospheric pollutant observations, including speciated and reactive VOCs, at representative regional background sites. Given this context, as part of the multidisciplinary regional research program MISTRALS (Mediterranean Integrated Studies at Regional and Local Scales; http://mistrals-home.org/, last access: 03/04/2020), the project ChArMEx (the Chemistry-Aerosol Mediterranean Experiment, http://charmex.lsce.ipsl.fr, last access: 03/04/2020; Dulac, 2014) focused on the





development and coordination of regional research actions. More precisely, ChArMEx aims at assessing the current and future state of the atmospheric environment in the Mediterranean along with examining its repercussions on the regional climate, air quality and marine biogeochemistry. In the frame of ChArMEx, Michoud et al. (2017) characterized the variations in VOC concentrations observed at Ersa in summer 2013 (from 15 July to 5 August 2013) by identifying and examining their sources.

5       The present study was designed to characterize the seasonal variations in the sources of VOCs affecting the western Mediterranean atmosphere. An extensive chemical composition dataset was collected at a receptor site considered to be representative of the northwestern basin. In this article, we present and discuss ambient levels and variations of a selection of VOCs observed at the Ersa station of the Corsican Observatory for Research and Studies on Climate and Atmosphere-ocean environment (CORSiCA - https://corsica.obs-mip.fr/, last access: 03/04/2020; Lambert et al., 2011), over more than two years

as part of the ChArMEx project. Selected species include alkanes, alkenes, alkyne, aromatic compounds and oxygenated VOCs (OVOCs), which were measured using off-line techniques. To reach its objective, this study will (i) quantify concentration levels of the targeted VOCs, (ii) specify their temporal variations at seasonal and interannual scales, (iii) identify and characterize their main sources by statistical modelling, (iv) assess and examine their source contributions on seasonal bases, together with (v) examine the representativeness of the Ersa station in terms of seasonal variations in VOC concentrations

impacting continental Europe.

## 2 Material and Methods

### 2.1 Study site

Located in the northwestern part of the Mediterranean Sea, Corsica Island is a French territory situated 11 km northerly from Sardinian coasts, 90 km easterly from Tuscany (Italy) and 170 km southerly from the French Riviera (France). Being the 4th

largest Mediterranean island, its land corresponds to an area of 8681 km$^2$ encompassed by around 1000 km of coastline (Encyclopædia Britannica, 2018). Corsica contrasts to other Mediterranean islands due to the importance of its forest cover (about a fifth of the island).

       Within the framework of the ChArMEx project, an enhanced observation period has been set-up at a ground-based station in the north of Corsica (Ersa; 42.969°N, 9.380°E) over 25 months, from early June 2012 to late June 2014, with the

aim of providing a high quality controlled climatically relevant gas/aerosol database following the recommendations and criteria of the international atmospheric chemistry networks ACTRIS, EMEP, and GAW. This remote site is located on the highest point of a ridge equipped with windmills (see the orographic description of the surroundings in Cholakian et al., 2018), at an altitude of 533 m above sea level (a.s.l.). Given its position on the north of the 40-km long Cape Corsica peninsula (Fig. 1), the Mediterranean Sea is clearly visible from the sampling site on west, north, and east sides (2.5-6 km from the sea; see

also the figure presented in Michoud et al., 2017). The station was initially set up in order to monitor and examine pollutions advected to the receptor site by air masses advected over the Mediterranean and originating from the Marseille-Fos-Berre





region (France; Cachier et al., 2005), the Rhone Valley (France), and the Po Valley (Italy; Royer et al., 2010), namely largely industrialized regions. The study site is about 30 km north of Bastia (Fig. 1), the second largest Corsican city (44121 inhabitants; census 2012) and the main harbour. An international airport (Bastia-Poretta) is located 16 km further south of Bastia city centre. Note that more than two millions of passengers transited in Corsica per Bastia during the tourist season

(May-September) in 2013, (ORT Corse, 2013; http://www.corse.developpement-durable.gouv.fr/IMG/pdf/Ete2013.pdf, last access: 30/05/2020). However, as the Cape Corsican peninsula benefits in the south from a mountain range (peaking between 1000 and 1500 m a.s.l.) acting as a natural barrier, the sampling site is therefore not affected by transported pollutions originating from Bastia. Only small rural villages and a small local fishing harbour (Centuri) are found in the surroundings within 5 km of the measurement site. Additionally, the station is accessible by a dead end road serving only the windmill site,

surrounded by vegetation made up of Mediterranean maquis, a shrubland biome characteristically consisting of densely growing evergreen shrubs, and also roamed by a herd of goats from a nearby farm. Some forests are also located nearby (78 % of holm oaks, with some cork oaks and chestnuts), thus ensuring that local anthropogenic pollution does not contaminate in-situ observations. As a result, the Ersa station can be characterized as a remote background Mediterranean site.

## 2.2 Experimental Set-up

### 2.2.1 VOC measurements

During a period of two years, non-methane hydrocarbons (NMHCs) and OVOCs were measured routinely employing complementary off-line methods. Four-hours-integrated (09:00-13:00 or 12:00-16:00 UTC) ambient air samples were collected bi-weekly (every Monday and Thursday) into steel canisters and on sorbent cartridges. The inlets were roughly 1.5 m above the roof of a container housing the analysers. Table 1 describes our measurements set up throughout the observation

period.

As generally realized in the EMEP network, 21 $C_2$-$C_9$ NMHCs were collected into Silcosteel canisters of a volume of 6 L, conforming to the TO-14 technique, which is considered adequate for many non-polar VOCs (US-EPA, 1997). 152 air samples were realized with a homemade device (PRECOV) for sampling air at a steady flow rate regulated to 24 mL min$^{-1}$ by canisters previously placed under vacuum. NMHC analysis was performed by a gas chromatograph coupled with a flame

ionization detector (GC-FID) within three weeks following sampling. Separation was performed by a system of dual capillary columns supplied with a switching device: the first one was a CP Sil5CB (50 m x 0.25 mm x 1 µm), suitable for the elution of VOCs from six to nine carbon atoms and the other one was a Plot $Al_2O_3/Na_2SO_4$ (50 m x 0.32 mm x 5 µm), in order to effectively elute VOCs from two to five carbon atoms. Four main steps constituted the quality assurance/quality control program: (i) the implementation of standard operating procedures, (ii) canister cleaning and certification (blank levels <0.02

ppb), (iii) regular intercomparison exercises and (iv) sampling tests carried out in field conditions and concomitant to in-situ measurements (Sauvage et al., 2009).



About 150 off-line 4-h-integrated air samples were gathered using sorbent cartridges (63 air samples on multi-sorbent cartridges and 91 ones on 2,4-dinitrophenylhydrazine - DNPH - cartridges), by means of an automatic clean room sampling system (ACROSS, TERA Environment, Crolles, France). $C_1$-$C_{16}$ VOCs were collected via a 0.635 cm diameter 3-m long PFA line. They are then trapped into one of the two cartridge types: a multi-sorbent one consisted of carbopack C (200 mg) and

carbopack B (200 mg; marketed under the name of carbotrap 202 by Perkin-Elmer, Wellesley, Massachusetts, USA), and a Sep-Pak DNPH-Silica one (proposed by Waters Corporation, Milford, Massachusetts, USA). These off-line techniques are further characterized in Detournay et al. (2011) and their satisfying use in-situ has already been discussed by Detournay et al. (2013) and Ait-Helal et al. (2014). Succinctly here, the sampling of 44 $C_5$-$C_{16}$ NMHCs, comprising alkanes, alkenes, aromatic compounds and four monoterpenes, as well as six $C_6$-$C_{11}$ n-aldehydes, was conducted at a flow rate fixed at 200 mL min$^{-1}$ and

using the multi-sorbent cartridges. These latter were preliminary prepared by means of a RTA oven (French acronym for *"régénérateur d'adsorbant thermique"* – manufactured by TERA Environment, Crolles, France) in order to condition them during 24 h with purified air at 250 °C and flow rate regulated at 10 mL min$^{-1}$. In parallel, 15 additional $C_1$-$C_8$ OVOCs were collected using the DNPH cartridges at a flow rate fixed at 1.5 L min$^{-1}$. During the field campaign, several ozone scrubbers have been successively inserted in the sampling lines in order to limit any eventual ozonolysis of the measured VOCs: a $MnO_2$

ozone scrubber was retained for the multi-sorbent cartridges while KI ozone scrubber was placed upstream of the DNPH cartridges. Moreover, stainless-steel particle filters of 2 µm diameter porosity (Swagelok) were installed in order to prevent particle sampling. Then, samples were transferred to the laboratory to be analysed within 6 weeks using a GC-FID (for multi-adsorbent cartridges) or by high-performance liquid chromatograph connected to an ultraviolet detector (HPLC-UV; for DNPH cartridges).

The reproducibility of each analytical instrument has been frequently checked, firstly by analysing a standard, and examining results by plotting them on a control chart realized for each compound. The VOC detection limit was determined as 3 times the standard deviation of the blank variation. Obtained detection limits in this study were all below 0.05 µg m$^{-3}$ for the steel canisters and the DNPH cartridges, and of 0.01 µg m$^{-3}$ for the multi-sorbent cartridges. The uncertainties for each species were evaluated respecting the ACTRIS-2 guidelines for uncertainty evaluation (Reimann et al., 2018) considering

precision, detection limit and systematic errors in the measurements. Evaluated relative uncertainties ranged from 7% to 43% concerning steel canisters, between 7% and 65% for multi-sorbent cartridges and from 6% to 41% concerning DNPH cartridges. Finally, the VOC dataset was validated following the ACTRIS protocol (Reimann et al., 2018).

### 2.2.2 Ancillary measurements

Other trace gases (CO and $O_3$) and meteorological parameters were ancillary monitored at the Ersa site during the observation

period. CO was measured from 22 November 2012 to 16 December 2013 by a commercial analyser (G2401; Picarro, Santa Clara, California, USA) using a cavity ring-down spectroscopy (CRDS) at a time resolution of 5 min. $O_3$ was measured from 31 May 2012 to 26 December 2013 by means of a UV absorption analyser (TEI 49i manufactured by Thermo Environmental Instruments Inc., Waltham, Massachusetts, USA) at a time resolution of 5 min. Meteorological parameters (temperature,





pressure, relative humidity, wind speed, wind direction and total – direct and diffuse - solar radiation) were measured every minute from 8 June 2012 to 14 August 2012, and every 5 min from 15 August 2012 to 11 July 2014, with a weather station (CR1000 manufactured by Campbell Scientific Europe, Antony, France) placed at approximately 1.5 m above an adjacent container roof. Note that trace gases and meteorological results presented in this study are 4-hour averages concurrent to periods when the VOC sampling periods were realized.

### 2.2.3 Additional high frequency VOC measurements performed at Ersa

Additional VOC measurements were realized during summer campaigns performed in 2012, 2013 and 2014. One hundred of 3-h-integrated air samples were collected at Ersa using DNPH cartridges from 29 June to 11 July 2012 at a frequency of 8 samples per day. Additionally, the ChArMEx special observation period 1b (SOP-1b) occurred from 15 July to 5 August 2013 at Ersa. More than 80 VOCs were measured during the SOP-1b intensive field campaign using different on-line and off-line techniques, which have already been presented in Michoud et al. (2017). Furthermore, formaldehyde measurements realized during the SOP-1b field campaign with DNPH cartridges are used in this study. Finally, around 70 3h-integrated air samples were collected at Ersa from 26 June to 10 July 2014 on DNPH cartridges (54 samples realized at a frequency of 4 cartridges per day from 6h-18h UTC) and on stainless steel canisters (20 samples realized at a frequency of 3 canisters per day from 9h-18h UTC). These campaign measurements will be confronted with the two years of VOC measurements investigated in the present study, in order to examine the representativeness of the study period (see Sect. 3.4.4).

### 2.2.4 Concurrent VOC measurements performed at other European background monitoring stations

From June 2012 to June 2014, VOC measurements were concurrently conducted at 17 other European background monitoring stations, allowing us (i) to examine the representativeness of Ersa station in terms of seasonal variations in VOC concentrations impacting continental Europe and (ii) to provide some insights on dominant drivers for VOC concentration variations in Europe built on Ersa's VOC observations (see Sect. 5). These European stations are part of EMEP and GAW networks. Figure 2 shows their geographical distribution. They cover a large part of western and central Europe from Corsican island in the south to northern Scandinavia in the north, are located at different altitudes (up to 3580 m a.s.l.) and most of them are categorized as GAW regional stations for Europe. More information on these stations can be found on EMEP (https://www.nilu.no/projects/ccc/sitedescriptions/index.html, last access: 03/04/2020) or GAW station information system (https://gawsis.meteoswiss.ch/GAWSIS//index.html#/, last access: 03/04/2020) sites. VOC measurements were realized by different on-line (GC or proton-transfer-reaction mass spectrometer - PTR-MS) or off-line techniques (VOCs collected by steel canisters) and were reported in the EMEP EBAS database (http://ebas.nilu.no/Default.aspx, last access: 03/04/2020).

### 2.3 Identification and contribution of major sources of VOCs

In order to characterize VOC concentrations measured at Ersa, we apportioned VOC sources in this study using the positive matrix factorization approach (PMF; Paatero, 1997; Paatero and Tapper, 1994) applied to our concentration dataset. The PMF





mathematical theory has already been presented in Debevec et al. (2017) and is therefore reminded in Section S1 of the Supplement. We used the PMF version 5.0, an enhanced tool developed by the Environmental Protection Agency (EPA) and including a multilinear engine ME-2 (Paatero, 1999), and followed the guidance on the use of PMF (Norris et al., 2014).

In order to have sufficient completeness (in terms of observation number), only VOC measurements from bi-weekly ambient air samples collected into steel canisters from 04 June 2012 to 27 June 2014 were retained in this factorial analysis. The chemical dataset includes 14 selected single or grouped VOCs, i.e. those showing significant concentration levels during the study period (see Sect. 3.3). They were divided into five compound families: alkanes (ethane, propane, i-butane, n-butane, i-pentane, n-pentane and n-hexane), alkenes (ethylene and propene), alkyne (acetylene), diene (isoprene) and aromatics (benzene, toluene, and EX, the sum of ethylbenzene, m,p-xylenes and o-xylene). The final VOC dataset encompassed 152 atmospheric data points having a time resolution of 4 hours. Input information are detailed in Table 2. Moreover, the data processing and quality analysis of the VOC dataset are presented in the supplement material of Debevec et al. (2017).

In order to optimize the PMF solution, we followed the same procedure as in Debevec et al. (2017) (see Sect. S1 in the Supplement). As a result, a five-factor PMF solution has been chosen in this study considering a $F_{peak}$ parameter fixed at 0.8 which allowed a finer decomposition of the VOC dataset following an acceptable change of the Q-value (Norris et al., 2014).

Quality indicators provided by the EPA PMF application have been indicated in Table 2. The PMF model results reconstruct on average 99% of the total concentration of the 14 selected compounds of this study. Individually, almost all chemical species also showed both good determination coefficients and slopes (close to 1) between reconstructed and measured concentrations, apart from propene, n-pentane, n-hexane, toluene and EX. Therefore, PMF model limitations to explain these species should be kept in mind when examining PMF results.

The evaluation of rotational ambiguity and random errors in a given PMF solution can be realized with DISP (displacement) and BS (bootstrap) error estimation methods (Brown et al., 2015; Norris et al., 2014; Paatero et al., 2014). As no factor swap occurred in the DISP analysis results, the 5-factor PMF solution is considered adequately robust to be interpreted. Then, bootstrapping was realized by performing 100 runs, and considering a random seed, a block size of 18 samples and a minimum Pearson correlation coefficient of 0.6. Each modeled factor of the selected PMF solution was well mapped over at least 95% of realized runs, assuring their reproducibility.

## 2.4 Geographical origins of VOCs

### 2.4.1 Classification of air mass origins

In order to identify and classify air-mass origins, back trajectories calculated by the on-line version of the HYSPLIT Lagrangian model (the Hybrid Single Particle Lagrangian Integrated Trajectory Model developed by the National Oceanic and Atmospheric Administration – NOAA – Air Resources Laboratory; Draxler and Hess, 1998; Stein et al., 2015) using Ersa as the receptor site (arrival altitude at Ersa: 600 m a.s.l.) were analysed. For each 4h-atmospheric data point of the field campaign





used for the factorial analysis, five back-trajectories of 48 h were computed using GDAS one-degree resolution meteorological data, in order to follow the same methodology as Michoud et al. (2017). The first back trajectory of a set corresponds to the hour when the air sampling was initiated (i.e. 09:00 or 12:00 UTC – see Table 1) and the 4 other ones were calculated every following hour. The time step between each point along the back-trajectories was fixed at 1 hour.

5        Then, the computed back trajectories were visually classified. Firstly, having several back trajectories per sample allows us to check if air masses transported at the station over 4 hours were globally of the same origin. As a result, samples with air masses showing contrasted trajectories (e. g., due to a transitory state between two different origins) were classified as of mixed origins and discarded (9% of the air masses). Remaining air masses were then manually classified into five trajectory clusters (marine, Corsica-Sardinia, Europe, France and Spain - Fig. 3 and Table 3) in function of their pathway when

they reached the measurement site, their residence time over each potential source region and length of their trajectories. Additionally, air masses of each cluster were sub-divided in function of their distance travelled during their 48-h course in order to highlight potential more distant sources from local ones. This sub-division is also given in Table 3 to pinpoint differences in transport times.

### 2.4.2 Identification of potential emission areas

Since the initial origin of an air mass cannot be unquestionably assessed using wind measured at a receptor site, source type contributions from the PMF were coupled with back-trajectories in order to investigate potential emission regions contributing to long-distance pollution transport to the receptor site. To achieve this, the concentration field (CF) statistical method established by Seibert et al (1994) was chosen in the present study. The CF principle has already been presented in Debevec et al. (2017) and is therefore reminded in Section S2 of the Supplement.

For each VOC observation, 3-day back-trajectories together with meteorological parameters of interest (i. e., precipitation), were retrieved from the GDAS meteorological fields with a PC-based version of the HYSPLIT lagrangian model (version 4.4 revised in February 2016), following the same methodology as those of 48 h previously presented. The arrival time of trajectories at the station corresponds to the hour when half of the sampling was done (i.e. 11:00 or 14:00 UTC – see Table 1). Note that longer back-trajectories were considered for CF analyses than those for air mass origin classification, in

order to be in the same conditions as Michoud et al. (2017).

        CF analyses applied to VOC source contributions were carried out by means of the ZeFir tool (version 3.50; Petit et al., 2017). Note that wet deposition condition was assumed when a precipitation higher than 0.1 mm was reported along the trajectory (Bressi et al., 2014). Furthermore, as also done by Michoud et al. (2017), back-trajectories have been shortened when air mass altitudes gone beyond 1500 m a.s.l. in order to discard biases related to the significant dilution impacting air

masses reaching the free troposphere. Some studies applied an empirical weighing function so as to limit the influence of high concentrations which may be observed during occasional episodes (e.g., Bressi et al., 2014; Waked et al., 2014). We preliminary tried to apply this weighing function in this study. Tests revealed that results only highlighted local contributions, given the total number of air masses considered in this study. The farther a cell is from the receptor site, the lower its





corresponding $n_{ij}$ value (number of points of the total number of back-trajectories contained in the ij$^{th}$ grid cell, Sect. S2 of the Supplement), and more the weighing function tended toward downweighting low $n_{ij}$ value. Therefore, these limitations should be taken into account when examining CF analyses, which are hence considered as indicative information.

Finally, the spatial coverage of grid cells is set from (9° W; 32° N) to (27° E; 54° N), with a grid resolution of 0.3° x
0.3°. Allocated contributions were smoothed following a factor (corresponding to the strength of a Gaussian filter) set to 5 to take into account the uncertainties in the back-trajectory path (Charron et al., 2000).

## 3 Results

### 3.1 Meteorological conditions

Seasonal variations in pollutant ambient concentrations are commonly recognized to be significantly governed by
meteorological parameters (namely temperature, total solar radiation, relative humidity and wind speed). Their monthly variations are depicted in Fig. 4. As field measurement covered a period of two years (i.e. from June 2012 to June 2014), their interannual variations are also shown in Fig. 4b.

Air temperature observed during the observation period showed typical seasonal variations, i.e. highest temperatures recorded in summer (i.e. from July to September) and the lowest ones in winter (i.e. from January to March). They were
globally in the range of normal values determined by *Météo-France* (the French national meteorological service; minimal and maximal mean values over the period 1981-2010 for Bastia available at http://www.meteofrance.com/climat/france/bastia /20148001/normales, last access: 03/04/2020). The range of temperatures recorded in June was rather expanded over the 3 years. In fact, June 2013 mean temperature was lower than June 2012 and 2014 mean ones (mean temperature of 24.7 ±5.8 °C, 19.4 ±4.1 °C and 22.5 ±5.4 °C for June 2012, 2013 and 2014, respectively), which could have influenced biogenic
emissions. Additionally, temperatures recorded during winter 2013 were colder than winter 2014 ones (mean temperature of 7.0 ±4.1 °C, and 9.7 ±1.5 °C for winter 2013 and 2014, respectively). This finding could be explained by different climatic events which have occurred during these two winter periods and have concerned a large part of continental Europe. On one hand, western European winter 2013 was considered rigorous and may be caused by a destabilization phenomenon of the stratospheric polar vortex. In early January 2013, the established stratospheric polar vortex underwent a sudden stratospheric
warming (SSW; Coy and Pawson, 2015), inducing air warming inside the vortex and a weakening of the cyclonic air circulation around the vortex. Consequently, the polar vortex was moved out of its polar position towards Europe and the SSW ended up splitting the vortex into two lobes, including one setting on western Europe and the Atlantic. These events had repercussions on the tropospheric polar vortex which also broke, collapsing several times towards Europe. All these elements modified air flux orientation from north to east, bringing cold air, and hence causing a particularly rigorous European winter. On the other
hand, most of the western European countries experienced a mild winter 2014 (Photiadou et al., 2015) characterized by its lack of cold outbreaks and nights, caused by an anomalous atmospheric circulation (Rasmijn et al., 2016; Van Oldenborgh et al.,



2015; Watson et al., 2016). In fact, the north Atlantic jet stream took a rather zonal orientation and with it the usual storm tracks shifted south. On the other side of the Atlantic Ocean, the eastern part of the USA and Canada were struck by cold polar air being advected southward due to the anomalously persistent deflection of the jet stream over the USA. The contrast between cold air advection south across the USA, and the warm tropical Atlantic was likely to have been partly responsible for the

persistence and unusual strength of the north Atlantic jet stream. This situation created ideal conditions for active cyclogenesis leading to the generation of successive strong extratropical storms being carried downstream across the north Atlantic toward the British Isles (Kendon and McCarthy, 2015).

Solar radiation also followed typical seasonal variations, with higher values recorded from May to August and lower ones in December and January. Variable solar radiations were observed in spring (i.e., from April to June) and in summer

periods. Mean solar radiation was higher by 29% in spring 2014 compared to spring 2013 one (mean solar radiation of 371 ±157 W m$^{-2}$ and 478 ±153 W m$^{-2}$ for spring 2013 and 2014, respectively) while mean solar radiation was higher by 24% in summer 2013 compared to summer 2012 one (mean solar radiation of 332 ±164 W.m$^{-2}$ and 395 ±128 W.m$^{-2}$ for summer 2012 and 2013, respectively). These radiation conditions could have affected biogenic VOC (BVOC) emissions and photochemical reactions.

Relative humidity globally followed opposite seasonal variations than temperature and solar radiation ones. In June 2012, air was dryer compared to June 2013 and 2014 mean relative humidity values (mean relative humidity of 57 ±15%, 77 ±16% and 67 ±33% for June 2012, 2013 and 2014, respectively). The wind speed did not show a clear seasonal variation over the two years studied, except maybe higher wind speeds in April and May that could induce higher dispersion of air pollutants and could advect air pollutants from more distant sources to the receptor site. May 2014 encountered particularly windy

conditions.

## 3.2 Air mass origins

Occurrences of air mass origins which have influenced Ersa throughout the observation period are indicated in Table 3. The receptor site was predominantly under the influence of continental air masses coming from Europe (corresponding to cluster 3, 31%), France (cluster 4, 26%), Corsica-Sardinia (cluster 2, 14%) and Spain (cluster 5, 5%) and to a lesser extent by air

masses of predominant marine origin (cluster 1, 15%). Each of these five clusters is mostly associated with a particular trajectory sector (e.g. south for air masses originating from Corsica and/or Sardinia) and is defined by a different transit time from continental coasts, viewed as an indicator of the potential moment when an air mass could have been contaminated for the last time (Table 3), as observed by Michoud et al. (2017). Continental air masses spent less time over the sea than marine ones. Nonetheless, transit times of continental air masses over the sea differ in function of how they are categorized. Air masses

originating from Corsica-Sardinia, France and Europe have spent 0-8 h (median values – Table 3) above the sea before reaching the receptor site, while the air masses originating from Spain have spent about 36 h. These contrasting transit times may denote distinctive atmospheric processing times for the air masses.



In particular, European and French air masses showed lower transit times over the sea (median values of 6 h and 8 h, respectively) when their trajectories are categorized as long; compared to short ones (23 h and 19 h, respectively; Table 3). Note that European and French air masses were more frequently characterized by long trajectories (20% of the air masses observed at Ersa during the studied period, for each) than short ones (11 and 6%, respectively). On the other hand, air masses
categorized as marine showed relatively close transport times between their short and long trajectories (median transit times comprised between 40 and 48 h – Table 3) and Corsican-Sardinian air masses only concerned long ones.

## 3.3 VOC mixing ratios

Descriptive statistical results for a selection of 25 VOCs, which showed significant concentration levels during the 2-yr studied period, are summarized in Table 4. These VOCs were organized into three principal categories: anthropogenic, biogenic, and
OVOCs (16, 5 and 4 targeted species, respectively). Isoprene and four monoterpenes were classified into biogenic compounds, while primary hydrocarbons (alkanes, alkenes, alkynes and aromatic compounds), were included into anthropogenic compounds, since their emissions are especially in connection with human activities. OVOCs have been presented separately, as these compounds come from both biogenic and anthropogenic (primary and secondary) sources. Although represented by only four compounds, OVOCs were the most abundant, accounting for 54% of the total concentration of selected compounds
in this study. They were mainly composed of acetone (contribution of 60% to the OVOC cumulated concentration). Anthropogenic VOCs also contributed significantly (41%) to the total concentration of measured VOCs and principally consisted of ethane and propane (which represented 34 and 17% of the anthropogenic VOC mass, respectively) as well as n-butane (7%). The high contribution of species with generally the longest lifetime in the atmosphere (see Sect. 3.4) is consistent with the remote location of the site and in agreement with Michoud et al. (2017). Biogenic VOCs only contributed little to the
total VOC concentration on annual average (5%), reaching 13% in summer. They were mainly composed of isoprene and α-pinene (contribution of 42 and 23% to the biogenic VOC mass, respectively). These compounds are among the major BVOCs in terms of emission intensity for the Mediterranean vegetation (Owen et al., 2001) and accounted for half of isoprenoid concentrations recorded during the intensive field campaign conducted in summer 2013 at Ersa (Debevec et al., 2018; Kalogridis, 2014). On the contrary, larger α-terpinene contribution was noticed during the summer intensive campaign than
the 2-yr observation period.

## 3.4 VOC variability

The variability in VOC concentration levels is governed by an association of factors involving source strength (e.g., emissions), dispersion, dilution processes and transformation processes (photochemical reaction rates with atmospheric oxidants; Filella and Peñuelas, 2006). At this type of remote site, it is also important to consider the origin of air masses impacting the site as
distant sources can play a significant role comparatively to local sources (see Sect. 3.5). Monthly and interannual variations of selected primary (anthropogenic and biogenic) VOCs along with OVOCs observed at Ersa are hence discussed in this section.


### 3.4.1 Biogenic VOCs

Variations in two selected BVOC concentrations, isoprene and α-pinene, were analysed at different timescales (monthly/interannual variations; Fig. 5). These BVOCs exhibited high concentrations from June to August, consistently with temperature and solar radiation variations (see Sect. 3.1). Indeed, throughout the summer 2013 SOP, Michoud et al., (2017) and Kalogridis (2014) observed that isoprene and α-pinene emissions were merely governed by temperature and solar radiation, considering the diurnal variations in their concentrations (Geron et al., 2000a; 2000b; Guenther et al., 2000). Furthermore, these biogenic compounds showed significant interannual variations over the two years studied, linked to temperature and solar radiation variations. For instance, higher mean concentrations of isoprene and α-pinene were noticed in June 2012 (1.0 ±1.1 and 2.6 ±1.4 µg m$^{-3}$ for isoprene and α-pinene, respectively) compared to June 2013 (0.2 ±0.2 and <0.1 µg m$^{-3}$) and June 2014 ones (0.7 ±0.5 and 0.2 µg m$^{-3}$), which may be related to the fact that temperature and solar radiation were more favourable to enhance biogenic emissions in June 2012 compared to June 2013 and 2014 meteorological conditions (Sect 3.1). Surprisingly, isoprene and α-pinene concentrations were drastically lower in July 2012 (0.3 ±0.3 and 0.9 ±0.3 µg m$^{-3}$ for isoprene and α-pinene, respectively) than June 2012 ones. Mean temperature and solar radiation were slightly lower in July 2012 and mean wind speed was slightly higher (Fig. 4), which could favour dispersion. Additionally, significant concentrations of α-pinene were noticed from September to November (Fig. 5), while isoprene concentrations were close to the detection limit and temperature and solar radiation were decreasing. This finding could be the result of a weaker degradation of α-pinene due to lower ozone concentrations observed from October to December compared to summer ones (O$_3$ concentration variations are depicted in Fig. S1 of the Supplement).

### 3.4.2 Anthropogenic VOCs

Variations of a selection of NMHCs, illustrating contrasted primary anthropogenic sources and reactivity (according to their atmospheric lifetimes), were analysed at different timescales (monthly/interannual variations; Fig. 6). Despite lifetimes in the atmosphere ranging from a few hours to some days (Atkinson, 1990; Atkinson and Arey, 2003), all selected NMHCs were characterized by almost the same seasonal variation, with an increasing winter trend followed by a decrease in spring/summer. This seasonal trend can be explained by seasonal variations in (i) emission sources (e.g., residential heating), (ii) OH concentrations, typically higher in summer inducing higher photochemical decay, and (iii) planetary boundary layer (PBL) height, inducing enhanced accumulation of VOCs in winter.

Among the selected VOCs, ethane is the species with the highest atmospheric lifetime, considering its low photochemical reaction rate with OH radicals (Atkinson and Arey, 2003). It is typically emitted by natural gas use and can be also considered as a tracer of the most distant sources. Ethane concentrations recorded at Ersa did not show any interannual variation over the two years studied, suggesting a stable influence of most distant sources. Note that ethane concentration levels were still relatively important during summer (mean concentration of 1.0 ±0.2 µg m$^{-3}$), suggesting a high importance of long-range transport contribution to VOC levels at Ersa during this season (its contribution during other seasons will be





discussed in Sect 4). Furthermore, propane, n-butane, acetylene, and benzene are characterized by photochemical reaction rates with OH radicals from four to ten times higher than ethane one (Atkinson, 1990; Atkinson and Arey, 2003) and are tracers of various anthropogenic sources (solvent use for propane, road traffic and/or residential heating for acetylene and benzene and evaporative sources for propane and n-butane; e.g., Leuncher et al., 2015). These NMHCs exhibited different concentration

levels during the two studied winter periods. Indeed, their mean concentrations during winter 2013 were from 0.1 to 0.3 µg m$^{-3}$ higher than winter 2014 ones. Additionally, CO covaried well with acetylene and benzene, as expected for combustion tracers of medium-to-long lifetimes (Figs. 6 and S1 of the Supplement). Ethylene has the lowest lifetime among the selected species depicted in Fig. 6 (considering its photochemical reaction rate with OH radicals referred in Atkinson and Arey, 2003), it is considered as typically emitted by combustion processes and can be used as tracer of local sources (e.g., Sauvage et al., 2009).

Even ethylene showed higher concentrations during winter 2013 compared to winter 2014 ones (mean ethylene concentration of 0.6 ±0.2 and 0.3 ±0.1 µg m$^{-3}$ during winter 2013 and 2014, respectively). As a result, winter variations of concentration levels concerned at a time close sources and more distant ones and will be more investigated thereafter (Sect. 4.2).

### 3.4.3 Oxygenated VOCs

Variations of selected OVOCs were analysed at different timescales (monthly/interannual variations; Fig. 7). Firstly,

formaldehyde and acetaldehyde are mainly produced through the chemical transformation of anthropogenic and biogenic VOCs (Rottenberger et al., 2004; Seco et al., 2007), particularly in clean and remote areas. One of the main precursor hydrocarbons for formaldehyde is thought to be isoprene along with methane, methanol and acetaldehyde (Schade and Goldstein, 2001; Wolfe et al., 2016). Additional sources of formaldehyde are industrial processes, motor exhausts and forest fires (Seco et al., 2007, and references therein). According to the budget estimates of Millet et al. (2010), the largest

acetaldehyde source is provided by hydrocarbon oxidation, essentially alkanes and alkenes as well as isoprene and ethanol. Nevertheless, acetaldehyde is only produced as a second or higher-generation oxidation product of isoprene for all its reaction pathways with atmospheric oxidants (Millet et al., 2010). Besides photochemical production, acetaldehyde can also be released by terrestrial plants, in the process of ethanol production in leaves and roots following fermentation reactions (Jardine et al., 2008; Rottenberger et al., 2008; Winters et al., 2009). Formaldehyde and acetaldehyde have relatively short lifetime into the

atmosphere, considering their photochemical reaction rate with OH radicals (9.37 10$^{-12}$ and 15 10$^{-12}$ cm$^3$ molecule$^{-1}$s$^{-1}$, respectively - Atkinson and Arey, 2003) and hence they can be induced by relatively close sources. Formaldehyde and acetaldehyde concentrations showed clear seasonal variations (Fig. 7); with high summer and spring concentrations (mean concentrations of 2.1 ±1.5 and 1.1 ±0.5 µg m$^{-3}$, respectively). During summer, despite their lower lifetime, these OVOCs can be significantly induced by secondary and biogenic sources, which can explain their higher concentrations. During the summer

field campaign of 2013 conducted at Ersa, the OH reactivity showed notable diurnal profile consistent with air temperature one, which can denotes that BVOCs, including secondary species, were greatly influencing the local atmospheric chemistry (Zannoni et al., 2017). These findings are in agreement with a large result on BVOC oxidation on the local photochemistry. Moreover, formaldehyde and acetaldehyde concentration levels remained relatively significant during fall (i.e., from October





to December) and winter (mean concentration of 0.9 ±0.5 and 0.8 ±0.3 µg m⁻³, respectively), suggesting a significant contribution of anthropogenic (primary and/or secondary) sources. Additionally, these OVOCs showed significant interannual variations. In late spring and summer, their concentrations may be dependent on temperature and solar radiation variations which can influence biogenic emissions. For instance, formaldehyde and acetaldehyde mean concentrations were higher in

June 2012 (2.9 ±0.4 and 1.4 ±0.4 µg m⁻³ for formaldehyde and acetaldehyde, respectively) compared to June 2014 ones (2.2 and 0.8 µg m⁻³), in agreement with biogenic concentration variations previously discussed in Sect 3.4.1. Formaldehyde and acetaldehyde concentrations were lower in August 2012 (2.0 ±0.6 and 1.2 ±0.2 µg m⁻³ for formaldehyde and acetaldehyde, respectively) than August 2013 ones (3.6 ±1.5 and 1.5 ±0.6 µg m⁻³), in agreement with isoprene and α-pinene concentration variations (Fig. 5) and since meteorological conditions in August 2013 were more favorable to photochemical processes (Fig.

4). Furthermore, formaldehyde and acetaldehyde exhibited concentrations in February 2013 (1.5 ±0.7 and 1.2 ±0.3 µg m⁻³, respectively) twice higher than February 2014 ones (each at 0.7 ±0.1 µg m⁻³), consistent with NMHC concentration variations (Fig. 6).

Identified acetone sources include primary emissions from both biogenic (green plant and litter sources) and anthropogenic origins, but its emissions is thought to be globally of biogenic rather than anthropogenic origin (Goldstein and

Schade, 2000; Schade and Goldstein, 2006). Acetone can also be induced by secondary (biogenic/anthropogenic) sources from VOC oxidation (e.g. propane, i-butane, i-pentane, monoterpenes and methylbutanol) and biomass burning (Goldstein and Schade, 2000; Jacob et al., 2002; Singh et al., 2004). Acetone is the OVOC of the selection with generally the highest atmospheric lifetime, considering its photochemical reaction rate with OH radicals ($1.7 \ 10^{-13}$ cm³ molecule⁻¹s⁻¹ - Atkinson and Arey, 2003), and hence acetone can result from distant sources and/or be formed within polluted air masses before they reach

the receptor site. As a result, distant sources can significantly contribute to its concentrations at Ersa. Actually, roughly half of acetone concentrations measured at diverse urban or rural sites have been assigned to regional background pollution by several studies (e.g., Debevec et al., 2017; de Gouw et al., 2005; Legreid et al., 2007). Acetone showed similar seasonal variations than formaldehyde and acetaldehyde, i.e. high concentrations during spring and summer (mean acetone concentration of 5.4 ±3.0 µg m⁻³) suggesting that biogenic sources as well as secondary sources significantly contributed to its atmospheric

abundance at Ersa. Acetone concentrations remained significantly high during winter and fall (mean concentration of 3.2 ±1.6 µg m⁻³ and lowest ones not below 1.5 µg m⁻³), which can be potentially explained by significant contributions of anthropogenic sources and regional background pollution during these seasons. Similarly as formaldehyde and acetaldehyde, acetone showed significant interannual variations both in summer (mean concentrations in summer 2012 lower from 1.6 µg m⁻³ than summer 2013 one, related to meteorological parameter and biogenic VOC variabilities) and winter (mean concentrations in winter 2013

higher from 1.7 µg m⁻³ than winter 2014 one, related to NMHC variability).

Methyl ethyl ketone (MEK) can be emitted by terrestrial vegetation or by numerous anthropogenic sources, such as biomass burning, solvent evaporation as well as vehicle exhaust, and can be produced by the atmospheric oxidation of other VOCs like n-butane (Yáñez-Serrano et al., 2016 and references therein). MEK is characterized by photochemical reaction rates with OH radicals ten times higher than acetone one ($1.22 \ 10^{-12}$ cm³ molecule⁻¹s⁻¹ - Atkinson and Arey, 2003) but admitted





low enough to allow advection to the receptor site of MEK released by distant pollution sources or its formation during the transport of polluted air masses. MEK showed distinct variations from other OVOC ones discussed in this section. Indeed, MEK concentrations did not show seasonal variations except an increasing winter trend (mean concentration of 0.4 ±0.1 µg m$^{-3}$). This finding suggests that anthropogenic (primary and secondary) sources significantly contribute to MEK concentrations.

As observed for most NMHCs in Sect. 3.4.2, MEK exhibited different concentration levels during the two studied winter periods since its mean concentration in February 2013 was by 0.2 µg m$^{-3}$ higher than its February 2014 one.

### 3.4.4 Comparisons with other VOC measurements performed at Ersa

The comparison between the VOC monitoring measurements as investigated in this study with concurrent campaign measurements performed during the summers of 2012, 2013 and 2014 is detailed in Section S3 of the Supplement. The purpose

is to examine the representativeness of the 2-yr observation period in terms of summer concentration levels. As a synthesis of these comparisons, VOC concentration levels and variations of the three summer field campaigns were globally in consistency with those previously described in this study. This finding can suggest that the annual temporal coverage of VOC measurements realized over the two years was sufficiently adapted to well characterize VOC concentration variations (at seasonal scale). However, campaign data in the Supplement show that BVOC day to day variations can be significant especially

in summer, as established by Kalogridis (2014) and Michoud et al. (2017). Interpretation of interannual variations of BVOC measurements is based on a limited number of sampling days during the studied period, it should then be considered cautiously.

### 3.5 VOC factorial analysis

In the coming section, the PMF solution composed of 5 factors (from simulations presented in Sect. 2.3) is described and examined. Figure 8 depicts factor contributions to the species chosen as input for the PMF tool along with VOC contribution

to the 5 factors defined by the factorial analysis. Figures 9 and 10 show PMF factor contribution time series and their seasonal and interannual variations, respectively. Note that winter variations will be investigated thereafter (Sect. 4.2). In the present section, lifetimes were assessed from kinetic rate constants of the reactions of selected VOCs with OH (Atkinson and Arey, 2003) given an average OH concentration of 0.5 $10^6$ and 2.5 $10^6$ molecules cm$^{-3}$ in winter and summer, respectively (Spivakovsky et al., 2000).

As VOC concentrations arised from direct emissions, chemistry, transport and mixing, each individual computed factor cannot be attributed solely to one source category, especially for such a remote receptor site as Ersa. A part of them may not be precisely associated with emission profiles but should rather be explained as aged profiles originating from several sources assimilating to several source categories (Sauvage et al., 2009). PMF analysis was hence performed to define co-variation factors of VOCs that were characteristic of aged or local primary emissions along with secondary photochemical

transformations taking place during the transport of air masses observed at this remote site (Michoud et al., 2017).



### 3.5.1 Biogenic source (factor 1)

The average contribution of factor 1 to the sum of measured VOC concentrations is of 0.2 ±0.4 µg m$^{-3}$ on average during the observation period (corresponding to 4% of the sum), peaking up at 3.1 µg m$^{-3}$ on 20 June 2012. In late spring/summer, it was one of the main factors observed (16% on average and up to 53%; Fig. 9). The chemical profile of factor 1 depicts an elevated

contribution of isoprene, recognized as a chemical marker for biogenic emissions, having its variability fully related to this factor. The relative load of this VOC for the factor 1 is 70%. The estimated lifetime of isoprene in the troposphere was quite short (winter: 5.6 h and summer: 1.1 h), indicating that this compound was emitted mostly by local vegetation. Consequently, factor 1 is labelled "biogenic source".

Average seasonal contributions exhibited a seasonal cycle (Fig. 10a1) with high values in summer (July-September

mean contribution of 0.5 ±0.4 µg m$^{-3}$) and spring values (April-June mean contribution of 0.3 ±0.6 µg m$^{-3}$), in agreement with isoprene variability investigated in Sect. 3.4.1. Biogenic emissions were directly related to ambient temperature and solar radiation (Sect. 3.4.1.), inducing these factor 1 contribution variations. As already observed for isoprene, factor 1 contributions showed significant interannual variations over the two years studied (Fig. 10b1), confirming that the biogenic source strength was dependent on meteorological conditions (Fig. 4).

### 3.5.2 Short-lived anthropogenic sources (factor 2)

19% of the sum of measured VOCs was attributed to factor 2. This latter is mainly consisted of primary anthropogenic compounds, such as toluene (73% of its variability attributed to this factor), C$_8$ aromatic compounds (EX; 93%), ethylene (48%) and propene (83%), typically emitted by combustion processes and with short-to-medium lifetime (winter: 24 h-4.1 days; summer: 5-20 h), with an average contribution to the sum of measured VOC concentrations from this factor of 66%.

Note that factor 2 did not show a good correlation with CO (Pearson correlation coefficient only of 0.2). Besides road traffic, toluene is also a good marker for solvents generated by industrial sources (Buzcu and Fraser, 2006; Leuchner et al., 2015; Zhang et al., 2014), suggesting that this profile could also be imputed to industrial sources. Additionally, a significant proportion of C$_5$-C$_6$ alkanes, i.e. i-pentane (32% explained), n-pentane (37%) and n-hexane (51%), typically emitted by gasoline evaporation and with medium lifetime (winter: 4-6 days; summer: 21 h-1.3 day), also contributes to this factor by

19%. Factor 2 is hence attributed to short-lived anthropogenic sources, partly related to gasoline combustion and/or evaporation and solvent use.

Average seasonal contributions showed slightly higher contributions during fall (October-December mean factor 2 contribution of 1.3 ±0.6 µg m$^{-3}$) and winter (January-March: 1.1 ±0.7 µg m$^{-3}$; Fig. 10a2). As a consequence of lower available UV light and temperatures, OH concentrations decrease in fall and winter, inducing reduced chemical reaction rates.

Consequently, VOCs were not depleted as rapidly as in spring/summer months. Moreover, the PBL height was significantly lower in winter, conducting to less dilution of emissions, favouring relative accumulation of pollutants, and so increasing VOC concentrations. However, factor 2 contributions were also significant in spring and summer (mean factor 2 contributions of



0.9 ±0.4 µg m$^{-3}$, each), which could illustrate an enhanced evaporation of gasoline, solvent inks, paints and additional applications during these months as a result of higher temperatures. Interannual comparison of mean monthly factor 2 contributions (Fig. 10b2) pointed out no clear seasonal variation over the study period, suggesting this source was of different origins during the two years studied, probably related to air mass origin occurrences and trajectories (discussed in Sect. 4.2).

Regarding factor 2 contributions coupled with air mass clusters (Fig. 11), more elevated contributions were noticed under the influence of continental air masses coming from France and Europe. The distinction of short-trajectories from long ones (see Sect. 3.2) highlighted that factor 2 was potentially influenced by relatively close sources when Ersa received air masses from continental France, whereas other continental European sources were probably more distant. Furthermore, CF analysis applied to factor 2 contributions (see Fig. S2 of the Supplement) confirmed that this factor was influenced by various potential emission

areas, either located in Italy (the Po Valley and Central Italy), France (southeast region) or possibly in central Europe (western Hungary, Croatia and Slovenia). Ship emission contribution cannot be discarded as well, as already suggested by Michoud et al. (2017). Indeed, ship emissions are predominantly composed of light alkenes, aromatic compounds and heavy alkanes (> C$_6$ compounds; Eyring et al., 2005).

### 3.5.3 Evaporative sources (factor 3)

The average contribution of factor 3 to the sum of measured VOC concentrations is approximately estimated at 1.2 ±1.0 µg m$^{-3}$ (22% of their sum) during the studied period. The profile of this anthropogenic factor displays an important contribution from alkanes, principally i-/n-butanes (having lifetimes of 10-11 days in winter, and ~2 days in summer) and with more than 69% of their variabilities explained by factor 3, along with i-pentane (50%), n-pentane (59%), n-hexane (42%) and propane (43%; lifetimes of 4-21 days in winter, and 21 h-4 days in summer). The C$_3$-C$_6$ alkanes are identified in the gasoline

composition and evaporation sources (storage, extraction and distribution of gasoline or liquid petroleum gas - LPG; Sauvage et al., 2009 and references therein). Additionally, propane can be viewed as a relevant profile signature of natural gas use (Leuncher et al., 2015). The cumulated contribution of these VOCs to factor 3 is up to 88%. As a result, this factor can be viewed as "evaporative sources".

Average seasonal contributions exhibited a seasonal cycle (Fig. 10a3) with high winter and fall values (mean

contribution of 2.2 ±0.7 and 1.9 ±1.0 µg m$^{-3}$, respectively). During the cold season, evaporative emissions can be expected to be lower. But, as a result of lower OH concentrations and weaker solar radiation in this period than in summer, the chemical lifetimes of the involved compounds were intensified in winter and fall, which may have favoured their advection to the site and their accumulation. Factor 3 contributions and CO concentrations were quite correlated especially in winter and fall (Pearson correlation coefficient of 0.6), similarly as factors 3 and 2 (0.5). These findings suggest that the high contributions in

these seasons of evaporation tracers may be related to the combustion processes as well and factors 2 and 3 can be partly related to the same sources. Mean factor 3 contributions were only 0.5-0.6 µg m$^{-3}$ in spring and summer. This low contribution can be partly explained as it was mainly composed of VOCs with medium lifetimes which may rapidly react during these seasons before reaching the receptor site. Interannual comparisons of mean monthly factor 3 contributions (Fig. 10b3) pointed





out a clear and reproducible seasonal variation over the two years studied. This finding could suggest that this factor was largely influenced by chemical processes and regional contributions. Regarding factor contributions as a function of air mass clusters (Fig. 11), more elevated contributions were noticed when aged air masses originated from France and Europe and probably transported toward Ersa by relatively distant sources. According to the CF analysis applied to factor 3 contributions (Fig. S2 of the Supplement), the Po Valley (especially Emilia-Romagna, an Italian region centre for food and automobile production), central Italy, the southeast of France and the Sardinian region seemed to be identified as main potential emission areas for factor 3, as well as possibly more distant areas in central Europe such as western Hungary (i.e. western Transdanubian region specialised in automotive and machinery industries).

### 3.5.4 Long-lived combustion sources (factor 4)

The average contribution of factor 4 to the total VOC concentration is roughly evaluated at 0.9 ±0.7 µg m$^{-3}$ (16% of the sum) on average during the observation period. Its profile displays an important contribution from acetylene (100% explained), benzene (49%) and propane (37%), with lifetimes of 19-26 days in winter and of 4-5 days in summer and with an average total contribution to the sum of measured VOC concentrations from this factor of 80%. Aromatic compounds and acetylene are generally attributed to combustion sources, such as vehicle exhaust (e.g., Badol et al., 2008; Pang et al., 2014). However, factor 4 is characterized by a loading of benzene much more superior to toluene one (49 and 3 %, respectively), suggesting it is more related to a residential heating source than a traffic one (Elbir et al., 2007; Leuchner et al., 2015; Sauvage et al., 2009). Factor 4 profile, mainly composed of long-lived species together with a low contribution of shorter-lived species, may indicate partly aged air masses advected towards the measurement site. These suggestions are consistent with the fact that this factor correlated particularly well with CO (Pearson correlation coefficient of 0.8). As a result, this factor can be viewed as "long-lived combustion sources", including residential heating.

Average seasonal contributions (Fig. 10a4) exhibited a clear seasonal cycle with intense winter values and really low summer ones (mean contribution of 1.8 ±0.8 and 0.3 ±0.1 µg m$^{-3}$, respectively). This factor 4 strength variation is consistent with an elevated use of heating systems during wintertime due to typical low temperatures. Moreover, factor 3 showed good correlation with factor 4 in winter and fall (Pearson correlation coefficient of 0.5), suggesting a contribution of these combustion sources to evaporative sources. A clear and reproducible seasonal variation of factor 4 contributions was observed over the two years studied (Fig. 10b4) that could suggest this factor was largely influenced by chemical processes and regional contributions, along with source strength. Furthermore, factor 4 showed higher contributions when the Ersa station received European air masses (see Fig. 11), especially by ones having long trajectories. The CF analysis depicted in Fig. S2 of the Supplement only pointed out western Hungary and to a lesser extent the Po Valley, as main potential emission areas for factor 4. Note that the CF analysis mostly highlighted factor contribution origins observed in wintertime (detailed in Sect. 4). As a reminder, the interannual variations observed in winter will be analysed in section 4.2.



### 3.5.5 Regional background (factor 5)

Contributing at 39% to the total concentration of measured VOCs, factor 5 corresponds to the dominant VOC source detected at Ersa during the study period. The profile of this factor is principally dominated by ethane, having its variability fully explained by factor 5, and is also composed of propane (18% explained). These compounds, with lifetimes of 21-93 days in

winter and of 4-19 days in summer, typically result from the use of natural gas and their contribution to factor 5 was up to 96%. Additional anthropogenic VOCs with shorter lifetimes are attributed to this factor, including ethylene (16% explained) and propene (12%; lifetime of 21 h to 3 days in winter and of 4-13 h in summer) despite a low contribution to factor 5 mass (~3%). Hence, the high abundance of long-lived species may result here from aged air masses advected to the study site. Consequently, factor 5 can be viewed as a regional contribution of diverse remote sources of the Mediterranean region, thus

indicating the continental regional background (Hellén et al., 2003; Leuchner et al., 2015; Sauvage et al., 2009). These sources were advected towards the sampling site by aged air masses, which have not been recently in contact with supplementary anthropogenic sources. Within the time of emission transport from distant sources, atmospheric oxidation depletes a large proportion of the reactive species and the remaining fraction is mainly constituted of the less-reactive VOCs, like ethane and propane. As a result, we associate factor 5 with the "regional background".

15        Mean seasonal contributions (Fig. 10a5) exhibited a characteristic feature for this source with a maximum in winter (mean contribution of 3.3 ±0.6 µg m$^{-3}$) in link with tropospheric accumulation, succeeded by a decline in spring and summer (mean contribution of 2.1 ±0.7 and 1.2 ±0.2 µg m$^{-3}$, respectively) probably related to photochemical decay and dilution processes. As already observed for ethane (Sect. 3.4.2), factor 5 contributions did not show any interannual variation over the two years studied (Fig. 10b5), confirming that this factor was largely influenced by chemical and dilution processes and long-

range transport. Note that factor 5 showed good correlation both with factor 4 and CO (Pearson correlation coefficient of 0.8 and 0.7, respectively), which can suggest similar origins. Mean factor 5 contributions in function of air mass origins were in the same range, except that more elevated contributions were noticed under the influence of European air masses (especially those potentially connected to distant contributions; Fig. 11) compared to the ones related to others continental origins. As expected, the CF analysis applied to factor 5 contributions did not clearly pinpoint a specific potential emission area (Fig. S2

of the Supplement), apart from maybe western Hungary and to a lesser extent the Po Valley, which are areas experiencing high anthropogenic emissions.

### 3.6 Towards the best experimental strategy to characterize variation in VOC concentrations

The 5-factor PMF solution, modelled with a two year VOC dataset (from June 2012 to June 2014) was compared with the PMF solution modelled with the short summer SOP-1b VOC dataset (from 15 July to 5 August 2013), composed of 6 factors,

namely primary biogenic factor, secondary biogenic factor, short-lived anthropogenic factor, medium-lived anthropogenic factor, long-lived anthropogenic factor and oxygenated factor (Michoud et al., 2017). Note that the SOP-1b PMF source apportionment was performed considering a dataset composed of 42 VOCs, comprising six oxygenated compounds and





collected with three different on-line techniques (see Sect. S3 of the Supplement). Chemical profile, variability and origin of these summer factors were examined in Michoud et al. (2017). Benefiting from these results, the two source apportionment analyses can be confronted to evaluate the representativeness of their source composition and contributions. Comparison results are hence presented in Sect. S4 of the Supplement, supporting the investigation of the contribution of both experimental strategies to characterize the main sources influencing VOC levels observed at the receptor site of long-range transported pollution impacting the western Mediterranean region. On one hand, the SOP-1b intensive field campaign occurred in summer and offered good conditions to (i) monitor at a specific period anthropogenic sources, influenced by several geographic origins along with biogenic local sources and secondary oxygenated sources and (ii) to assess their diurnal variations. On the other hand, the 2-yr monitoring period had the advantage to cover seasonal and interannual variations of main primary sources impacting VOC concentrations observed at the receptor site. Globally, sources identified as influencing VOC concentrations at Ersa had similar chemical compositions. The longer time scale of VOC measurements (i.e. the 2-yr period) presented here helped to deconvolve long-lived combustion sources from regional background. Nevertheless, the time resolution of VOC measurement of the 2-yr period (4 hours compared to 1 hour and a half during the SOP-1b period) and the limited number of sampling days during this study period did not help to support the clear deconvolution of the 5 factors, as factors related to anthropogenic sources were quite correlated between them (as a consequence of their seasonal variations – see Sect. 3.5; and unlike SOP-1b anthropogenic sources showing between them Pearson correlation factors from -0.5 to 0.1). Finally, the incorporation of OVOCs in the source apportionment had little impact on the identification of main primary sources influencing VOC concentrations observed at the receptor site but can modify their relative contributions, emphasizing the contribution of local biogenic/anthropogenic sources and decreasing the contribution of regional anthropogenic sources.

## 4. Discussions on the seasonal variability of VOC concentrations

### 4.1 Determination of controlling factors

In this section, source contributions are examined regarding their seasonal variations so as to identify the prevailing drivers for VOC concentration variations. Benefiting from two years of observations, the discussion will also be focused on interannual variations. Seasonal accumulated contributions of the main sources identified as contributing to the total VOC mass observed at Ersa during the study period were summarized in Fig. 12. Note that spring 2012 data are not presented in Fig. 12, since the monitoring period started in June 2012 and did not cover the whole season. During this particular month, the biogenic source was determined to be the largest contributor to the cumulated VOC concentrations (34% of the total VOC mass explained) since its emissions were enhanced by the high temperatures monitored (Fig. 4).

Firstly, low total contributions of the five VOC factors were observed during summer and spring periods (mean total contributions of 3.4 ±0.8 and 4.4 ±0.9 µg m$^{-3}$, respectively). The regional background and the short-lived anthropogenic sources were identified as the largest contributors to the total VOC concentrations monitored at Ersa in spring and summer. The regional background contributed from 30 to 55% of the total VOC concentration observed at the Ersa station in these


seasons. Especially since this source is principally constituted of long-lived compounds, it suggests that aged air masses advected to the study site significantly influenced VOC concentrations observed during these seasons. Additionally, short-lived anthropogenic sources explained from 19 to 30% of the total VOC mass. As the short-lived anthropogenic sources were composed of VOCs with short to medium lifetimes (Sect. 3.5.2), they can be rapidly depleted in spring and summer.

Additionally, PBL height can be higher in these seasons (von Engeln and Teixeira, 2013), favouring phenomena of vertical dispersion. However, their contributions were elevated, which can be probably attributed to high influences of ship transport and relatively close potential emission areas (e.g. Italian coastline areas – see Fig. S2 of the Supplement). Evaporative and long lived combustion sources only contributed from 7 to 17% in spring and summer. Evaporative sources were mainly composed of VOCs with medium lifetimes along with of regional origins (Sect. 3.5.3), and hence, during these seasons, these

compounds can have reacted and/or been dispersed before reaching the receptor site. Low contributions of long-lived combustion sources can be partly explained by a lower source strength.

Looking now at the interannual variations during summer and spring periods, Fig. 12 highlights that the total contributions of the five VOC factors were in the same range during the two summer periods as well as the two spring ones (absolute difference was of 0.3 µg m$^{-3}$ between summer 2012 and 2013 total contributions and below 0.1 µg m$^{-3}$ between

spring 2013 and 2014 ones). As depicted in Fig. 10, monthly contributions of evaporative sources, long-lived combustion sources, and regional background, were in the same range and followed the same variation from April to September over the studied period. This finding can suggest that these sources were largely influenced by chemical processes, vertical dispersion phenomena and regional contributions in spring and summer. Contributions of short-lived anthropogenic sources were in the same range during spring and summer (0.7-1.1 µg m$^{-3}$; Fig. 12) but did not seem to follow a specific variation (Fig. 10). This

finding suggests that these sources were largely influenced by origins of air masses which advected to Ersa some relatively close source emissions, potentially of variable strength. As a result, main parameters influencing VOC concentrations in spring and summer were meteorological conditions (i.e. high temperatures/solar radiation enhancing biogenic source contributions), OH concentrations (typically high especially in summer) inducing higher photochemical decay, PBL height (typically high in summer) favouring vertical dispersion of pollutants and long-range transport.

During fall and winter periods, total contributions of the five VOC factors depicted in Fig. 12 were comprised between 6.1 and 9.4 µg m$^{-3}$. Several parameters can explain these levels. Firstly, chemical reaction rates dropped in fall and winter as a consequence of decreased OH concentrations owing to lower available UV light and temperatures (Fig. 4). Hence, VOCs were not removed from the atmosphere as quickly as in the summer/spring months. PBL height also decreased during these seasons, impacting vertical dispersion phenomena. Moreover, the regional background and evaporative sources were identified

as the dominant contributors to the VOC concentrations collected at Ersa in fall and winter (contribution of 35-45% and of 24-33% to the total VOC concentration, respectively), suggesting that regional contributions significantly influenced Ersa's VOC concentrations during these seasons. Long-lived combustion sources also contributed significantly to VOC concentrations specifically in winter (explaining 18-24% of the total VOC concentration) since the typical low ambient temperatures during these seasons (Fig. 4) may involve an increased use of residential heating. To better identify regional influences, average



seasonal contributions of the anthropogenic sources were investigated in function of air mass origins in Fig. 13. During fall and winter, the receptor site was mostly influenced by continental air masses coming from Europe and France. Anthropogenic sources showed higher accumulated contributions when European air masses were advected to Ersa (Figs. 11 and 13), and could be attributed to potential emission areas located in Italy (the Po Valley and Central Italy) and possibly more distant potential emission areas in central Europe (western Hungary, Croatia, Slovenia - Fig. S2 of the Supplement). To a lesser extent, high anthropogenic source contributions were also noticed when Ersa received air masses originating from continental France (potential emission areas located in the southeast of France) and Corsica-Sardinia.

Concerning the interannual variations during fall and winter periods, Fig. 12 shows that total contributions of the five VOC factors were in the same range during the two fall periods (absolute difference of 0.4 µg m$^{-3}$ between fall 2012 and 2013 total contributions). Contrariwise, the five factors total contribution in winter 2013 (9.4 ±2.5 µg m$^{-3}$) was higher by 1.9 µg m$^{-3}$ than winter 2014 one on average. Looking now at source contributions during these two winter periods individually, regional background contributions were in the same range (absolute difference below 0.1 µg m$^{-3}$; Fig. 12), while contributions of long-lived combustion sources, short-lived anthropogenic sources and to a lesser extent evaporative sources were higher during winter 2013 compared to winter 2014 ones (absolute difference from 0.3 to 0.9 µg m$^{-3}$; Fig. 12).

## 4.2 The particular case of winter

The difference of contributions between the two winter periods could be partly explained by air mass origin occurrences and their respective contributions. On one hand, Ersa was more under the influence of European air masses during winter 2013 than during winter 2014 (occurrences of 37 and 18%, respectively). When continental European and French air masses were advected to Ersa, anthropogenic sources showed higher accumulated contributions in winter 2013 (10.8-9.6 µg m$^{-3}$, respectively; Fig. 13) compared to winter 2014 ones (8.2-7.8 µg m$^{-3}$). On the other hand, the station more frequently received air masses originating from Corsica-Sardinia in winter 2014 than in winter 2013 (occurrences of 24 and 0%, respectively) and hence was influenced by closer anthropogenic sources in winter 2014. Otherwise, the average accumulated anthropogenic source contribution associated to Corsican-Sardinian influence was only 7.0 µg m$^{-3}$ in winter 2014 (Fig. 13).

Moreover, Fig. 14 presents the CF analyses realized for the 4 anthropogenic sources using only winter 2013 and 2014 observations. These analyses globally showed that Ersa station was influenced by air masses of different potential origins in winters 2013 and 2014. During winter 2013, the main potential emission areas for the 4 anthropogenic sources were located in Italy (Tuscan coasts and the Po Valley), central Europe (Slovenia and western Hungary) and to a lesser extent the south of France. On the other hand, VOC concentrations observed at Ersa during winter 2014 were mostly influenced by contributions from relatively close (but rather low) potential emission areas located in Corsica and Sardinia (Fig. 14). Moreover, in winter 2014, Ersa did not seem to be influenced at all by air masses originating from central Europe (especially from Slovenia and Hungary - Fig. 14), that could partly explain the difference of VOC concentrations observed at the receptor site during the two winter periods. Surprisingly, potential emission areas located in the Po Valley and the southeast of France, known to experience high anthropogenic emissions, did not seem to have contributed significantly to VOC concentrations in winter 2014.



Even though contributions of long-lived combustion sources, short-lived anthropogenic sources and evaporative sources were significantly reduced in winter 2014 compared to winter 2013 ones (Sect. 4.1), the shape of their variations remained similar in winters 2013 and 2014, as depicted in Fig. 9, despite different potential origins have influenced VOC concentrations monitored at Ersa between these two winter periods. These findings could be an evidence of homogenous

regional background pollution distribution at synoptic scale. Mean regional background contributions monitored in winters 2013 and 2014 (Sect. 4.1) are also in agreement with this suggestion. Moreover, the different amplitudes of anthropogenic source contributions observed between the two winter periods may result in different influences of meteorological conditions. These latter can have affected anthropogenic source emission strengths as well as chemical transformations occurring inside air masses all along their transport to the receptor site.

To support these suggestions, we can notice that most countries of western Europe experienced different winters in 2013 and 2014, induced by different climatological events occurring during these two winter periods (see Sect. 3.1). Winter 2013 was considered rather rigorous (Fig. S3 of the Supplement), since e.g., French temperatures were lower up to 1-1.5 °C than average value for 1981-2010 according to Météo France (http://www.meteofrance.fr/climat-passe-et-futur/bilans-climatiques/bilan-2013/bilan-de-lhiver-2012-2013, last access: 03/04/2020). On the other hand, winter 2014 was rather mild

and temperatures were the hottest of the 1981-2010 period (mean temperature for Europe reached 11.2 °C, i.e. ~1 °C higher than the normal value; Photiadou et al., 2015 and Fig. S3 of the Supplement). This difference of temperatures between the two winters studied could have affected OH concentrations. Indeed, meteorological conditions in winter 2014 were probably more favourable to induce higher OH concentrations than in winter 2013 ones, leading to higher photochemical decay, and so lower VOC concentrations observed at the receptor site. Higher temperatures along with the lack of cold nights in winter 2014

(Photiadou et al., 2015) may also have affected the source strength especially of long-lived combustion sources in winter 2014. Furthermore, rain event intensities and occurrences in winters 2013 and 2014 could also impact enrichment in anthropogenic sources of air masses advected to Ersa, and hence VOC concentrations observed at the receptor site. Note that in northern Italy a very high monthly rainfall was recorded in winter 2014 (higher by 300% than the seasonal normal value for 1981-2010; see Fig. S3 of the Supplement). Abundant rainfalls were also noticed in southeast of France during winter 2014 (the highest one

recorded over the 1959-2014 period, according to Météo France; http://www.meteofrance.fr/climat-passe-et-futur/bilans-climatiques/bilan-2014/bilan-climatique-de-l-hiver-2013-2014, last access: 03/04/2020). As a consequence, these meteorological conditions should have reduced anthropogenic source contributions from the Po Valley and the southeast of France in winter 2014.

As a summary, the main parameters governing VOC concentration variations in winter seem to be the emission

strength of the main anthropogenic sources identified in this study, and the continental regional background level constrained by external parameters, i.e. dispersion phenomena (long-range transport, enrichment in anthropogenic sources of continental air masses advected to the site as well as air mass origin occurrences) and pollutant depletion (in relation to the oxidizing capacity of the environment). This study also highlights that meteorological conditions can significantly affect the importance of these parameters in controlling VOC concentration variations in winter. As a consequence, this finding also point out that


shorter observation periods (i.e., up to two months) may be reflected the variability of the identified parameters under the specific meteorological conditions of the studied period.

## 5. VOC concentration variations in continental Europe

From June 2012 to June 2014, VOC measurements were concurrently conducted at 17 other European background monitoring stations (Sect. 2.2.4), allowing us (i) to examine the representativeness of Ersa station in terms of seasonal variations in VOC concentrations impacting continental Europe and (ii) to provide some insights on dominant drivers for VOC concentration variations in Europe built on what we have learned from Ersa's VOC observations. Figure 15 depicts monthly concentration time series of a selection of NMHCs measured at the 18 considered European monitoring stations (including Ersa).

NMHCs with typically medium-to-long lifetimes in the atmosphere, i.e. ethane, propane, n-butane, acetylene and benzene (Sect 3.4.2), were examined here since their concentrations can be significantly influenced by regional contributions (Debevec et al., 2017; Michoud et al., 2017; Sauvage et al., 2009). Globally, these selected anthropogenic VOCs measured at Ersa showed the same seasonal variations as observed at other European stations (Sect 3.4.2), i.e. with an increasing winter trend followed by a decrease in spring/summer and hence assuring the representativeness of Ersa station for monitoring regional pollution in Europe. As a reminder, concentrations observed at Ersa were mainly explained by regional background for ethane, by long-lived combustion sources for propane, acetylene and benzene, and by evaporative sources for n-butane (see Section 3.5 and Fig. 8). As a result, the study of concentration variations of these source tracers may help highlighting temporal and spatial variations in source contributions to VOC concentrations observed in most continental Europe. In addition, despite its shorter lifetime compared to other VOCs of the selection, ethylene concentration variations were also be taken into account in this study to investigate short-lived anthropogenic source importance and variability in continental Europe.

Monthly NMHC concentrations were globally lower and relatively homogeneous from June to August whatever the location and the typology of the considered station (the highest absolute difference between anthropogenic VOC concentrations measured at two stations in summer was of 0.4-0.7 µg m$^{-3}$ for ethane, 0.1-0.2 µg m$^{-3}$ for acetylene, 0.1-0.7 µg m$^{-3}$ for propane and benzene, 0.2-0.6 µg m$^{-3}$ for n-butane and 0.3-1.2 µg m$^{-3}$ for ethylene), suggesting a high importance of photochemistry processes and vertical dispersion phenomena in regulating concentration levels. Note that ethane concentration levels were still relatively important during summer (mean concentrations $>1.0$ µg m$^{-3}$) suggesting long-range transport (up to intercontinental pollution transport) was among main parameters governing VOC concentration in summer in continental Europe. On the other hand, anthropogenic VOC monthly concentration levels appear more spatially variable in continental Europe in winter. Indeed, the highest absolute differences between VOC concentrations measured at two stations in winter was of 1.3-2.6 µg m$^{-3}$ for ethane, 0.6-1.6 µg m$^{-3}$ for propane, acetylene and benzene, 0.6-1.4 µg m$^{-3}$ for n-butane, and 1.2-4.1µg m$^{-3}$ for ethylene. These concentration level differences probably highlight spatial variations in anthropogenic source contributions to VOC concentrations observed in continental Europe in winter. Lower concentrations of the selected NMHCs in winter were observed at stations located in southern and western Europe, including Ersa, other French sites and high-altitude





ones (see Figs. 2 and 15). Note that high-altitude sites may have the particularity, compared to the other European sites, of being frequently in free-tropospheric conditions. Additionally, southwestern France and Po valley experienced a wet winter in 2013 and 2014 (see Fig. S3 of the Supplement) that may have had a significant impact on the enrichment in VOC anthropogenic sources of air masses advected to these regions and hence can have participated in the decrease of VOC concentrations

monitored at nearby stations. Then, at stations located in central Europe (i.e. stations located in Switzerland, Germany and Czech Republic - see Fig 2), NMHC concentrations tended to be more elevated in winter compared to southern and western European observations, especially for VOC species potentially mainly explained by long-lived combustion sources, evaporative sources and short-lived anthropogenic sources, which could suggest these stations were under different influences. These findings are consistent with VOC concentration variations as a function of air mass origins observed at Ersa and CF

analyses examined in this study (Sect. 4.2 - Figs. 13 and S2 of the Supplement). Furthermore, precipitations in central Europe were less frequent and/or intense in winters 2013 and 2014 compared to normal values (Fig. S3 of the Supplement) which may have favoured VOC source contribution accumulation and transport and hence can have induced higher VOC concentrations measured at nearby monitoring stations. Additionally, high NMHC concentrations were also observed in northern Europe in winter, especially for VOCs mainly explained by evaporative sources and long-lived combustion sources.

15       To go further, Fig. 16 depicts accumulated concentrations of a selection of 15 VOCs measured at 14 European monitoring stations (including Ersa) in winters 2013 and 2014, in order to investigate dominant drivers for VOC concentration variations in Europe in winter build on what we have learned from Ersa's VOC observations in Sect. 4.2. These selected VOCs are those taken into account in the PMF analysis applied to Ersa VOC measurements (Sect. 2.3.2). Stations located in southwestern France and Po Valley showed relatively stable VOC concentrations in winters 2013 and 2014 (total differences

ranged from -0.1 to 0.4 µg m$^{-3}$). At these sites, concentrations of VOCs potentially explained by long-lived combustion sources have slightly decreased in winter 2014 compared to winter 2013 ones (reduction of 0.1-0.4 µg m$^{-3}$, i.e. of 7-24%), which is consistent with synoptic phenomena (Sect. 3.1) inducing warmer temperatures in winter 2014 compared to normal values (Fig. S3 of the Supplement). However, VOC concentrations and their variations observed at Ersa and in northwestern France were more similar to central European ones than southwestern French and southern European ones. VOC concentrations

measured in central Europe were generally significantly higher in winter 2013 compared to winter 2014 ones (total differences of 2.6-3.2 µg m$^{-3}$), with the exception of stations located in northeastern Germany (i.e. WAL, NGL and ZGT - Fig. 2; total differences of -2.8-0.7 µg m$^{-3}$). As observed at Ersa again, in central Europe, VOC concentrations potentially related to anthropogenic sources that have influenced Ersa VOC observations were higher in winter 2013 compared to winter 2014 ones, especially for those influenced by long-lived combustion sources (reduction of 0.7-1.1 µg m$^{-3}$, corresponding to 21-44% of

winter 2013 concentrations) and short-lived anthropogenic source contributions (reduction of 0.6-1.3 µg m$^{-3}$, i.e. of 21-43%). VOC concentrations related to evaporative sources also decreased but to a lesser extent (reduction of 0.3-0.5 µg m$^{-3}$, i.e. of 13-24%). These findings are consistent with winter variations in anthropogenic source contributions impacting VOC concentrations at Ersa when air masses were advected to the site from central Europe (Fig. 13). They also highlight interannual variations in local contributions to VOC concentrations observed in central Europe in winter. Furthermore, synoptic



phenomena that have occurred in winters 2013 and 2014 as discussed in Sect. 3.1, have impacted meteorological conditions in central Europe, i.e. temperatures were respectively colder and warmer compared to normal values (Fig. S3 of the Supplement). That could partly explain VOC concentration variations in central Europe in these winters. Note that meteorological differences compared to normal values in winters 2013 and 2014 were more marked in central Europe

compared to southern France and southern Europe (Fig. S3 of the Supplement), which could partly explain their respective interannual variations. Then, VOC concentrations monitored in Scandinavia (represented by PAL station observations on Fig. 16) were higher in winter 2014 compared to winter 2013 ones (total difference of 0.7 µg m$^{-3}$), as well as those measured at stations located in northeastern Germany (especially NGL and ZGT stations - differences of 1.7-2.8 µg m$^{-3}$). These increases in winter 2014 in these regions concerned especially VOCs related to long-lived combustion (increases of 0.2-0.5 µg m$^{-3}$, i.e.

9-25%) and evaporative sources (0.3-0.4 µg m$^{-3}$, i.e. 11-18%). Even though these regions experienced a cold winter in 2013 (Fig. S3 of the Supplement), winter 2014 was not as warmer-than-average as in central Europe, since an intense cold wave occurred in January 2014 and was associated with a strong anticyclone centered on western Russia and extending from Finland to Crimea. Additionally, precipitations in these regions were less frequent and/or intense than normal values (Fig. S3 of the Supplement) in winter 2014 compared to winter 2013 which may have favoured accumulation and transport and hence induced

higher VOC concentrations measured at nearby monitoring stations.

In conclusion, the study of NMHC variabilities in continental Europe showed that Ersa can be considered as a good regional representative station. Summer VOC concentration levels did not vary much spatially in Europe suggesting that photochemistry, vertical dispersion phenomena and long-range transport were the main drivers of VOC concentration variations in Europe in summer. Nevertheless, winter concentration levels can significantly vary temporally (at interannual

scale) and spatially (lower concentrations in southern and western Europe than in central and northern Europe), pointing out local influence and spatial variations in anthropogenic source contributions to VOC concentrations observed in continental Europe. Ersa concentration variations in winter were more representative of central Europe than southern/western or northern Europe. These comparisons also revealed that meteorological conditions, especially in winter, can significantly influence anthropogenic source contributions by acting on their emission strengths, accumulation, transport or deposition, and hence

they can affect VOC concentration levels impacting continental Europe.

## 6. Conclusions

The western Mediterranean is known as a sensitive region sorely affected by air pollution, making this region relevant for investigation. This atmospheric pollution is partly owing to the conjunction of intense local anthropogenic emissions, specifically concentrated in coastal cities, natural emissions enhanced by favourable climatic conditions as well as

contributions of more distant sources. This complex mixture of air pollutants will have impacts on human health, ecosystems and climate. However, to clearly assess the various incidences of this complex pollution impacting the Mediterranean region,



supplementary observational data collected in the region were needed, since they remained scarce, especially long-term measurements.

Considering the variability of VOCs at different timescales, it is particularly interesting to carry out measurements over long periods to better understand seasonal and interannual variations in VOC sources impacting the region, especially in the view of the expected progression of climate change with regional warming. As a result, within the framework of the ChArMEx project, a background monitoring station has been set up and operated from June 2012 to June 2014 at a remote background site of Corsica Island (Ersa) in the northwestern part of the Mediterranean. Around 300 atmospheric measurements of a wide range of VOCs (primary anthropogenic and biogenic species as well as oxygenated compounds) were conducted at Ersa with different off-line techniques. This study presents seasonal variabilities of 25 selected VOCs, which showed significant concentration levels during the study period, and their various associated sources.

Particular attention in this study was brought to identifying and evaluating the respective contributions of the various potential sources influencing VOC observations at Ersa. Apportionment factorial analysis was hence conducted on the database composed of 14 primary VOCs (or grouped VOCs) using the positive matrix factorization technique. The objective was to define and examine covariation factors of VOCs that were characteristic of aged or local primary emissions. The selected PMF solution was composed of 5 factors, including a biogenic factor (average relative contribution of 4% to the total concentration of measured VOCs), three anthropogenic factors (short-lived anthropogenic sources, evaporative sources and long-lived combustion sources; cumulated contribution of 57%), determined as originate from either local or more distant emission zones (e.g. Italy and potentially central Europe); and a remaining one (39%) related to regional background pollution (aged air masses advected to the site from a large part of continental Europe).

Five primary biogenic compounds were measured (including isoprene and α-pinene). Biogenic compounds were principally imputed to a local origin and exhibited high concentrations from June to August, consistent with temperature and solar radiation variations. During late spring/summer periods, biogenic source was one of the main sources impacting VOC concentrations observed at the receptor site (16% on average and up to 53%). Biogenic compounds also showed significant interannual variations, related to temperature and solar radiation variations.

16 selected primary anthropogenic species, having atmospheric lifetimes ranging from a few hours to some days and tracers of various sources, were monitored at the receptor site. Anthropogenic VOC concentrations observed at the receptor site were low in spring and summer since anthropogenic sources were largely influenced by chemical processes, vertical dispersion phenomena and regional contributions. The regional background contributed from 30 to 55% to the total measured VOC concentration observed at the Ersa station in these seasons. Short-lived anthropogenic sources also explained from 19 to 30% of the total measured VOC mass possibly contributed by ship transport and relatively close potential emission areas (southeastern French and Italian industrialized and populated coastline areas). Furthermore, all selected anthropogenic compounds were characterized by high concentration levels in fall and winter. During these seasons, the regional background and evaporative sources were identified as the dominant contributors to VOC concentrations monitored at the study site (contributions of 35-45% and of 24-33% to the total VOC concentration, respectively), suggesting that regional contributions





significantly influenced VOC concentrations in fall and winter. Long-lived combustion sources also contributed significantly to VOC concentrations especially in wintertime (explaining 18-24% of the total measured VOC mass), partly due to an increased use of residential heating in the cold season. Anthropogenic sources showed higher accumulated contributions when European air masses were advected to Ersa and could be attributed to potential emission areas located in Italy (the Po Valley

and central Italy) and possibly more distant ones in central Europe (western Hungary, Croatia and Slovenia). To a lesser extent, high anthropogenic source contributions were also noticed when Ersa received air masses originating from France (potential emission areas located in the southeast of France) and Corsica-Sardinia. Interannual variations in anthropogenic VOC concentrations highlighted significant differences between winter periods of 2013 and 2014. VOC concentrations were particularly higher during winter 2013 compared to winter 2014 ones, associated with anthropogenic source contribution

variations. Main parameters governing VOC concentration variations in winters 2013 and 2014 seem to be the emission strength of the main anthropogenic sources identified in this study, continental regional background together with external parameters, i.e. dispersion phenomena (long-range transport, enrichment in anthropogenic sources of continental air masses advected to the site as well as air mass origin occurrences) and pollutant depletion (in relation to the oxidizing capacity of the environment). This study also highlights that meteorological conditions can significantly affect the importance of these

parameters in controlling VOC concentration variations in winter.

Moreover, four oxygenated VOCs (such as acetone) largely prevailed in the VOC speciation during the 2-yr monitoring period. OVOC sources can include primary emissions from both biogenic and anthropogenic origins and OVOCs may also be induced by secondary (biogenic/anthropogenic) sources from oxidation of various VOCs. OVOC concentrations measured at Ersa were generally higher in summer, which could be the result of a high contribution of secondary and biogenic

sources, whereas their concentrations during fall and winter were potentially more influenced by anthropogenic primary/secondary sources, more specifically for MEK.

Concurrent datasets of VOC concentrations from 3 summer campaigns performed at Ersa help to comfort the representativeness of the 2-yr monitoring period in terms of summer concentration levels, variations and source apportionment. The consistency in VOC concentration levels between the 2-yr monitoring period and the three summer ones can suggest that

the annual temporal coverage of VOC measurements realized over the two years of the observation period was sufficiently adapted to well characterize VOC concentration variations (at seasonal scale). Moreover, sources identified in this study and those for the summer 2013 Ersa monitoring period showed globally similar chemical compositions, regarding VOCs in common between the two factorial analyses. These comparisons also pointed out the contribution of the larger time scale of VOC measurements to deconvolve long-lived combustion sources from regional background and to highlight interannual

variations in anthropogenic source contributions. However, they also raised the importance of the consideration of a finer time resolution and higher temporal coverage of VOC measurements conducted at remote background sites to comfort results from source apportionment, in terms of deconvolution of anthropogenic sources, which can show some significant correlations between them, as a consequence of their similar seasonal variations. The consideration in the factorial analysis of diurnal variations could help to limit this potential statistical bias.



Finally, during the same 25-month period as the Ersa monitoring campaign, VOC measurements were conducted at 17 other European monitoring stations, allowing us to examine the representativity of Ersa station in terms of seasonal and interannual variations in VOC concentrations impacting continental Europe and to provide some insights on dominant drivers for VOC concentration variations in Europe built on what we have learned from Ersa's VOC observations. The study of NMHC

variabilities in continental Europe showed that Ersa can be considered as a good regional representative station. Summer VOC concentration levels did not vary much spatially, suggesting that photochemistry processes, vertical dispersion phenomena and long-range transport were the main drivers of VOC concentration variations in continental Europe in summer. Nevertheless, winter VOC concentration levels can significantly vary temporally (at interannual scale) and spatially (lower concentrations in southern and western Europe than in central and northern Europe), pointing out spatial variations in anthropogenic source

contributions to VOC concentrations observed in continental Europe. Ersa concentration variations in winter were more representative of central Europe than southern/western or northern Europe. These comparisons also revealed that meteorological conditions, especially in winter, can significantly influence anthropogenic source contributions by acting on their emission strengths, accumulation, transport or deposition, and hence they can affect VOC concentration levels impacting continental Europe. As a result, these findings point out the interest in conducting multi-site and multi-year measurements to

be sufficiently representative of interannual and spatial variations in regional pollution impacting continental Europe in winter. As a consequence, this finding also point out that shorter observation periods (i.e., up to two months) may reflect the variability of the identified parameters under the specific meteorological conditions of the studied period.

After this study, some questions remain in terms of identification and characterization of OVOC sources and origins at seasonal and interannual scales. Hence, it would be interesting to conduct at Ersa additional long-term VOC measurements,

including OVOCs and tracers of various primary sources, at a finer time resolution and a higher temporal coverage, which would help to complete the understanding of determinants governing OVOC concentration variations initiated both by Michoud et al. (2017) and this study.

**Data availability**

Access to EOP and summer 2013 SOP-1b VOC datasets used for this publication is open to registered users following the data

and publication policy of the ChArMEx program (http://mistrals.sedoo.fr/ChArMEx/Data-Policy/ChArMEx_DataPolicy.pdf, last access: 03/04/2020). VOC datasets from Ersa summer field campaigns of 2012 and 2014 are available upon request. Please contact Stéphane Sauvage (stephane.sauvage@imt-lille-douai.fr) for further information.

**Author contributions**

SS, NL, JS and FD designed the research and were involved in the logistics and the collection of VOC samples on field. TS

calculated uncertainties related to VOC measurements conducted with DNPH cartridges and canisters and validated them





following the ACTRIS protocol. CD, SS, VG and NL analyzed VOC data, conducted and interpreted the VOC PMF analysis and examined dominant factors controlling VOC concentrations. CD wrote the manuscript. All co-authors were involved in data discussion and edited the paper.

**Acknowledgements**

5   This study received financial support from the MISTRALS/ChArMEx programme funded by CNRS/INSU, CEA, and Météo-France, from ADEME, the French Environmental Ministry, the CaPPA projects, and the Communauté Territoriale de Corse (CORSiCA project). The CaPPA project (Chemical and Physical Properties of the Atmosphere) is funded by the French National Research Agency (ANR) through the PIA (Programme d'Investissement d'Avenir) under contract ANR-11-LABX-0005-01 and by the Regional Council Nord-Pas de Calais and the European Funds for Regional Economic Development

10  (FEDER). This research was also funded by the European Union Seventh Framework Programme under grant agreement number 293897, the DEFI-VOC project, CARBO-SOR/Primequal and SAF-MED (ANR grant number ANR-12-BS06-0013-02). The authors are thankful for the $O_3$ and CO datasets made available, respectively, by François Gheusi from Laboratoire d'Aérologie and by ICOS-France monitoring network. The authors also want to thank Thierry Leonardis, Emmanuel Tison, Vincent Gaudion, Laurence Depelchin and Isabelle Fronval for their contributions to obtaining of VOC datasets, from the

15  organization and the setting of instruments at Ersa to the analysis of VOC samples at the laboratory, but also Thierry Bourianne and Qualitair Corse team for their involvement on site concerning VOC sample routines as well as the associated logistic procedure. The authors finally thank the ChArMEx project manager Eric Hamonou for his logistical help.





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



**Table 1: Technical details of the set-up for VOC measurements during the field campaign from June 2012 to June 2014. Air samples were collected bi-weekly (every Monday and Thursday) at Ersa from 09:00-13:00 UTC (from early November 2012 to late December 2012 and from early November 2013 to late June 2014) or 12:00-16:00 UTC (from early June 2012 to late October 2012 and from early January 2013 to late October 2013).**

5   [a] ethane, ethylene, propane, propene, i-butane, n-butane, acetylene, i-pentane and n-pentane, n-hexane, isoprene, benzene, toluene, ethylbenzene, m,p-xylenes, o-xylene

[b] formaldehyde, acetaldehyde, acetone, acrolein, propanal, methyl vinyl ketone (MVK), methacrolein (MACR), methyl ethyl ketone (MEK), i-/n-/butanals, benzaldehyde, glyoxal, methylglyoxal, hexanal

| Instrument | Steel canisters | DNPH cartridges – Chemical desorption (acetonitrile) – HPLC-UV | Solid adsorbent – Adsorption/thermal desorption – GC-FID |
|---|---|---|---|
| Time Resolution (min) | 240 | 240 | 240 |
| Number of samples | 152 | 91 | 63 |
| Detection limit ($\mu g\ m^{-3}$) | 0.01-0.05 | 0.02-0.05 | 0.01 |
| Uncertainties $\frac{U(X)}{X}$ mean [min - max] (%) | 25 [7-43] | 23 [6-41] | 26 [7-65] |
| Species | 21 $C_2$ - $C_5$[a] VOCs | 15 $C_1$ - $C_6$[b] OVOCs | 6 $C_6$ - $C_{11}$ n-aldehydes<br>28 $C_5$ - $C_{16}$ alkanes/alkenes<br>10 $C_6$ - $C_9$ aromatics<br>6 Monoterpenes |
| References | Sauvage et al., 2009 | Detournay, 2011; Detournay et al., 2013 | Ait-Helal et al., 2014; Detournay, 2011; Detournay et al., 2011 |



**Table 2: Input information and mathematical diagnostic for the results of PMF analysis.**

| **Input information** | | |
|---|---|---|
| Samples | N | 152 |
| Species | M | 14 |
| Factors | P | 5 |
| Runs | | 100 |
| Nb. Species indicated as weak | | 0 |
| $F_{peak}$ | | 0.8 |
| **Model quality** | | |
| Q robust | Q(r) | 2589.7 |
| Q true | Q(t) | 2119.9 |
| Maximum individual standard deviation | IM | 0.27 |
| Maximum individual column mean | IS | 1.52 |
| Mean ratio (modelled vs. measured) | Slope(TVOC) | 1.01 |
| $TVOC_{modelled}$ vs. $TVOC_{measured}$ | R²(TVOC) | 0.99 |
| Nb. of species with $R^2 > 0.6$ | | 10 |
| Nb. of species with $1.1 > slope > 0.6$ | | 9 |



**Table 3: Back-trajectory clusters for air masses observed at Ersa from June 2012 to June 2014. The transit time (expressed in h) corresponds to the time spent since the last anthropogenic contamination, i.e. since air masses left continental coasts.**

| Clusters | Source regions (wind sectors) | Transit time (h) Median [min-max] | Occurrence (%) |
|---|---|---|---|
| **C1** | **Marine** | **48 [18-48]** | **15** |
| | **Marine (SW)** | | |
| | *Short trajectories* | 48 [39-48] | 7 |
| | *Long trajectories* | 40 [18-48] | 5 |
| | **Marine (SE)** | | |
| | *Long trajectories* | 42 [25-48] | 3 |
| **C2** | **Corsica-Sardinia (S)** | **0 [0-38]** | **14** |
| | *Short trajectories* | 2 [0-38] | 9 |
| | *Long trajectories* | 0 [0-15] | 5 |
| **C3** | **Europe (NE-E)** | **6 [2-44]** | **31** |
| | *Short trajectories* | 23 [4-44] | 11 |
| | *Long trajectories* | 6 [2-16] | 20 |
| **C4** | **France (NW-N)** | **8 [3-48]** | **26** |
| | *Short trajectories* | 19 [10-48] | 6 |
| | *Long trajectories* | 8 [3-19] | 20 |
| **C5** | **Spain (W)** | | |
| | *Long trajectories* | 36 [20-45] | 5 |





**Table 4: Statistics (µg.m⁻³), standard deviations (σ - µg m⁻³), detection limits (DL - µg m⁻³) and relative uncertainties U(X)/X (Unc. - %) of selected VOC concentrations measured at the site from June 2012 to June 2014.**

|  | Species | Min | 25 % | 50 % | Mean | 75 % | Max | σ | DL | Unc. |
|---|---|---|---|---|---|---|---|---|---|---|
| **ALKANES** | **Ethane** | 0.57 | 1.13 | 1.85 | 1.86 | 2.46 | 4.28 | 0.81 | 0.01 | 7 |
|  | **Propane** | 0.18 | 0.44 | 0.77 | 0.94 | 1.41 | 2.60 | 0.61 | 0.02 | 11 |
|  | **i-Butane** | 0.01 | 0.09 | 0.17 | 0.24 | 0.35 | 1.02 | 0.19 | 0.02 | 22 |
|  | **n-Butane** | 0.05 | 0.16 | 0.26 | 0.37 | 0.57 | 1.09 | 0.26 | 0.02 | 13 |
|  | **i-Pentane** | 0.06 | 0.15 | 0.22 | 0.25 | 0.31 | 0.90 | 0.14 | 0.03 | 25 |
|  | **n-Pentane** | 0.02 | 0.09 | 0.18 | 0.20 | 0.27 | 0.80 | 0.13 | 0.03 | 33 |
|  | **n-Hexane** | 0.02 | 0.04 | 0.07 | 0.08 | 0.10 | 0.27 | 0.05 | 0.04 | 43 |
| **ALKENES** | **Ethylene** | 0.09 | 0.19 | 0.28 | 0.32 | 0.39 | 0.87 | 0.17 | 0.01 | 14 |
|  | **Propene** | 0.01 | 0.04 | 0.06 | 0.07 | 0.09 | 0.17 | 0.03 | 0.02 | 40 |
| **ALKYNE** | **Acetylene** | 0.03 | 0.09 | 0.18 | 0.26 | 0.36 | 1.23 | 0.23 | 0.01 | 12 |
| **DIENE** | **Isoprene** | 0.01 | 0.01 | 0.04 | 0.16 | 0.16 | 2.28 | 0.31 | 0.03 | 32 |
| **TERPENES** | **α-Pinene** | <0.01 | 0.02 | 0.07 | 0.29 | 0.23 | 3.61 | 0.54 | 0.01 | 40 |
|  | **Camphene** | <0.01 | <0.01 | <0.01 | 0.02 | 0.01 | 0.40 | 0.06 | 0.01 | 73 |
|  | **α-Terpinene** | <0.01 | <0.01 | <0.01 | 0.05 | 0.04 | 0.88 | 0.14 | 0.01 | 47 |
|  | **Limonene** | <0.01 | <0.01 | 0.02 | 0.17 | 0.29 | 1.73 | 0.27 | 0.01 | 45 |
| **AROMATICS** | **Benzene** | 0.07 | 0.16 | 0.26 | 0.31 | 0.39 | 1.11 | 0.19 | 0.03 | 25 |
|  | **Toluene** | 0.04 | 0.15 | 0.23 | 0.28 | 0.34 | 0.84 | 0.17 | 0.04 | 26 |
|  | **Ethylbenzene** | 0.02 | 0.02 | 0.02 | 0.04 | 0.05 | 0.15 | 0.03 | 0.04 | 50 |
|  | **m,p-Xylenes** | 0.02 | 0.07 | 0.10 | 0.12 | 0.14 | 0.41 | 0.08 | 0.04 | 45 |
|  | **o-Xylene** | 0.02 | 0.02 | 0.06 | 0.07 | 0.10 | 0.32 | 0.06 | 0.04 | 44 |
| **CARBONYL COMPOUNDS** | **Formaldehyde** | 0.28 | 0.68 | 1.17 | 1.53 | 1.89 | 6.30 | 1.24 | 0.03 | 7 |
|  | **Acetaldehyde** | 0.40 | 0.67 | 0.83 | 0.96 | 1.23 | 2.87 | 0.41 | 0.03 | 22 |
|  | **Acetone** | 1.50 | 2.46 | 3.57 | 4.31 | 4.98 | 16.49 | 2.64 | 0.03 | 6 |
|  | **MEK** | 0.18 | 0.27 | 0.33 | 0.36 | 0.45 | 0.90 | 0.14 | 0.03 | 10 |





**Figure 1: Maps of the Mediterranean region and Corsica (source Google earth) and view of the sampling station. (a) Position of Corsican island in the Mediterranean region. (b) The sampling site and major Corsican agglomerations are displayed as a blue star and yellow diamonds, respectively. (c) Picture of the sampling site, during the observation period. Maps provided by Google Earth Pro software (v.7.3.3; image Landsat/Copernicus; data SIO, NOOA, U.S, Navy, NGA, GEBCO; © Google Earth).**





**Figure 2: Locations of 18 European monitoring stations that included VOC measurements conducted from June 2012 to June 2014. These stations are part of EMEP/GAW networks. They are characterized by their GAW ID and the figures in parentheses correspond to their altitudes which are given in reference to standard sea level. AUC, BIR, ERS, KOS, NGL, PYE, RIG, SMR, SSL and TAD are categorized as GAW regional stations for Europe. CMN, HPB, JFJ and PAL are categorized as GAW global stations. AHRL, SMU, WAL and ZGT are considered as GAW other elements stations in Europe, more precisely, ZGT is a coastal station while HRL, SMU and WAL are rural stations. Note that high-altitudes stations as CMN and HPB could be frequently in free-tropospheric conditions. More information on these stations can be found on EMEP (https://www.nilu.no/projects/ccc/sitedescriptions/index.html, last access: 03/04/2020) or GAW station information system (https://gawsis.meteoswiss.ch/GAWSIS//index.html#/, last access: 03/04/2020) sites. Ersa site is underlined in red. Square markers indicate that VOCs were collected by steel canisters and analyzed thereafter at laboratories (i.e. off-line measurements). Triangle and diamond markers indicate that VOC measurements were conducted in-situ using PTR-MS or GC systems, respectively. NMap**





provided by Google Earth Pro software (v.7.3.3 image Landsat/Copernicus – IBCAO; data SIO, NOOA, U.S, Navy, NGA, GEBCO; © Google Earth).



**Figure 3: Classification of air masses which impacted the Ersa site during the observation field campaign as a function of their trajectory. Back trajectories simulated with the HYSPLIT model (NOAA-ARL) were classified into five clusters: Marine (cluster 1 – wind sectors SW & SE), Corsica-Sardinia (cluster 2 – S), Europe (cluster 3 – NE-E), France (cluster 4 – NW-N) and Spain (cluster 5 – W). These five clusters were illustrated by example maps with five-trajectories (interval of 1h between each, time of arrival indicated by different colors of trajectory, receptor site represented by a black star) for five single days representative of an isolated cluster. To complete, areas covered by back-trajectories of each cluster are also indicated. Maps provided by Google Earth Pro software (v.7.3.3; image Landsat/Copernicus; data SIO, NOOA, U.S, Navy, NGA, GEBCO; © Google Earth).**





**Figure 4: (a) Monthly variations in meteorological parameters (temperature expressed in °C, global solar radiation in W m⁻², relative humidity in % and wind speed in m s⁻¹) represented by box plots and (b) their average values as a function of the year. Blue solid line represents the median value, the red marker represents the mean value and the box shows the interquartile range. The bottom and the top of box depict the first and the third quartiles (i.e. Q1 and Q3). The ends of the whiskers correspond to first and the ninth deciles (i.e. D1 and D9). Note that, meteorological parameter data used in this study were restricted to periods when VOC measurements were realized.**



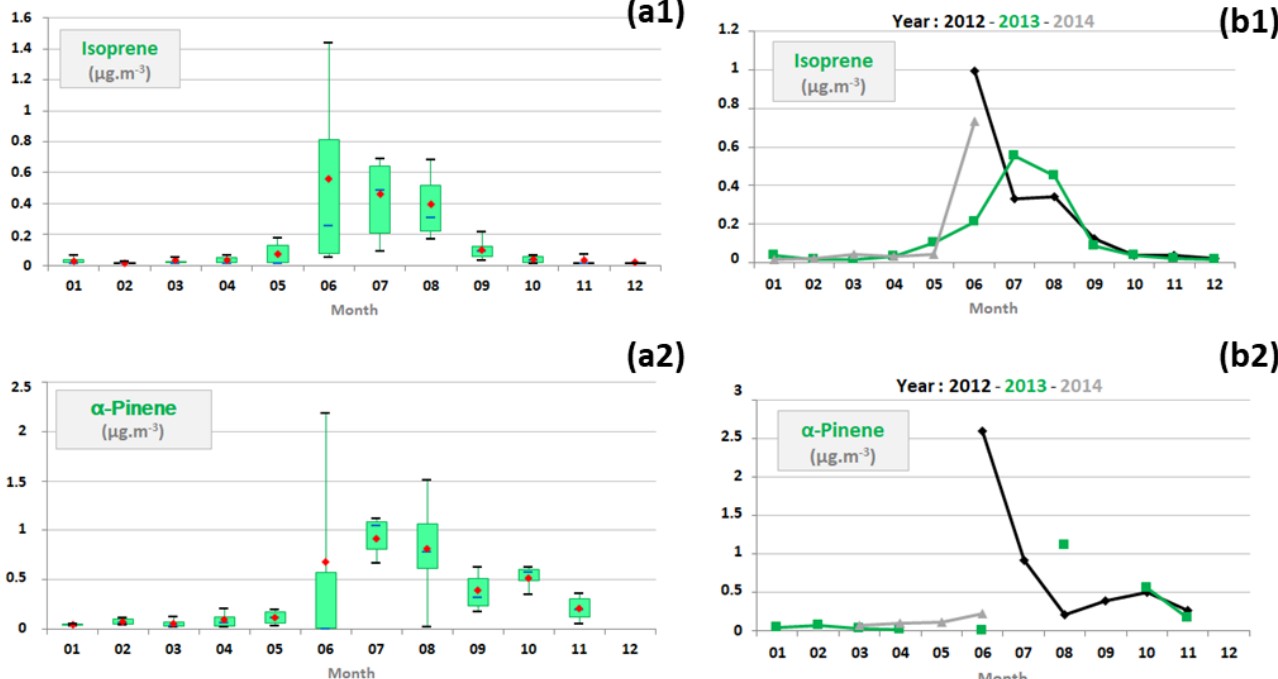

**Figure 5: (a) Monthly variations in a selection of biogenic VOC concentrations (expressed in µg m⁻³) represented by box plots and (b) their average monthly concentrations as a function of the year. Blue solid line represents the median value, the red marker represents the mean value and the box shows the interquartile range. The bottom and the top of box depict the first and the third quartiles (i.e. Q1 and Q3). The ends of the whiskers correspond to first and the ninth deciles (i.e. D1 and D9).**







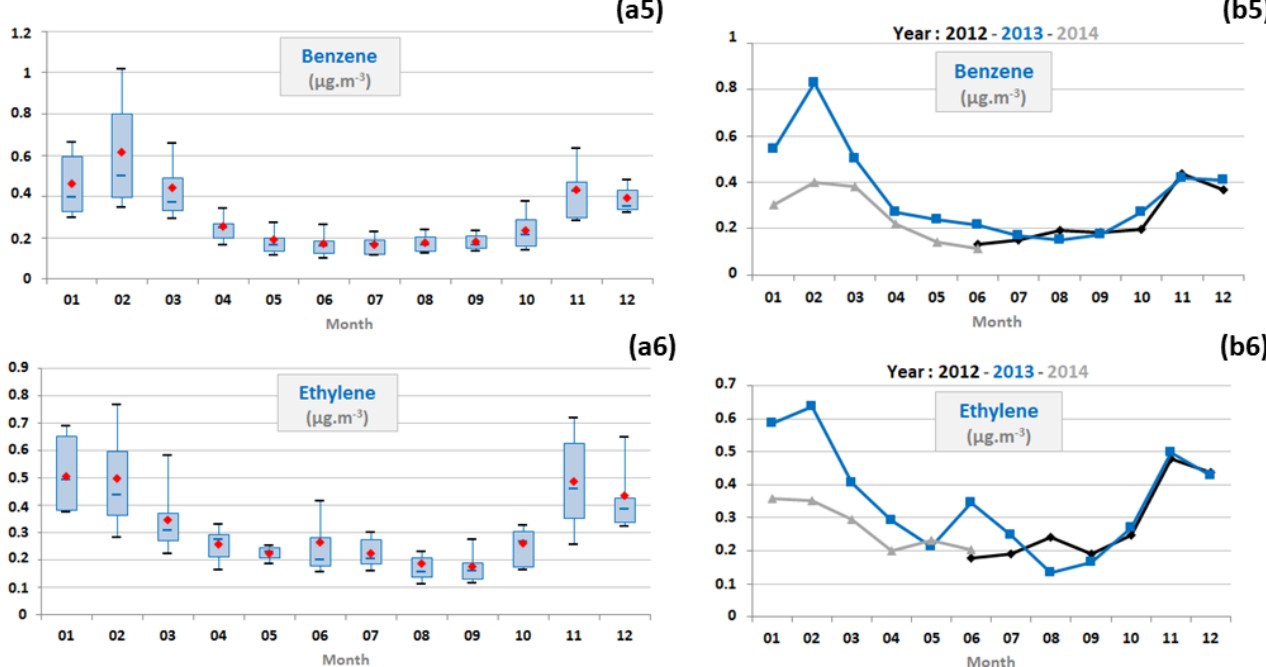

Figure 6: (a) Monthly variations in a selection of anthropogenic VOC concentrations (expressed in µg m⁻³) represented by box plots and (b) their average monthly concentrations as a function of the year. Blue solid line represents the median value, the red marker represents the mean value and the box shows the interquartile range. The bottom and the top of box depict the first and the third quartiles (i.e. Q1 and Q3). The ends of the whiskers correspond to first and the ninth deciles (i.e. D1 and D9).





**Figure 7: (a) Monthly variations in a selection of oxygenated VOC concentrations (expressed in µg m⁻³) represented by box plots and (b) their average monthly concentrations as a function of the year. Blue solid line represents the median value, the red marker represents the mean value and the box shows the interquartile range. The bottom and the top of box depict the first and the third quartiles (i.e. Q1 and Q3). The ends of the whiskers correspond to first and the ninth deciles (i.e. D1 and D9).**

**Figure 8: Chemical profiles of the 5-factor PMF solution (14 VOCs). Factor contributions to each species (µg m$^{-3}$) and the percent of each species apportioned to the factor are displayed as a grey bar and a color circle, respectively. Factor 1 - biogenic source; factor 2 - short-lived anthropogenic sources; factor 3 – evaporative sources; factor 4 – long-lived combustion sources; factor 5 – regional background.**





**Figure 9: (a) Time series of VOC factor contributions (µg m⁻³) and (b) accumulated relative VOC contributions. Factor 1 - biogenic source; factor 2 - short-lived anthropogenic sources; factor 3 – evaporative sources; factor 4 – long-lived combustion sources; factor 5 – regional background.**








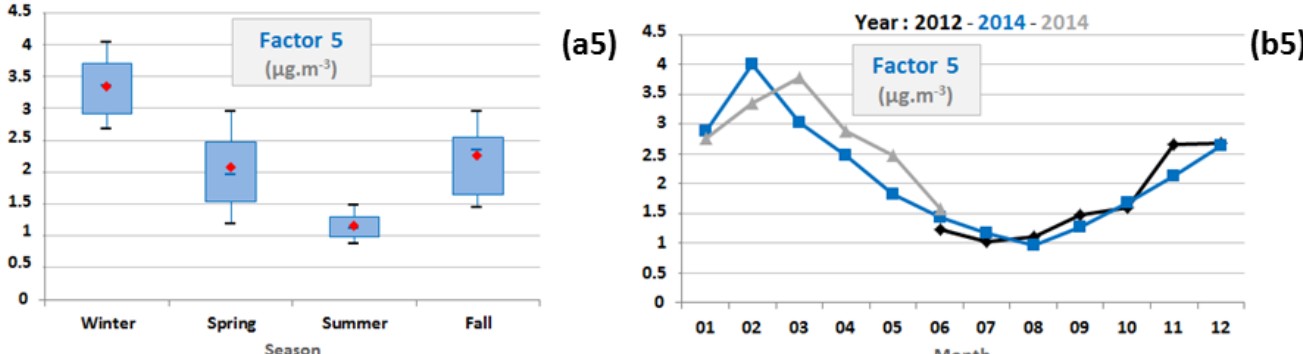

**Figure 10: (a) Seasonal variations in VOC factor contributions (expressed in µg m$^{-3}$) represented by box plots and (b) their average contributions as a function of the year. VOC factors: factor 1 - biogenic source; factor 2 - short-lived anthropogenic sources; factor 3 – evaporative sources; factor 4 – long-lived combustion sources; factor 5 – regional background. Winter: 01/01-31/03 periods – spring: 01/04-30/06 periods – summer: 01/07-30/09 periods – fall: 01/10-31/12 periods. Blue solid line represents the median value, the red marker represents the mean value and the box shows the interquartile range. The bottom and the top of box depict the first and the third quartiles (i.e. Q1 and Q3). The ends of the whiskers correspond to first and the ninth deciles (i.e. D1 and D9).**





**Figure 11: VOC factor contributions (µg m⁻³) as a function of air mass origins. VOC factors: factor 2 - short-lived anthropogenic sources; factor 3 – evaporative sources; factor 4 – long-lived combustion sources; factor 5 – regional background. Air masses originating from France and Europe are subdivided into short and long trajectories to highlight local and more distant contributions (see Sect. 3.2).**



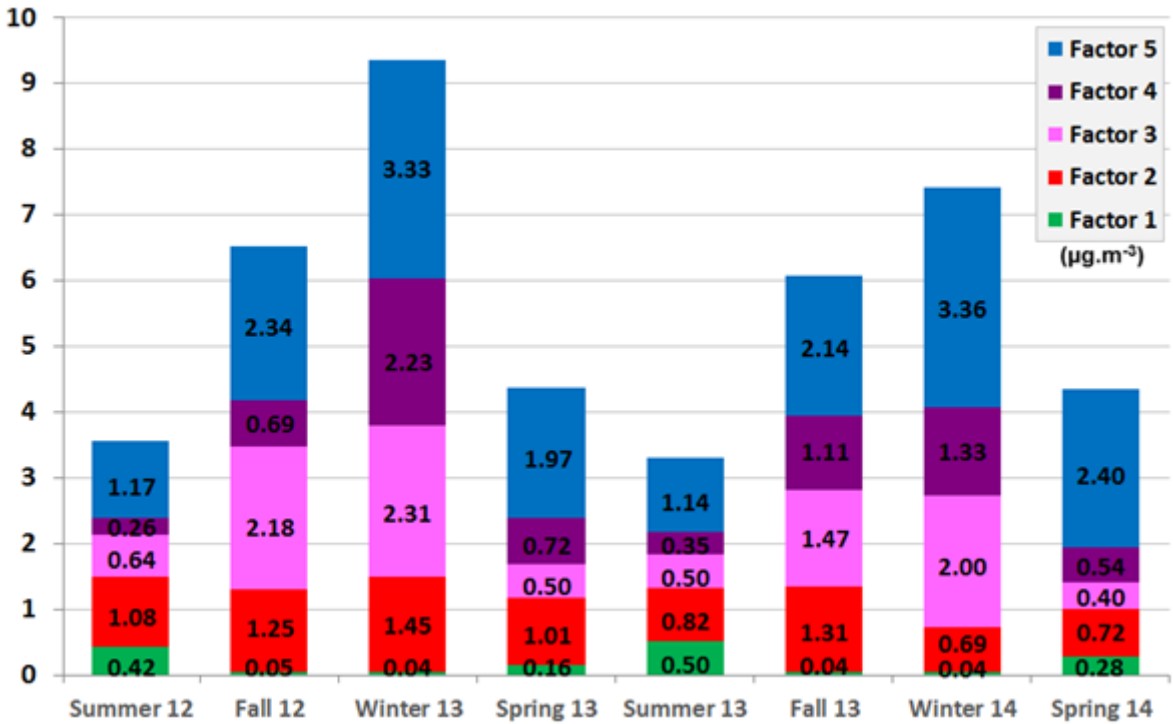

**Figure 12: Variations of seasonal averaged accumulated contributions of the five modeled VOC sources (expressed in μg m⁻³). Factor 1 - biogenic source; factor 2 - short-lived anthropogenic sources; factor 3 – evaporative sources; factor 4 – long-lived combustion sources; factor 5 – regional background. Winter: 01/01-31/03 periods – spring: 01/04-30/06 periods – summer: 01/07-30/09 periods – fall: 01/10-31/12 periods.**

**Figure 13: Accumulated average contributions (expressed in µg m⁻³) of the anthropogenic VOC factors per season as a function of air mass origins.  Factor 2 - short-lived anthropogenic sources; factor 3 – evaporative sources; factor 4 – long-lived combustion sources; factor 5 – regional background. Winter: 01/01-31/03 periods – fall: 01/10-31/12 periods.**





**Figure 14: Potential source areas contributions to factors 2-5 during winters 20013 and 2014. Contributions are expressed in µg m⁻³. Factor 2 - short-lived anthropogenic sources; factor 3 – evaporative sources; factor 4 – long-lived combustion sources; factor 5 – regional background. Winter: 01/01-31/03 periods – fall: 01/10-31/12 periods.**







**Figure 15: Monthly concentration time series of a selection of NMHCs (expressed in µg m⁻³) measured at 18 European monitoring stations (see Sect. 5)**


**Figure 16: Accumulated average concentrations of a selection of 15 VOCs (expressed in µg m$^{-3}$) measured at 14 European monitoring stations (see Sect. 5) in winters 2013 and 2014. Selected VOCs in this study are those taken into account in the factorial analysis applied to Ersa two-year VOC dataset. Note that for some stations, accumulated concentrations are incomplete since only VOC measured at a station at both winter periods were taking into in this comparison analysis, at the exception of TAD and RIG stations. For these latter, VOC measurements did not cover winter 2014 period, that's why accumulated concentrations were only indicated**



for winter 2013 period. For AUC, HRL, BIR and SMR stations, represented by grey markers, VOC measurements were not realized both during winters 2013 and 2014, or were conducted with a PTR-MS and hence we considered accumulated concentrations only of aromatic compounds are not sufficiently representative for interannual VOC concentration variations. Ersa location and results are highlighted. Map provided by Google Earth Pro software (v.7.3.3 image Landsat/Copernicus – IBCAO; data SIO, NOOA, U.S, Navy, NGA, GEBCO; © Google Earth).