# Peer review of "Seasonal variation and origins of volatile organic compounds observed during two years at a western Mediterranean remote background site (Ersa, Cape Corsica)"

_Atmospheric Chemistry and Physics, 2020_

## Referee Comment (RC1) · Anonymous Referee #1 · 30 Jul 2020

This manuscript presents an analysis of the VOC concentrations and temporal trends over a 2 years long observation campaign at a remote site located in Corsica and representative of the northwest part of the Mediterranean background atmosphere. The monthly, seasonal and interannual variabilities of 21 NMHCs and 4 OVOCs are reported. Source apportionment using positive matrix factorization in combination with back trajectories analysis was carried out on a selection of 14 NMHC species. Not surprisingly, the five factors solution chosen fails to apportion the selected VOCs into their specific emission sources, as the air masses that reach the remote site are already

mixed and processed. Finally, the NMHC temporal concentration trends are compared to those observed in the 17 other European background stations at the same period. Overall, while a significant spatial variability in concentration levels is found, especially in winter, similar seasonal trends are observed.

The work provides valuable scientific information as long term VOCs datasets in background sites are still rather scarce in the Mediterranean region. To my point of view this topic together with the comparison with the 17 other background sites located in Europe is the most interesting feature of the manuscript. On the other hand, I have some concerns regarding the added value brought by the PMF analysis on a limited set of VOCs and the relevance of the solution as meteorology (boundary layer, air mass circulation, temperature) seems to be the main factor driving the temporal trends in a remote background site. The authors should more clearly elaborate on the limits of the PMF with respect to the limited set of VOCs and samples in such a remote site. As a general comment on language and structure, I find the manuscript in its current state unnecessarily long. Topics are repeated saying pretty much the same thing in different sections. For example, many points raised in section 4.1 (Determination of controlling factors) have already been discussed in the section 3.5 (VOC factor analysis) and points raised in section 4.2 (The particular case of winter) have partly been discussed in section 4.1 when fall-winter interannual trends are discussed. Same remark for sections 3.3 and 3.5. Some sections lack clarity (see specific comments). Overall, I find it very difficult to extract the main messages of the sections/paragraphs.

A major revision according to these mentioned general comments is consequently required before publication to ACP.

Specific comments • The introduction could be shortened and focused on the scientific context and goals of the study. For example, information given on the various national and international programs is not essential here (Page 3, line 8-13 ; Page 3, line 32 to Page 4, line 4) • It appears that the biweekly samples were collected between 09h00-13h00 UTC for 7 months and between 12h00-16h00 UTC for 15 months.

The authors should comment the sampling strategy and tell whether this sampling time shift could impact the PMF analysis and the interannual variations, especially for species with strong diurnal variations such as biogenic VOCs. • Page 6, o It is not clear when these additional 150 off-line air samples were taken; over the 2-year period? at which frequency? Also some of the VOCs listed (6 C6 - C11 n-aldehydes) are not discussed at all afterwards, why ? o line 8 "44 C5-C16" Check consistency with table 1 where 50 VOC are listed to be sampled with the solid adsorbent. • Page 7, section 2.2.3 ("Additional high frequency VOC measurements performed at Ersa") and 3.4.4 : the information provided in this section is already given in Section S3 ("comparison of VOC measurements with other ones performed at Ersa"). I suggest removing this section from the main text and to merge it with S3 • Page 7, section 2.2.4 ("Concurrent VOC measurements performed at other..") I would suggest shortening this section merge it with section 5. Both sections start with the same 4 lines. • Page 8, line 6 : could the authors elaborate on the choice of the VOCs included in the PMF ? This is a rather limited range of VOCs, compared to other studies (see for example Abeleira et al., 2017, 46 VOCs; Yuan et al., 2012, 73 VOCs). Can the authors provide a rough estimate of their contribution to the total VOCs mass concentration? • Page 8, lines 4-11 are duplicates of lines 22-29- from the Section S1 • Page 8 line 16-20 It is said that the PMF model results reconstructs on average 99% of the total concentration of the 14 selected species, but in the meantime 5 out of the 14 selected are not properly captured by the PMF solution. Also, ethane and propane account for 50% of the VOC mass. In these given circumstances, is the percentage of total reconstructed mass relevant to assess the quality of the PMF solution? It would be helpful to include more information on the PMF preprocessing, and on the diagnostic plots (Q/Q(exp) values vs number of factors, scatter plots of the measured vs reconstructed concentrations, scaled residuals, if and why outliers were removed from the time series, etc..) • I understand that species not properly reconstructed by the PMF model should be categorized as "weak" in the model. This is not the case here, as seen in Table 2, where none of the 14 species are indicated as "weak". Could the

author justify this choice? • Page 8, line 19 The authors specify that "PMF model limitations to explain these species should be kept in mind when examining PMF results". A more detailed discussion would be helpful for the reader to appreciate the limits of the proposed PMF solution. A rough estimate based on figure 8 indicates that these species make up approximatively 80% of the concentration of Factor 2 (short lived species). • Page 9, starting to line 32 to page 10, line 3 : It is unclear what you mean here : "tests that revealed that results only highlighted local contributions . . ." is it related to exploratory tests with empirical weighting function ? Was it finally decided to apply such a weighting function? • Page 9, line 28 Can the author explain the meaning of shortening the back-trajectories? • Page 9, line 24-25 Please rephrase this sentence as we understand that longer 3-day back trajectories were considered in order to be in the same conditions as Michoud et al. • Page 11, section 3.2 Air mass origins and table 3: it is not clear why trajectories categorized as long have median transit time always shorter than the trajectories categorized as short. • Page 14 section 3.4.3 Oxygenated VOCs: this section is hard to follow, line 14-26 are only general considerations with no direct link to the observations. Why not starting with the trends observed (end of line 26, "Formaldehyde and acetaldehyde concentrations. . ." and use some of the general information to support the discussion. Same comment for acetone and MEK • Page 17, 3.5.1 Biogenic source: "Local biogenic source" instead ? • Page 17, line 6 "troposphere is" instead of "troposphere was"? • Page 17, line 1: "to the sum of measured VOC concentrations" : do you include OVOC in this calculation ? Anyway, because the list of the VOCs included in the PMF is not exhaustive, the average individual contribution of each factors to the sum of the measured VOCs should be considered with care. • Line 20, The term "regional" is rather vague, can you indicate which geographical areas are included? • Page 20 "Towards the best experimental strategy to characterize variation in VOC concentrations" Larges parts of this section are copy/paste of the section S4 of the SI (page 20, line 28 to page 21, line 2 ; similar to page 10 of the supporting information, lines 1-6 ; page 21, lines 8-19 similar to page 12 of the SI, lines 22-34) • Page 21, line 25 "are" instead

of "were"  c Page 21 line 25 starting at "Note that.." to line 28 : I suggest to remove this information, it is not essential and distracts the reader from the topic of the section  c Page 22, line 6-8 The authors attribute the high contribution of factor 2 (related to short lived species) in spring and summer to relatively nearby sources. Have the authors checked if the correlation with CO was improved in these specific conditions?  c Page 24, line 24 I don't understand the meaning of the last sentence. Please rephrase "As a consequence, this finding . . . may be reflected. . ."  c Page 25 line 18 Please rephrase "were also be taken.."

Literature cited Abeleira, A., Pollack, I. B., Sive, B., Zhou, Y., Fischer, E. V., and Farmer, D. K.: Source characterization of volatile organic compounds in the Colorado Northern Front Range Metropolitan Area during spring and summer 2015, J. Geophys. Res.-Atmos., 122, 3595–3613, https://doi.org/10.1002/2016jd026227, 2017. Yuan, B., Shao, M., de Gouw, J., Parrish, D. D., Lu, S., Wang, M., Zeng, L., Zhang, Q., Song, Y., Zhang, J., and Hu, M.: Volatile organic compounds (VOCs) in urban air: How chemistry affects the interpretation of positive matrix factorization (PMF) analysis, J. Geophys. Res.-Atmos., 117, 117, D24302, https://doi.org/10.1029/2012jd018236, 2012

---

## Referee Comment (RC2) · Anonymous Referee #2 · 6 Aug 2020

The manuscript presents observations of VOCs at the Ersa site in Cape Corsica over a two-year period and provides a comprehensive description and analysis of their behaviour during this time. PMF analysis of the data is presented along with comparison to other station across Europe. The plots are clear and generally well presented. The length of the manuscript, however, is something of a problem with lots of repetition throughout. I am sure there is an interesting story here, but it is difficult to assess what that is from the current article. Careful consideration of each section, it's findings and their relevance is required in order to make the manuscript worthy of publication. I

therefore recommend a major revision of the manuscript.

I do however, urge the authors to continue to work on this as I feel it can be a very nice piece. The authors should consider writing the manuscript in terms of the features observed at the site and then use the data and plots to explain that behaviour. In its current format, each compound or group is considered separately and methodically (which leads to a comprehensive, but repetitive narrative) whereas, the behaviour of these different compounds can often be explained by the same phenomena (e.g. a changing boundary layer or temperature difference).

I have included suggested changes to individual sections, but these may not be relevant to the newly written article.

Please also note the supplement to this comment:
https://acp.copernicus.org/preprints/acp-2020-607/acp-2020-607-RC2-supplement.pdf

**Supplement:**

**Reviewer comments**

**Overall:**

The manuscript presents observations of VOCs at the Ersa site in Cape Corsica over a two-year period and provides a comprehensive description and analysis of their behaviour during this time. PMF analysis of the data is presented along with comparison to other station across Europe. The plots are clear and generally well presented. The length of the manuscript, however, is something of a problem with lots of repetition throughout. I am sure there is an interesting story here, but it is difficult to assess what that is from the current article. Careful consideration of each section, it's findings and their relevance is required in order to make the manuscript worthy of publication. I therefore recommend a major revision of the manuscript. I do however, urge the authors to continue to work on this as I feel it can be a very nice piece.

The authors should consider writing the manuscript in terms of the features observed at the site and then use the data and plots to explain that behaviour. In its current format, each compound or group is considered separately and methodically (which leads to a comprehensive, but repetitive narrative) whereas, the behaviour of these different compounds can often be explained by the same phenomena (e.g. a changing boundary layer or temperature difference).

I have included suggested changes to individual sections, but these may not be relevant to the newly written article.

**Abstract**
**Page1, Line17:**
"… The VOC speciation was largely dominated by oxygenated VOCs …"
Should this be the VOC abundance or mass? I'm not sure how speciation can be dominated.

**P1, L18:**
"VOC temporal variations are then examined…"
Past tense, should be "were examined"

**P1, L19:**
"… and solar radiation ones."
Delete "ones"

**P1, L20:**
"Anthropogenic compounds have shown an increasing concentration trend in winter (JFM months) followed…"
This reads as though the concentrations increase between these months – ie March is bigger than February which is bigger than January – this doesn't appear to be the case from figure 6
"Anthropogenic compounds showed increased concentrations in winter (JFM months) followed…"

**P1, L21:**
"… and different concentration levels in winter periods of 2013 and 2014."
These are inevitably different, but the question is by how much are they different?
Suggest including "by up to *__XX%__* in the case of *__compoundY__*"

"OVOC concentrations were generally higher in summertime, mainly due to secondary and biogenic sources, whereas their concentrations during fall and winter were potentially more influenced by anthropogenic primary/secondary sources."
This sentence seems a little confusing to me. I agree that the secondary sources of OVOCs will be increased during summertime and that the contribution from biogenic sources will also be greater during summer. As it is written though, it sounds like the anthropogenic secondary sources only contribute to the OVOC concentrations during the winter months. This is not the case, the anthropogenic secondary sources will also increase during the summertime.

P1, L26:
When listing the PMF factors, I suggest that these be listed in order of significance in terms of relative contribution.

P1, L30:
 at the receptor site are also
Suggest changing "receptor site" to ERSA station or observatory

P2, L2:
"… winter 2014 ones could …"
Delete "ones"

**2.2.1. VOC measurements**

Where there any compounds measured by the multiple measurement techniques used in the study? If so, were there any comparison exercises performed to ensure consistency?

P10, L14:
"… and the lowest ones in winter …"
Delete "ones"

P10, L22:
Paragraph beginning "On one hand, western European …" and ending "…across the north Atlantic toward the British Isles (Kendon and McCarthy, 2015)." on P11, L7 seems excessively long. Of key relevance here (to VOC observations) is that the temperatures were different (lower in 2013) and a short statement/sentence to say the lower temperatures were observed across Europe, with relevant citations, would suffice.

P11, L15:
"Relative humidity globally followed opposite seasonal variations than temperature and solar radiation ones."
Should read:
"Globally, relative humidity followed opposite seasonal variation to temperature and solar radiation."

P11, L15:
"Relative humidity globally followed opposite seasonal variations than temperature and solar radiation ones."

P11, L15:
"In June 2012, air was dryer compared to June 2013 and 2014 mean relative humidity values …"
Delete:
"mean relative humidity values …" they're not need ed here since these are described in the parentheses.

P11, L15:
"The wind speed did not show a clear seasonal variation over the two years studied, except maybe higher wind speeds in April and May that could induce higher dispersion of air pollutants and could advect air pollutants from more distant sources to the receptor site."

Suggest changing to ""The wind speed did not show a clear seasonal variation over the two years studied. Slightly higher wind speeds in April and May 2014 which could induce higher dispersion of air pollutants and advect air pollutants from more distant sources to the receptor site."

P11, L19:
"May 2014 encountered particularly windy conditions."
I don't think this sentence is warranted (only 1.5 m/s higher than April) and would suggest removing it this sentence, it is not needed

P11, L31:
Air masses spending longer periods over the ocean will indeed have undergone more atmospheric processing, but they may also have more influence from oceanic sources of VOCs. While these are likely insignificant compared to the anthropogenic inputs from Continental Europe, I feel they should be mentioned here.

P12, L4:
"showed relatively close transport times"
Change to "… short transport times …"

**3.3 VOC mixing ratios**

P12, L8:
The statement "Descriptive statistical results for a selection of 25 VOCs, which showed significant concentration levels during the 2-yr studied period, are summarized in Table 4" implies that more VOCs were measured during the period, but are not reported here because they were below some threshold value decided upon by the authors. If this is the case, there should be a statement describing the selection criteria used to define the "significant concentration levels".

P12, L24:
"On the contrary, larger α-terpinene contribution was noticed during the summer intensive campaign than the 2-yr observation period."
Were these observations made using the same technique or could there be some instrument bias associated with this result? It is important to clarify and state that in the text here.

P12, L28:
"… dispersion, dilution processes …"
are these the same thing?

**Section 3.4.1. Biogenic VOCs**

P13, L2 - 18:
The authors state "Surprisingly, isoprene and α-pinene concentrations were drastically lower in July 2012…" and then go on to state that the temperature and solar radiation during July were lower, therefore, I fail to see the surprise here.

The bigger surprise here seems to be that the July 2013 isoprene falls below the July 2012 level despite the temperature and solar radiation being higher (increasing emissions) and the wind speed being lower

(increasing dispersion) during that period.  Perhaps including the wind direction in figure 4 may help to explain this?

P13, L10:
"… which may be related to the fact that temperature and solar radiation were more favourable to enhance biogenic emissions in June 2012 compared to June 2013 and 2014 meteorological conditions …"
There is also the effect of relative humidity to consider here.  Figure 4 shows the relative humidity was lower in June 2012 and 2014 compared to 2013, see the work of Ferraci et al. for the effect of drought conditions on the emissions of isoprene.  Links to this research would be useful here.

P13, L16:
"This finding could be the result of a weaker degradation of α-pinene due to lower ozone concentrations observed from October to December compared to summer …"

Emissions of isoprene are light and temperature dependant while monoterpenes are thought to be solely temperature dependant.  I'd suggest that the difference in seasonal cycles of isoprene and alpha-pinene is due to the difference between the solar radiation and temperature profiles:  solar radiation falls much quicker than temperature which may have the effect of "switching off" the isoprene emissions before the alpha-pinene emissions.

3.4.2 Anthropogenic VOCs

P13, L16:
"… characterized by almost the same seasonal variation …"
Replace with "similar":  "… characterized by similar seasonal variation …"

P13, L27:
"… with the highest atmospheric lifetime …"
Replace with "Longest":  "… with the longest atmospheric lifetime …"

P13, L27:
"… considering its low photochemical reaction rate with OH radicals …"
"… due to its low photochemical reaction rate with OH radicals …"

P13, L28:
"… It is typically emitted by natural gas use and can be also considered as a tracer of the most distant sources."
Transport and storage of natural gas are also important sources here.

P14, L2:
"… four to ten times higher than ethane one (Atkinson, 1990; Atkinson and Arey, 2003)"
Remove "one", it's not required here.

P14, L4:
"… e.g., Leuncher et al., 2015)."
Should be "Leuchner"

P14, L6:
"… winter 2014 ones."
Remove "ones", it's not required here.

As a result, winter variations of concentration levels concerned at a time close sources and more distant ones and will be more investigated thereafter (Sect. 4.2)."
I don't think this sentence makes sense. Just a statement that the winter period will be investigated later in the manuscript would suffice.

**3.4.3 Oxygenated VOCs**

"Nevertheless, acetaldehyde is only produced as a second or higher-generation oxidation product of isoprene for all its reaction pathways with atmospheric oxidants (Millet et al., 2010)."
I don't know what this means – further clarity in the text is needed here.

"… with air temperature one,"
Remove "one", it's not required here.

"… which can denotes that"
Replace with "denote": "… which can denote that"

"These findings are in agreement with a large result on BVOC oxidation on the local photochemistry.!
Remove this sentence, not required.

"… remained relatively significant during fall …"
significant to what?

and since meteorological conditions in August 2013 were more favorable to photochemical processes
What "meteorological conditions" do the authors refer to here? If it is just the higher solar radiation, then state this in the text.

"Acetone is the OVOC of the selection with generally the highest atmospheric lifetime, considering its photochemical reaction rate …"
Unclear, suggest changing to: "Of the measured OVOCs, acetone has the longest atmospheric lifetime, considering its photochemical reaction rate …"

"Acetone showed similar seasonal variations than formaldehyde and acetaldehyde, …"
Replace with "to": "Acetone showed similar seasonal variations to formaldehyde and acetaldehyde, …"

"… remained significantly high during winter …"
Significant to what? Either include a parameter or remove "significantly" from the sentence.

"… than summer 2013 one, …"

Remove "one", it's not required here.

"… than winter 2014 one, …"
Remove "one", it's not required here.

"… but admitted low enough to allow advection to the receptor site …"
Delete admitted", not needed here.

"… from other OVOC ones …"
Remove "ones", it's not required here.

"Indeed, MEK concentrations did not show seasonal variations except an increasing winter trend …"
Increased concentrations in winter sounds very much like a seasonal variation.  From figure 7, it appears that there is a weak seasonal cycle droning 2013, but this is not replicated (or at least not so clear) in the other years.

"… in February 2013 was by 0.2 µg m-3 higher than …"
Remove "by", it's not required here.

**3.4.4 Comparisons with other VOC measurements performed at Ersa**

I don't think this section is needed as it doesn't say a lot.  Perhaps the link to the supplementary material could be included in one of the earlier discussion sections.

**3.5 VOC factorial analysis**

Perhaps some further explanation of the reasoning behind choosing a subset of the measured compounds is required here?  Along with a discussion of whether limiting the number of species that are included in the PMF analyses may well affect the result and the number of factors.  Perhaps a discussion of how the results here compare to the shorter, intensive campaign results published earlier would help here?

"… should rather be explained as aged profiles originating from several sources assimilating to several source categories …"
Should this be:  "… should rather be explained as aged profiles originating from several source regions comprising several source categories …"?

**3.5.1 Biogenic source (factor 1)**

"The relative load of this VOC for the factor 1 is 70%."
Clarify what is meant by this statement.

"This latter is mainly consisted of primary anthropogenic ..."
What is meant by "This latter"?

**3.5.2 Short-lived anthropogenic sources (factor 2)**

The description of Factor2 and its influences is rather vague and contains a number of potential contributing sources. This is a result of this type of analysis, but the authors need to be wary of making contradicting statements, for example describing "slightly higher contributions during fall" (P17, L27), then "factor 2 contributions were also significant in spring and summer" (P17, L32) and then "mean monthly factor 2 contributions (Fig. 10b2) pointed out no clear seasonal variation over the study period" I think this is due to the differences observed between different years and so care should be taken not to generalise here.

"... with an average contribution to the sum of measured VOC concentrations from this factor of 66%."
Is this correct? Looking at figure 9(b), factor 2 does not appear to ever be 66% of the total.

"... winter, conducting to less dilution of emissions, ..."
suggest changing to "leading to"

"However, factor 2 contributions were also significant in spring and summer ..."
This only appears to be the case in 2013.

"... which could illustrate an enhanced evaporation of gasoline, solvent inks, paints and additional applications during these months as a result of higher temperatures."
This is contradicted by the temperature data shown in figure 4(b1) which shows lower temperatures in June 2013 compared to 2012 and 2014 which have smaller factor 2 contributions shown in figure 10(b2). The authors go on to give explanation of these differences, but I feel it's important to highlight this anomaly here.

**3.5.5 Regional background (factor 5)**

"... probably related to photochemical decay and dilution processes."
Earlier in this section the authors state that natural gas may be an important source for factor 5 so presumably a summer decrease in emissions may also contribute to the observed seasonal variation?

"Mean factor 5 contributions in function of air mass origins were in the same range, except that more elevated contributions were noticed under the influence of European air masses (especially those potentially connected to distant contributions; Fig. 11) compared to the ones related to others continental origins."
This is a confusing sentence; can it be re-written for improved clarity?

**4.1 Determination of controlling factors**

This whole section appears to re-cap the information given in section 3.5. In order to reduce the size of the manuscript, I would suggest these sections be combined to give a more concise explanation of the observations at the site. This could be by either including extra information in section 3.5 (and removing section 4

"… favouring phenomena of vertical dispersion."
Delete "phenomena of", not required here: "… favouring vertical dispersion."

**4.2 The particular case of winter**

Figure 13, referred to in the text needs further explanation and a legend describing the colour scheme and the meaning of C1 – C5. These are described elsewhere, but should be included again here in the figure.

"… compared to winter 2013 ones …"
Remove "ones", it's not required here.

"As a consequence, this finding also point out that shorter observation periods (i.e., up to two months) may be reflected the variability of the identified parameters under the specific meteorological conditions of the studied period."
Sentence is poorly written and doesn't make sense, needs to be re-written for clarity.

**5. VOC concentration variations in continental Europe**

Figure 15 is referred to in the text. The ERSA site should be highlighted in the caption to identify the station under study here.

"… observed in most continental Europe …"
"… observed in most of continental Europe …"

"… were globally lower and …"
not globally, but European wide

"… suggesting a high importance of photochemistry processes and vertical dispersion phenomena in regulating concentration levels."
I would suggest that temperature (linked to boundary layer height) is the main driver here. As the authors state earlier in the manuscript, the majority of these compounds (with the exception of ethylene) have relatively long lifetimes and so photochemistry will likely be limited.

"Then, at stations located …"
Delete "then"

to normal values

How do the authors conclude which is "normal"?

"Then, VOC concentrations …"
Delete "then"

Sentence containing "… was not as warmer-than-average as …" is poorly written, please re-write for clarity

**6. Conclusions**

This section is far too long and needs to be re-written more concisely.

---

## Author Comment (AC1) · 13 Oct 2020

**acp-2020-607: "Seasonal variation and origins of volatile organic compounds observed during two years at a western Mediterranean remote background site (Ersa, Cape Corsica)"**

5 This manuscript presents an analysis of the VOC concentrations and temporal trends over a 2 years long observation campaign at a remote site located in Corsica and representative of the northwest part of the Mediterranean background atmosphere. The monthly, seasonal and interannual variabilities of 21 NMHCs and 4 OVOCs are reported. Source apportionment using positive matrix factorization in combination with back trajectories analysis was carried out on a selection of 14 NMHC species. Not surprisingly, the five factors
10 solution chosen fails to apportion the selected VOCs into their specific emission sources, as the air masses that reach the remote site are already mixed and processed. Finally, the NMHC temporal concentration trends are compared to those observed in the 17 other European background stations at the same period. Overall, while a significant spatial variability in concentration levels is found, especially in winter, similar seasonal trends are observed.

15 The work provides valuable scientific information as long term VOCs datasets in back-ground sites are still rather scarce in the Mediterranean region. To my point of view this topic together with the comparison with the 17 other background sites located in Europe is the most interesting feature of the manuscript. On the other hand, I have some concerns regarding the added value brought by the PMF analysis on a limited set of VOCs and the relevance of the solution as meteorology (boundary layer, air mass circulation, temperature)
20 seems to be the main factor driving the temporal trends in a remote background site. The authors should more clearly elaborate on the limits of the PMF with respect to the limited set of VOCs and samples in such a remote site. As a general comment on language and structure, I find the manuscript in its current state unnecessarily long. Topics are repeated saying pretty much the same thing in different sections. For example, many points raised in section 4.1 (Determination of controlling factors) have already been
25 discussed in the section 3.5 (VOC factor analysis) and points raised in section 4.2 (The particular case of winter) have partly been discussed in section 4.1 when fall-winter interannual trends are discussed. Same remark for sections 3.3 (VOC mixing ratios) and 3.5. Some sections lack clarity (see specific comments). Overall, I find it very difficult to extract the main messages of the sections/paragraphs.

A major revision according to these mentioned general comments is consequently required before
30 publication to ACP.

**Authors' Responses to Referee #1**
We would like to thank the Referee #1 for her/his general feedback and each of her/his useful comments/questions for improving the quality of this manuscript. All comments addressed by both referees have been taken into account in the revised version of the manuscript.

35   We hope that complementary information provided in the responses and incorporated in the manuscript and the supplement will further convince referee #1 on the relevance of the PMF solution examined in this study.

As suggested by both referees, the revised manuscript was largely rewritten. The introduction was shortened. Complementary information on the VOCs selected in this study and on the PMF analysis are now provided in the Supplement (Sects. S1 and S2, respectively). Sections on results have been reviewed in order to better separate information provided by them and hence removed repetitive ones. Section 3.1 ("Meteorological conditions") was shortened only keeping essential pieces of information to the explanation of seasonal and interannual VOC variations. Descriptive Sects. 3.3 and 3.4 ("VOC mixing ratios" and "VOC variability", respectively) have been limited to the presentation of VOC concentration levels, their abundance and their variations and elements of interpretations linked to factors controlling them were removed or moved in Sect. 4 ("Discussions on the seasonal variability of VOC concentrations"). Sect. 3.4.1-3.4.3 were rewritten grouping results to emphasize similar or different VOC behaviours. Note that a larger number of OVOCs is now considered in the Sect. 3.4.3 ("Oxygenated VOCs") and comparisons with other VOC measurements performed at Ersa were moved in the revised Supplement (Sect. S4). Section 3.5 ("Major NMHC sources") was limited to the presentation of the 5 NMHC factors identified in this study and results on their seasonal and interannual variations were removed or moved in Sect. 4. The other factorial analysis previously realized with the summer 2013 VOC dataset has been better used to support factor identification in this study. As a result, Sect. 3.5.6 ("Towards the best experimental strategy to characterize variation in VOC concentrations observed at a remote background site") has been reviewed to better highlight (i) the relevance of the PMF solution to identify NMHC sources and (ii) its limitation to examine VOC concentration variations observed at Ersa. The Sect. 3.5.6 is supported by commentary results presented in the Supplement (Sects. S5 and S6). Section 4 has been restructured in order to distinguish factors controlling VOC concentration variations in spring and summer (Sect. 4.1) from those in fall and winter (Sect. 4.2) and OVOC concentration variations have also been incorporated. The description of the 17 European sites, whose NMHC concentration variations are discussed in Sect. 5 ("VOC concentration variations in continental Europe"), was moved to the Supplement (Sect. S7). The conclusion has also been rewritten.

In this respect, several figures were notably modified including in the supplement. Please note that figures numbers are now different in this new version. Additional sections have been added to the Supplement to present and discuss the selection of the VOCs in this study and the relevance of the PMF solution.

In the present document, authors' answers to the specific comments addressed by Referee #1 are mentioned in **blue**, while changes made to the revised manuscript are shown in **green**. Note that, the comments are listed in the different order than initial one of referee #1, to make easier the understanding of interconnected responses. The comments on the manuscript are listed as follows:

**1/Sect. 1 Introduction**
The introduction could be shortened and focused on the scientific context and goals of the study. For example, information given on the various national and international programs is not essential here (Page 3, line 8-13 ; Page 3, line 32 to Page 4, line 4).

As advised by referee #1, the introduction was shortened in the revised manuscript (of 13 lines). We removed precisions on European short-term field campaigns as the development of long-term observations in Europe were sufficient to explain our motivations to conduct Ersa VOC measurements during two-years. We also removed the presentation of international programs (ACTRIS, EMEP and WMO-GAW) as proposed by referee #1. However, we

[revised manuscript text omitted]

**2/Sect. 2.2.1, Page 6, line 8**

"44 C5-C16" Check consistency with Table 1 where 50 VOC are listed to be sampled with the solid adsorbent.

The 44 $C_5$-$C_{16}$ NMHCs indicated in Sect. 2.2.1 ("VOC measurements") comprise 28 $C_5$ - $C_{16}$ alkanes/alkenes, 6

30 monoterpenes and 10 $C_6$ - $C_9$ aromatics. The six additional VOCs are the 6 $C_6$ - $C_{11}$ n-aldehydes that is why 50 VOCs are listed in Table 1 to be sampled with multi-sorbent cartridges. Moreover, as advised by referee #1, we checked the consistency of the number of VOCs measured per instrument in the revised manuscript. Some corrections were also applied to Table 1 in the revised manuscript to make this information clearer:

**"Table 1: Technical details of the set-up for VOC measurements during the field campaign from June 2012 to June 2014. Air**
35 **samples were collected bi-weekly (every Monday and Thursday) at Ersa from 09:00-13:00 UTC (from early November 2012 to late December 2012 and from early November 2013 to late June 2014) or 12:00-16:00 UTC (from early June 2012 to late October 2012 and from early January 2013 to late October 2013). VOCs are explicitly listed in Sect. S1 of the Supplement.**

| Instrument | Steel canisters | DNPH cartridges – Chemical desorption (acetonitrile) – HPLC-UV | Multi-adsorbent cartridges – Adsorption/thermal desorption – GC-FID |
|---|---|---|---|
| Time Resolution (min) | 240 | 240 | 240 |
| Number of samples | 152 | 91 | 63 |
| Detection limit ($\mu$g m$^{-3}$) | 0.01-0.05 | 0.02-0.05 | 0.01 |

| | | | |
|---|---|---|---|
| Uncertainties $\frac{U(X)}{X}$ mean [min - max] (%) | 25 [7-43] | 23 [6-41] | 26 [7-73] |
| Species | 24 $C_2$ - $C_5$ NMHCs | 15 $C_1$ - $C_6$ carbonyl compounds | 44 $C_5$ - $C_{16}$ NMHCs
6 $C_6$ - $C_{11}$ carbonyl compounds |
| References | Sauvage et al., 2009 | Detournay, 2011;
Detournay et al., 2013 | Ait-Helal et al., 2014;
Detournay, 2011;
Detournay et al., 2011 |

„

**3/Sect. 2.2.1, Page 6, line 1**

It is not clear when these additional 150 off-line air samples were taken; over the 2-year period? at which frequency?

To clarify when the VOC samples (from canisters, DNPH cartridges and multi-sorbent cartridges) were collected over the 2-year period, the Fig. S1 was added in the revised Supplement:

"

[Figure]

**Figure S1: Data collection status indicating when VOC samples were carried out over the two-year period and when concurrent ancillary measurements were realized. The numbers indicated within parentheses correspond to the total number of data observations."**

This figure is indicated in the first paragraph of the revised Sect. 2.2.1 ("VOC measurements"; Page 5 lines 9-13):

"During a period of two years, non-methane hydrocarbons (NMHCs) and OVOCs (carbonyl compounds) were measured routinely employing complementary off-line methods. Four-hour-integrated (09:00-13:00 or 12:00-16:00 UTC) ambient air samples were collected bi-weekly (every Monday and Thursday) into steel canisters and on sorbent

cartridges. The inlets were roughly 1.5 m above the roof of a container housing the analysers. Table 1 describes VOC measurements set up throughout the observation period and Fig. S1 specifies their collection periods."

The same collection days were considered for steel canisters and cartridges (i.e. DNPH and multi-sorbent ones).

**4/Sect. 2.2.3, Page 7**

Section 2.2.3 ("Additional high frequency VOC measurements performed at Ersa") and 3.4.4: the information provided in this section is already given in Section S3 ("comparison of VOC measurements with other ones performed at Ersa"). I suggest removing this section from the main text and to merge it with S3.

As proposed by referees #1 and #2 (comment 48), we removed the sections 2.2.3 ("Additional high frequency VOC measurements performed at Ersa") and 3.4.4 ("Comparisons with other VOC measurements performed at Ersa") and merged all these results in the revised Sect. S4 ("Comparison of VOC measurements with other ones performed at Ersa"). In the revised manuscript, the Sect. S4 is now introduced in Sect. 3.4. ("VOC variability"). Correction applied in the revised manuscript in Sect. 3.4 (Page 11 lines 13-16):

"In addition, the comparison between the VOC monitoring measurements investigated in this study with concurrent campaign measurements performed during the summers 2012-2014 is investigated in Sect. S4 of the Supplement, in order to examine the representativeness of the 2-yr observation period with regard to summer concentration levels."

**5/Sect. 2.3, Page 8, lines 4-11**

Page 8, lines 4-11 are duplicates of lines 22-29 from the Section S1 ("Identification and contribution of major sources of VOCs by EPA PMF 5.0 approach").

The authors decided to merge the information concerning inputs' selection and preparation, and the selection and the optimization of the PMF solution in the revised Sect. S2 ("Identification and contribution of major sources of NMHCs by EPA PMF 5.0 approach"). As a result, the Sect. 2.3 ("Identification and contribution of major sources of VOCs") in the revised manuscript has been shortened. Correction applied in the revised manuscript in Sect. 2.3 (Page 7 lines 2-9):

"In order to characterize NMHC concentrations measured at Ersa, we apportioned them within their sources in this study using the positive matrix factorization approach (PMF; Paatero, 1997; Paatero and Tapper, 1994) applied to our concentration dataset. The PMF mathematical theory has already been presented in Debevec et al. (2017) and is therefore reminded in Sect. S2 of the Supplement. We used the PMF version 5.0, an enhanced tool developed by the Environmental Protection Agency (EPA) and including a multilinear engine ME-2 (Paatero, 1999), and followed the guidance on the use of PMF (Norris et al., 2014). Using NMHC inputs composed of 152 atmospheric data points of 14 variables (13 single primary HCNMs and another one resulting of the grouping of $C_8$ aromatic compounds) and following the methodology presented in Sect S2, a five-factor PMF solution has been selected in this study."

The revised Sect. S2 is presented in the response to referee #1 comment 7.

**6/Sect 2.3, Page 8, lines 4-5**

It appears that the biweekly samples were collected between 09h00-13h00 UTC for 7 months and between 12h00-16h00 UTC for 15 months. The authors should comment the sampling strategy and tell whether this

sampling time shift could impact the PMF analysis and the interannual variations, especially for species with strong diurnal variations such as biogenic VOCs.

The different collection times of the biweekly samples are mainly explained by logistical reasons. The Ersa observation site was created in June 2012 on a windmill farm and involved different measurements carried out by several laboratories (such as IMT Lille-Douai, LSCE, CNRM and Qualitair Corse). Considering the remote location of the study site, laboratories involved in the ChArMEx enhanced observation period at Ersa decided to organize a common routine to manage all instruments. Each instrumental procedure was specified and an intervention was scheduled every two weeks with a turnover of the participants depending on their availability.

To assess the possible effect of this sampling time shift on the PMF analysis, we investigated correlations between reconstructed and observed VOC concentrations in function of the two periods. The results were incorporated in the revised Sect. S2 ("Identification and contribution of major sources of NMHCs by EPA PMF 5.0 approach", the whole revised Sect. S2 is presented in response to referee #1 comment 7):

"S2.4 Optimization of the selected PMF solution
[…]

Moreover, since sampling time of NMHC measurements from canister shifted several times during the two studied years (Sect. 2.2.1 and Table 1), correlations between reconstructed and observed NMHC concentrations as a function of sampling periods were examined (Table S6). Slightly different correlation results were observed for observations resulting from samples collected from 12:00-16:00 UTC (from early June 2012 to late October 2012 and from early January 2013 to late October 2013) compared to those from 09:00-13:00 UTC (from early November 2012 to late December 2012 and from early November 2013 to late June 2014). The PMF model slightly overestimated TVOC concentrations resulting from samples collected from 09:00-13:00 and slightly underestimated those collected from 12:00-16:00, mostly due to the reconstruction of ethane and propane concentrations in both cases. Concerning more reactive NMHCs, ethylene, i-butane, isoprene, toluene and EX concentrations are better reconstructed for samples collected from 12:00-16:00 while propene, i-pentane, n-pentane and n-hexane concentrations are better reconstructed for those from 09:00-13:00. The most impacted species by the sampling time shift was n-pentane, since the PMF model did not identify the sources influencing the high concentrations of n-pentane observed over short periods (see Fig S9) and mostly noticed with samples from 12:00-16:00. More generally, the influence of the sampling time shift on PMF results also depends on the frequency and the amplitude of NMHC concentration variations over short periods for the two cases.

**Table S6: Evaluation of reconstructed NMHC concentrations by the PMF model as a function of the sampling time shift.**

| | All period | | | Samples collected from 09:00-13:00 UTC | | | Samples collected from 12:00-16:00 UTC | | |
|---|---|---|---|---|---|---|---|---|---|
| | slope | intercept | $r^2$ | slope | intercept | $r^2$ | slope | intercept | $r^2$ |
| **Ethane** | 0.997 | 0.006 | 0.999 | 0.986 | 0.029 | 0.996 | 1.002 | 0.001 | 0.999 |
| **Ethylene** | 0.727 | 0.064 | 0.779 | 0.593 | 0.111 | 0.673 | 0.796 | 0.044 | 0.827 |
| **Propane** | 1.000 | -0.011 | 0.968 | 0.929 | 0.052 | 0.949 | 1.046 | -0.035 | 0.977 |
| **Propene** | 0.534 | 0.024 | 0.438 | 0.632 | 0.018 | 0.489 | 0.497 | 0.026 | 0.418 |
| **i-Butane** | 0.832 | 0.029 | 0.897 | 0.761 | 0.056 | 0.869 | 0.893 | 0.015 | 0.904 |
| **n-Butane** | 0.967 | 0.004 | 0.963 | 0.954 | 0.010 | 0.957 | 0.975 | 0.002 | 0.961 |
| **Acetylene** | 0.952 | 0.008 | 0.975 | 0.991 | 0.003 | 0.991 | 0.941 | 0.007 | 0.972 |
| **i-Pentane** | 0.686 | 0.053 | 0.644 | 0.783 | 0.054 | 0.765 | 0.623 | 0.052 | 0.600 |
| **n-Pentane** | 0.421 | 0.081 | 0.332 | 0.852 | 0.032 | 0.788 | 0.295 | 0.085 | 0.238 |

| | | | | | | | | | |
|---|---|---|---|---|---|---|---|---|---|
| Isoprene | 0.956 | 0.005 | 0.996 | 0.872 | 0.005 | 0.996 | 0.971 | 0.006 | 0.998 |
| n-Hexane | 0.510 | 0.032 | 0.519 | 0.668 | 0.026 | 0.738 | 0.419 | 0.035 | 0.410 |
| Benzene | 0.889 | 0.022 | 0.895 | 0.918 | 0.027 | 0.849 | 0.873 | 0.018 | 0.911 |
| Toluene | 0.582 | 0.092 | 0.599 | 0.526 | 0.105 | 0.501 | 0.626 | 0.083 | 0.669 |
| EX | 0.527 | 0.103 | 0.515 | 0.452 | 0.112 | 0.431 | 0.582 | 0.095 | 0.578 |
| TVOC | 1.004 | -0.186 | 0.992 | 0.988 | -0.083 | 0.987 | 1.009 | -0.213 | 0.993 |

"

Given these results, average concentrations are moderated in the revised Sect. 3.3 ("VOC mixing ratios") by clarifying them for the two periods in Table S1 of the revised Supplement:

"Table S1: Average concentrations ± standard deviations (µg m$^{-3}$) of selected VOCs measured at Ersa from June 2012 to June 2014 as a function of the measurement sampling times (see Table 1).

| | Species | Samples collected from 09:00-13:00 | Samples collected from 12:00-16:00 |
|---|---|---|---|
| BVOCs | Isoprene | 0.08 ± 0.21 | 0.21 ± 0.35 |
| | α-Pinene | 0.13 ± 0.11 | 0.49 ± 0.71 |
| | Camphene | 0.01 ± 0.03 | 0.03 ± 0.07 |
| | α-Terpinene | 0.02 ± 0.03 | 0.09 ± 0.18 |
| | Limonene | 0.08 ± 0.17 | 0.24 ± 0.34 |
| Anthropogenic NMHCs | Ethane | 2.43 ± 0.70 | 1.57 ± 0.80 |
| | Propane | 1.28 ± 0.62 | 0.81 ± 0.62 |
| | i-Butane | 0.36 ± 0.25 | 0.19 ± 0.16 |
| | n-Butane | 0.51 ± 0.29 | 0.31 ± 0.25 |
| | i-Pentane | 0.32 ± 0.26 | 0.26 ± 0.22 |
| | n-Pentane | 0.27 ± 0.28 | 0.23 ± 0.21 |
| | n-Hexane | 0.09 ± 0.06 | 0.08 ± 0.05 |
| | Ethylene | 0.38 ± 0.20 | 0.30 ± 0.18 |
| | Propene | 0.07 ± 0.04 | 0.07 ± 0.04 |
| | Acetylene | 0.31 ± 0.20 | 0.25 ± 0.27 |
| | Benzene | 0.35 ± 0.16 | 0.30 ± 0.22 |
| | Toluene | 0.37 ± 0.26 | 0.30 ± 0.24 |
| | Ethylbenzene | 0.06 ± 0.07 | 0.05 ± 0.07 |
| | m,p-Xylenes | 0.14 ± 0.15 | 0.15 ± 0.14 |
| | o-Xylene | 0.07 ± 0.09 | 0.09 ± 0.09 |
| OVOCs | Formaldehyde | 0.96 ± 0.48 | 1.82 ± 1.44 |
| | Acetaldehyde | 0.68 ± 0.17 | 1.11 ± 0.44 |
| | i,n-Butanals | 0.13 ± 0.07 | 0.34 ± 0.69 |
| | n-Hexanal | 0.15 ± 0.10 | 0.26 ± 0.32 |
| | Benzaldehyde | 0.15 ± 0.12 | 0.15 ± 0.12 |
| | n-Octanal | 0..07 ± 0.05 | 0.13 ± 0.24 |
| | n-Nonanal | 0.49 ± 0.43 | 0.18 ± 0.15 |
| | n-Decanal | 0.43 ± 0.34 | 0.14 ± 0.13 |
| | n-Undecanal | 0.09 ± 0.06 | 0.06 ± 0.06 |
| | Glyoxal | 0.07 ± 0.04 | 0.07 ± 0.05 |
| | Methylglyoxal | 0.07 ± 0.04 | 0.21 ± 0.16 |
| | Acetone | 3.32 ± 1.77 | 4.84 ± 2.95 |
| | MEK | 0.34 ± 0.11 | 0.37 ± 0.16 |

"

Factor contributions are also moderated in the revised Sect. 3.5 ("Major NMHC sources") by clarifying them for the two periods in Table 5 of the revised Supplement:

"Table 5: Average relative factor contributions ± standard deviations (%) for the whole period and as a function of the measurement sampling times (see Table 1).

| Factor | 2-yr period | Samples collected from 09:00-13:00 | Samples collected from 12:00-16:00 |
|---|---|---|---|
| Regional background | 39 ± 10 | 44 ± 10 | 38 ± 9 |
| Evaporative sources | 22 ± 10 | 23 ± 11 | 17± 9 |
| Short-lived anthropogenic sources | 19 ± 10 | 16 ± 7 | 23 ± 10 |
| Long-lived combustion sources | 16 ± 7 | 15 ± 5 | 14 ± 7 |
| Local biogenic source | 4 ± 10 | 2 ± 7 | 8 ± 11 |

"

Concerning interannual BVOC variations, some precautions were taken for BVOC results in Sect. 3.4.1 ("Biogenic VOCs"; Page 12 lines 11-14) of the revised manuscript:

"Note that the interpretation of interannual variations in BVOC measurements is based on a limited number of sampling days during the study period and different collection times (Table 1 and Sect. 2.2.1). It should then be considered cautiously given variable day-to-day and strong diurnal BVOC variations which were observed during the summer 2013 observation period (Kalogridis, 2014)."

Results were also moderated in Sect. 3.5.1 ("Local biogenic source (factor 1)"; Page 15, lines 5-11):

"Note that factor 1 contribution to selected NMHC concentrations observed at Ersa during the 2-yr period may be slightly influenced by the two different sampling times used during the 2-yr observation period (Table 5) and the number of VOCs and data points considered in the PMF analysis (see Sects. 3.4.1 and S2). However, Michoud et al. (2017) has provided additional information on this local primary biogenic source in summer, such as the contributions of additional primary BVOCs (the sum of monoterpenes) and some OVOCs (carboxylic acids, methanol and acetone) and the clear diurnal variations of the local primary biogenic source."

Reference

Kalogridis, A.: Caractérisation des composés organiques volatils en région méditerranéenne, Université Paris Sud - Paris XI. [online] Available from: https://tel.archives-ouvertes.fr/tel-01165005, 2014.

**7/Sect. 2.3, Page 8, lines 16-20**

It is said that the PMF model results reconstructs on average 99% of the total concentration of the 14 selected species, but in the meantime 5 out of the 14 selected are not properly captured by the PMF solution. Also, ethane and propane account for 50% of the VOC mass. In these given circumstances, is the percentage of total reconstructed mass relevant to assess the quality of the PMF solution? It would be helpful to include more information on the PMF preprocessing, and on the diagnostic plots (Q/Q(exp) values vs number of factors, scatter plots of the measured vs. reconstructed concentrations, scaled residuals, if and why outliers were removed from the time series, etc..).

Firstly, the high proportion of long-lived VOCs usually characterizes remote background sites such as Ersa (e.g., Sauvage et al., 2009 and Leuchner et al., 2015) since more reactive species are more prone to react before reaching the receptor sites. Moreover, the reconstruction of measured concentrations by the PMF depends

mainly on VOC concentration levels, their variability as well as their uncertainties (related to their signal-to-noise ratio, Debevec et al., 2017 – see Table A below). Lower relative uncertainties are globally related to long-lived VOCs in this study compared to those of more reactive VOCs (see Table 3; see response to referee #1 comment 15). As a consequence, the PMF model will favour in the study the reconstruction of the VOCs with the longest lifetime, the highest concentration levels and the highest S/N ratios (i.e. ethane, propane, i-butane, n-butane and acetylene – see Table A) to maximize the reconstruction of the total variable (TVOC in this study). Given these circumstances, the percentage of the total reconstructed mass is relevant to assess the quality of the PMF solution but is not sufficient alone. As a result, the methodology to identify the optimal factor of the PMF solution for this study is developed in the revised version of Sect. S2 ("Identification and contribution of major sources of NMHCs by PMF 5.0 approach"). Furthermore, note that no outlier was removed from the dataset in order to maximize the number of input data. Sharp increases in VOC concentrations were qualified by higher uncertainties than other input data. The PMF model was hence less sensitive to them, as it takes into account the quality of input data by means of their possible weighing as a function of their uncertainties. Difference between $Q_{robust}$ and $Q_{true}$ of the selected PMF solution in this study was only of 18% (Table S5), indicating some but not heavy impact of outliers on the Q-value (Norris et al, 2014).

**Table A: capture of our input data statistics obtained with the EPA PMF 5.0 tool.**

| Species | Cat | S/N | Min | 25th | 50th | 75th | Max | % Modeled Samples | % Raw Samples |
|---|---|---|---|---|---|---|---|---|---|
| TVOC | Weak | 8.5 | 2.05705 | 3.55691 | 4.79085 | 6.94215 | 12.95668 | 100.00 % | 100.00 % |
| Ethane | Strong | 10.0 | 0.57259 | 1.12026 | 1.84826 | 2.47359 | 4.28298 | 100.00 % | 100.00 % |
| Ethylene | Strong | 6.4 | 0.08883 | 0.18924 | 0.27503 | 0.38892 | 0.86785 | 100.00 % | 100.00 % |
| Propane | Strong | 8.5 | 0.18175 | 0.43437 | 0.77296 | 1.40625 | 2.60039 | 100.00 % | 100.00 % |
| Propene | Strong | 1.9 | 0.00877 | 0.04306 | 0.06392 | 0.08571 | 0.17491 | 100.00 % | 100.00 % |
| iButane | Strong | 4.7 | 0.01211 | 0.09467 | 0.16662 | 0.35604 | 1.01686 | 100.00 % | 100.00 % |
| nButane | Strong | 7.2 | 0.04877 | 0.15996 | 0.26471 | 0.57146 | 1.09023 | 100.00 % | 100.00 % |
| Acetylene | Strong | 7.4 | 0.03233 | 0.09128 | 0.17764 | 0.36635 | 1.23359 | 100.00 % | 100.00 % |
| iPentane | Strong | 2.9 | 0.05555 | 0.14615 | 0.22100 | 0.31031 | 0.89557 | 100.00 % | 100.00 % |
| nPentane | Strong | 2.1 | 0.01503 | 0.09355 | 0.18038 | 0.27692 | 0.79693 | 100.00 % | 100.00 % |
| Isoprene | Strong | 1.7 | 0.01419 | 0.01419 | 0.03640 | 0.16245 | 2.28200 | 100.00 % | 100.00 % |
| nHexane | Strong | 1.3 | 0.01795 | 0.04211 | 0.06887 | 0.09543 | 0.26892 | 100.00 % | 100.00 % |
| Benzene | Strong | 3.2 | 0.06840 | 0.16302 | 0.26098 | 0.38597 | 1.11023 | 100.00 % | 100.00 % |
| Toluene | Strong | 3.0 | 0.04303 | 0.15357 | 0.23233 | 0.34467 | 0.83595 | 100.00 % | 100.00 % |
| EX | Strong | 1.9 | 0.06636 | 0.11917 | 0.19071 | 0.27694 | 0.82052 | 100.00 % | 100.00 % |

Correction applied in the revised supplement materials in Sect. S2:

"To characterize VOC concentrations measured at Ersa, we apportioned VOC sources in this study using the positive matrix factorization approach (PMF; Paatero, 1997; Paatero and Tapper, 1994) applied to our VOC concentration dataset. We used the PMF version 5.0, an enhanced tool developed by the Environmental Protection Agency (EPA) and including a multilinear engine ME-2 (Paatero, 1999), and followed the guidance on the use of PMF (Norris et al., 2014).

**S2.1 PMF approach**

PMF is a tool elaborated for a multivariate factor analysis and used for the identification and the characterization of the "p" independent sources of "n" species measured "m" times at a given site. Note that the PMF mathematical theory is detailed elsewhere (Paatero, 1997; Paatero and Tapper, 1994). Concisely, the PMF method is based on the decomposition of a matrix of chemically speciated sample data (of dimension n x m) into two matrices of factor profiles (n x p) and factor contributions (p x m), interpreting each factor as a different source type. Species profiles of each source identified represent the repartition of each species into each given factor, and the amount of mass contributed by each factor to each successive individual sample represents the evolution in time of the contribution from each factor to the various species. The principle can be condensed as:

$$x_{ij} = \sum_{k=1}^{p} g_{jk} \times f_{ki} + e_{ij} = c_{ij} + e_{ij} , \tag{1}$$

where $x_{ij}$ is the $i^{th}$ species measured concentration (in µg m$^{-3}$ here) in the $j^{th}$ sample, $f_{ki}$ the $i^{th}$ mass fraction from $k^{th}$ source, $g_{jk}$ the $k^{th}$ source contribution of the $j^{th}$ sample, $e_{ij}$ the residual resulting of the decomposition and $c_{ij}$ the species reconstructed concentration. The Eq. (1) can be solved iteratively by minimizing the residual sum of squares Q following Eq. (2):

$$Q = \sum_{i=1}^{n} \sum_{j=1}^{m} \left( \frac{e_{ij}}{s_{ij}} \right)^2 , \tag{2}$$

with $s_{ij}$, the extended uncertainty (in µg m$^{-3}$) related to the measured concentration of the $i^{th}$ species in the $j^{th}$ sample. A user-provided uncertainty following the procedure presented in Polissar et al. (1998) is also required by the PMF tool to weight individual points. Moreover, negative source contributions are not allowed.

**S2.2 VOC dataset and data preparation**

In order to have sufficient completeness (in terms of observation number), only primary HCNM measurements from bi-weekly ambient air samples collected into steel canisters from 04 June 2012 to 27 June 2014 were retained in this factorial analysis. The NMHC dataset encompassed 152 atmospheric data points having a time resolution of 4 hours. VOC observations resulting from DNPH and multi-sorbent cartridges were not considered in the PMF analysis since they were sampled only 73 and 52 days concurrently to the collection of steel canisters (Fig. S1). Reconstruction of missing data points would significantly affect the dataset quality. Additionally, the restriction of the number of data points to those common to the three datasets (36 data points) would significantly impact the temporal representativeness of the VOC inputs of the study period and hence limit the discussion on interannual and seasonal variations for statistical robustness reasons. Note that no outlier was removed from the dataset.

NMHC inputs were built using the concentrations of the 17 HCNMs selected in this study (see Sect. S1). The final chemical dataset includes 13 single variables and a grouped one. This latter named "EX" grouped the concentrations of $C_8$ aromatic compounds, in order to maximize its concentration levels.

Moreover, the data preprocessing and quality analysis of the NMHC dataset are presented in the supplement material of Debevec et al. (2017). Since signal-to-noise (S/N) ratios of the 14 variables retained for the factorial analysis are all higher than 1.2, in this study no variable was categorized as "weak", and hence downweighted (categorize variables in "weak" means to triple their original uncertainties; Norris et al., 2014).

**S2.3 Selected PMF Solution**

In order to identify the optimal number of factors for the PMF solution selected in this study, the first step consisted in carrying out numerous successive base runs considering an incremented factor number according to the protocol defined by Sauvage et al. (2009). As a result, PMF solutions composed from 2 to 10 factors, considering 100 runs and a random start, were explored.

Firstly, the selection of the solution among PMF solutions of 2 to 10 factors is based on the analysis of diverse exploratory statistical parameters (Table S3 and Fig. S6) which are as follows:
- Variations in $Q_{true}$ and $Q_{theorical}$ as a function of the factor number of the PMF solution. $Q_{true}$ is provided by the EPA PMF tool (Norris et al., 2014) following the launch of a base model run. $Q_{theorical}$ is a calculated parameter following the

equation (3). $Q_{true}$ and $Q_{theorical}$ tend to decrease when the factor number increases. A PMF user can choose the PMF solution having a lower $Q_{true}$ compared to the associated $Q_{theorical}$.

- Variation in IM and IS (maximum individual standard deviation and maximum individual column mean, respectively) as a function of the factor number of the PMF solution. IM and IS can be defined following equations (4) and (5), respectively. A PMF user can choose the PMF solution corresponding to a significant break in the slope of IM and/or IS (see also the relative differences d(IM) and d(IS) in Table S3) in function of factor number.

- Variations in average determination coefficients between reconstructed concentrations of the total variable (called in this study TVOC, see Norris et al., 2014) and measured ones ($R^2$(TVOC)). A PMF user can choose the PMF solution of p factors corresponding to a significant increase of $R^2$(TVOC) compared to the PMF solution of p-1 factors.

- An optimal PMF solution should also present a symmetrical distribution of residual values related to the total variable as well as a large proportion of them ranging between -2 and 2, especially between -0.3 and 0.3.

$$for\ p \in [2, 10], Q_{theorical} = M \times N - p \times (M + N), \tag{3}$$

with M=152 and N=14 in this study (Sect S2.2).

$$IM = \max\left(\frac{1}{M}\sum_{j=1}^{M}\frac{e_{ij}}{s_{ij}}\right), among\ i \in [1, N] \tag{4}$$

$$IS = \max\left(\sqrt{\frac{1}{M-1}\sum_{j=1}^{M}\left[\frac{e_{ij}}{s_{ij}} - \overline{\left(\frac{e_{ij}}{s_{ij}}\right)}\right]^2}\right), among\ i \in [1, N] \tag{5}$$

The visual inspection of statistical indicators was realized following Fig. S6. Significant breaks in slope of variations of IM as a function of the factor number of the PMF solution were noticed for PMF solutions composed from 3 to 5 factors, from 4 to 6 factors and from 7 to 9 factors (Fig. S6c). Moreover, a significant break in slope of variations of IS as a function of the factor number of the PMF solution was only noticed for PMF solutions composed from 5 to 7 factors (Fig. S6d). $R^2$(TVOC) increases significantly between PMF solutions of 3 and 4 factors and to a lesser extent between PMF solutions from 4 to 7 factors (Fig. S6e). Contrarily, $R^2$(TVOC) decreases significantly between PMF solutions of 7 and 8 factors. However, $Q_{true}$ is lower than $Q_{theorical}$ from a PMF solution of 8 factors (Fig. S6a). From a PMF solution of 4 factors, the proportion of residual values ranging between -2 and 2 is higher than 90% and from a PMF solution of 5 factors, the proportion of residual values ranging between -0.3 and 0.3 is higher than 40% (Fig. S6b). As a result, we oriented our choice of optimal PMF solution from 4 to 6 factors.

In order to refine this choice, we also examined correlations between reconstructed concentrations and measured ones for individual species of the selected PMF solutions (Figs S7-S9 and Table S4), their distribution of residual values (Fig S10), the physical meaning of their factor profiles (Fig S11), their factor contribution time series (Fig S11) and correlations between their factors. From a PMF solution of 4 factors, the model identified a factor related to a biogenic source (factor 1 depicted in Fig. S11 and related to isoprene concentrations). A better reconstruction of ethane, acetylene and isoprene concentrations was noticed for a PMF solution of 4 factors (Fig S7). We did not observe any correlation between factors composing the 4-factor PMF solution. From a PMF solution of 5 factors, the model distinguished a factor related to the more reactive species (factor 2 profile composed of ethylene, propene, toluene and EX - Fig. S11) from the factor associated with evaporation sources (factor 3 profile composed of propane, i,n-butanes and i,n-pentanes - Fig. S11). These two factors are not correlated (determination coefficient: 0.35). This deconvolution notably improved the reconstruction by the PMF model of concentrations of ethylene, propene, toluene and EX (Figs. S7 and S9 and Table S4) and slightly improved the distribution of residual values for propene and toluene (Fig. S10). Ethane and isoprene concentrations are fully reconstructed with the PMF solution of 5 factors (Fig. S8 and Table S4) and their residual values were more symmetrical and gathered between -1 and 1 (Fig. S10). The additional factor composing the 6-factor PMF solution compared to the 5-factor one results from the split of the factor related to the more reactive species into two factors. The first one (factor 2 – Fig. S11) is mostly composed of ethylene and propene while the second one (factor 3 – Fig. S11) is composed of propene, i,n-pentanes, toluene and EX. These two factors are not correlated (determination coefficient: 0.02 – Fig. S11). This deconvolution notably improved the reconstruction of ethylene concentrations (Fig. S9 and Table S4), slightly improved the reconstruction of i,n-pentanes, toluene and EX

concentrations but degraded propene ones. In terms of residual value distribution, the 6-factor PMF solution mostly improved the ethylene one (Fig. S9). However, ethylene, propene, i,n-pentanes, toluene and EX concentrations observed at Ersa in summer 2013 were mainly explained by the same factor according to Michoud et al. (2017), which supported our choice of a 5-factor PMF solution for this study.

**Table S3: Exploratory statistical parameters for the identification of the optimal number of factors of the PMF solution.**

| Factor number | Q theorical | Q robust mod | Q true | IM | IS | Proportion of residuals between [-2 ;2] | Proportion of residuals > abs(0,3) | Determination coefficient PMF results vs Meas. ($R^2$) | d(IM) = (IM(p) - IM(p-1))/ IM(p-1) | d(IS) = (IS(p) - IS(p-1))/ IS(p-1) |
|---|---|---|---|---|---|---|---|---|---|---|
| 2 | 1796 | 6557 | 7472 | 0.9249 | 3.1160 | 0.7904 | 0.8008 | 0.9653 | - | - |
| 3 | 1630 | 4352 | 4749 | 0.8838 | 2.5538 | 0.8604 | 0.7702 | 0.9746 | 0.0444 | 0.1804 |
| 4 | 1464 | 3057 | 3169 | 0.4239 | 1.9464 | 0.9037 | 0.7049 | 0.9879 | 0.5204 | 0.2378 |
| 5 | 1298 | 2092 | 2120 | 0.2659 | 1.5157 | 0.9441 | 0.6109 | 0.9920 | 0.3727 | 0.2213 |
| 6 | 1132 | 1545 | 1547 | 0.2260 | 1.1361 | 0.9615 | 0.5550 | 0.9939 | 0.1503 | 0.2504 |
| 7 | 966 | 1161 | 1162 | 0.2255 | 1.0810 | 0.9737 | 0.5028 | 0.9952 | 0.0021 | 0.0485 |
| 8 | 800 | 777 | 777 | 0.1153 | 0.9373 | 0.9864 | 0.4384 | 0.9883 | 0.4885 | 0.1329 |
| 9 | 634 | 558 | 558 | 0.1133 | 0.8282 | 0.9915 | 0.3435 | 0.9873 | 0.0180 | 0.1164 |
| 10 | 468 | 380 | 380 | 0.0939 | 0.7596 | 0.9953 | 0.2740 | 0.9836 | 0.1713 | 0.0828 |

**Table S4: Evaluation of reconstructed NMHC concentrations of PMF solutions from 4 to 6 factors as a function of measured NMHC concentrations.**

| VOC | $r^2$ | | | slope | | | intercept | | |
|---|---|---|---|---|---|---|---|---|---|
| | 4 factors | 5 factors | 6 factors | 4 factors | 5 factors | 6 factors | 4 factors | 5 factors | 6 factors |
| (0) TVOC | 0.988 | 0.992 | 0.994 | 1.003 | 1.013 | 1.013 | -0.034 | -0.072 | -0.060 |
| (1) Ethane | 0.992 | 0.998 | 0.999 | 0.977 | 0.994 | 1.000 | 0.037 | 0.009 | -0.001 |
| (2) Ethylene | 0.666 | 0.771 | 0.985 | 0.618 | 0.722 | 0.938 | 0.086 | 0.065 | 0.017 |
| (3) Propane | 0.950 | 0.968 | 0.969 | 0.990 | 1.002 | 1.007 | -0.010 | -0.013 | -0.016 |
| (4) Propene | 0.275 | 0.454 | 0.411 | 0.350 | 0.534 | 0.488 | 0.034 | 0.024 | 0.026 |
| (5) i-Butane | 0.894 | 0.909 | 0.913 | 0.820 | 0.833 | 0.842 | 0.033 | 0.030 | 0.028 |
| (6) n-Butane | 0.946 | 0.969 | 0.969 | 0.953 | 0.969 | 0.968 | 0.009 | 0.005 | 0.006 |
| (7) Acetylene | 0.989 | 0.993 | 0.989 | 0.952 | 0.971 | 0.973 | 0.008 | 0.006 | 0.006 |
| (8) i-Pentane | 0.657 | 0.654 | 0.712 | 0.692 | 0.687 | 0.743 | 0.053 | 0.054 | 0.046 |
| (9) n-Pentane | 0.328 | 0.331 | 0.378 | 0.421 | 0.419 | 0.470 | 0.082 | 0.082 | 0.077 |
| (10) Isoprene | 0.568 | 0.995 | 0.995 | 0.362 | 0.998 | 1.009 | 0.054 | -0.0004 | -0.002 |
| (11) n-Hexane | 0.560 | 0.537 | 0.582 | 0.546 | 0.546 | 0.583 | 0.029 | 0.029 | 0.027 |
| (12) Benzene | 0.898 | 0.918 | 0.908 | 0.858 | 0.896 | 0.867 | 0.031 | 0.022 | 0.029 |
| (13) Toluene | 0.539 | 0.600 | 0.630 | 0.467 | 0.597 | 0.677 | 0.113 | 0.090 | 0.083 |
| (14) EX | 0.342 | 0.536 | 0.623 | 0.332 | 0.548 | 0.665 | 0.134 | 0.102 | 0.079 |

[Figure]

**Figure S6: Variations of exploratory statistical parameters as a function of the number of factors of PMF solutions.**

[Figure]

**Figure S7: Correlations between reconstructed NMHC concentrations by the PMF model and measured ones as a function of the factor number of PMF solutions.**

[Figure]

[Figure]

**Figure S8: Time series of reconstructed NMHC concentrations for PMF solutions from 4 to 6 factors compared to NMHC measured concentrations. Note that only results of NMHCs not well reconstructed by the PMF model (r² < 0.85, see Table S4) are presented.**

[Figure]

[Figure]

**Figure S9: Scatter plots of reconstructed NMHC concentrations for PMF solutions from 4 to 6 factors and NMHC measured concentrations. Note that only results of NMHCs not well reconstructed by the PMF model ($r^2 < 0.85$, see Table S4) are presented.**

[Figure]

[Figure]

Figure S10: Distributions of scaled residuals as a function of NMHC for PMF solutions composed from 4 to 6 factors.

[Figure]

[Figure]

Figure S11: Factor profiles and normalized contribution time series of PMF solutions from 4 to 6 factors. Note that NMHCs 0-14 are listed in Table S4.

**S2.4 Optimization of the selected PMF solution**

Generally, the non-negativity constraint alone is considered not enough to obtain a unique solution. To reduce the number of solutions, one possible approach is to rotate a given solution and assess the obtained results with the initial solution. Consequently, the optimization of the selected 5-factor PMF solution relies on the exploration of the rotational freedom of the selected PMF solution by acting on the $F_{peak}$ parameter (Paatero et al., 2005; Paatero et al., 2002) following recommendations of Norris et al. (2014) so as to reach an optimized final solution. As a result, a $F_{peak}$ parameter fixed at 0.8 was applied to the selected PMF solution which allowed a finer decomposition of the NMHC dataset following an acceptable change of the Q-value (Norris et al., 2014).

Quality indicators provided by the EPA PMF application have been indicated in Table S5. The PMF model results reconstruct on average 99% of the total concentration of the 14 selected compounds of this study. Individually, almost all chemical species also showed both good determination coefficients and slopes (close to 1 – Table S4) between reconstructed and measured concentrations, apart from propene, n-pentane, n-hexane and EX (see Fig. S9). The PMF model reconstructs well the variations of these species over long periods (Fig. S8) but not over short-periods, explaining their lower determination coefficients and slopes farther from 1 (Table S4). Therefore, PMF model limitations to explain these species should be kept in mind when examining PMF results.

**Table S5: Input information and mathematical diagnostic for the results of PMF analysis.**

| Input information | | |
|---|---|---|
| Samples | N | 152 |
| Species | M | 14 |
| Factors | P | 5 |
| Runs | | 100 |
| Nb. Species indicated as weak | | 0 |
| $F_{peak}$ | | 0.8 |
| **Model quality** | | |
| Q robust | Q(r) | 2589.7 |
| Q true | Q(t) | 2119.9 |
| Maximum individual standard deviation | IM | 0.27 |
| Maximum individual column mean | IS | 1.52 |
| Mean ratio (modelled vs. measured) | Slope(TVOC) | 1.01 |
| TVOC$_{modelled}$ vs. TVOC$_{measured}$ | R$^2$(TVOC) | 0.99 |
| Nb. of species with R$^2$ > 0.6 | | 10 |
| Nb. of species with 1.1 > slope > 0.6 | | 9 |

The evaluation of rotational ambiguity and random errors in a given PMF solution can be realized with DISP (displacement) and BS (bootstrap) error estimation methods (Brown et al., 2015; Norris et al., 2014; Paatero et al., 2014). As no factor swap occurred in the DISP analysis results, the 5-factor PMF solution is considered adequately robust to be interpreted. Then, bootstrapping was realized by performing 100 runs, and considering a random seed, a block size of 18 samples and a minimum Pearson correlation coefficient of 0.6. Each modeled factor of the selected PMF solution was well mapped over at least 95% of realized runs, assuring their reproducibility.

Moreover, since sampling time of NMHC measurements from canister shifted several times during the two studied years (Table 1), correlations between reconstructed and observed NMHC concentrations as a function of sampling periods are investigated (Table S6). Slightly different correlation results were observed for observations resulting from samples collected from 12:00-16:00 UTC (from early June 2012 to late October 2012 and from early January 2013 to late October 2013) compared to those from 09:00-13:00 UTC (from early November 2012 to late December 2012 and from early November 2013 to late June 2014). The PMF model slightly overestimated TVOC concentrations resulting from samples collected from 09:00-13:00 and slightly underestimated those collected from 12:00-16:00, mostly due to reconstruction of ethane and propane concentrations in both cases. Concerning more reactive NMHCs, ethylene, i-butane, isoprene, toluene and EX concentrations are better reconstructed for samples collected from 12:00-16:00 while propene, i-pentane, n-pentane and n-hexane concentrations are better reconstructed for those from 09:00-13:00. The most impacted species by the sampling time shift was n-pentane, since the PMF model did not identify the sources influencing the high concentrations of n-pentane observed over short periods (see Fig S9) and this was mostly noticeable with the 12:00-16:00 sample set. More generally, the influence of the sampling time shift on PMF results also depends on the frequency and the amplitude of NMHC concentration variations over short periods for the two sample sets.

**Table S6: Evaluation of reconstructed VOC concentrations by the PMF model as a function of the sampling time shift.**

| | All period | | | Samples collected from 09:00-13:00 UTC | | | Samples collected from 12:00-16:00 UTC | | |
|---|---|---|---|---|---|---|---|---|---|
| | slope | intercept | r² | slope | intercept | r² | slope | intercept | r² |
| Ethane | 0.997 | 0.006 | 0.999 | 0.986 | 0.029 | 0.996 | 1.002 | 0.001 | 0.999 |
| Ethylene | 0.727 | 0.064 | 0.779 | 0.593 | 0.111 | 0.673 | 0.796 | 0.044 | 0.827 |
| Propane | 1.000 | -0.011 | 0.968 | 0.929 | 0.052 | 0.949 | 1.046 | -0.035 | 0.977 |
| Propene | 0.534 | 0.024 | 0.438 | 0.632 | 0.018 | 0.489 | 0.497 | 0.026 | 0.418 |
| i-Butane | 0.832 | 0.029 | 0.897 | 0.761 | 0.056 | 0.869 | 0.893 | 0.015 | 0.904 |
| n-Butane | 0.967 | 0.004 | 0.963 | 0.954 | 0.010 | 0.957 | 0.975 | 0.002 | 0.961 |
| Acetylene | 0.952 | 0.008 | 0.975 | 0.991 | 0.003 | 0.991 | 0.941 | 0.007 | 0.972 |
| i-Pentane | 0.686 | 0.053 | 0.644 | 0.783 | 0.054 | 0.765 | 0.623 | 0.052 | 0.600 |
| n-Pentane | 0.421 | 0.081 | 0.332 | 0.852 | 0.032 | 0.788 | 0.295 | 0.085 | 0.238 |
| Isoprene | 0.956 | 0.005 | 0.996 | 0.872 | 0.005 | 0.996 | 0.971 | 0.006 | 0.998 |
| n-Hexane | 0.510 | 0.032 | 0.519 | 0.668 | 0.026 | 0.738 | 0.419 | 0.035 | 0.410 |
| Benzene | 0.889 | 0.022 | 0.895 | 0.918 | 0.027 | 0.849 | 0.873 | 0.018 | 0.911 |
| Toluene | 0.582 | 0.092 | 0.599 | 0.526 | 0.105 | 0.501 | 0.626 | 0.083 | 0.669 |
| EX | 0.527 | 0.103 | 0.515 | 0.452 | 0.112 | 0.431 | 0.582 | 0.095 | 0.578 |
| TVOC | 1.004 | -0.186 | 0.992 | 0.988 | -0.083 | 0.987 | 1.009 | -0.213 | 0.993 |

"


**9/Sect. 2.3, Page 8, line 19**

The authors specify that "PMF model limitations to explain these species should be kept in mind when examining PMF results". A more detailed discussion would be helpful for the reader to appreciate the limits of the proposed PMF solution. A rough estimate based on figure 8 indicates that these species make up approximatively 80% of the concentration of Factor 2 (short-lived species).

Firstly, the high proportion of long-lived VOCs (such as ethane and propane) usually characterized remote background sites such as Ersa (e.g., Sauvage et al., 2009 and Leuchner et al., 2015) since more reactive species are more prone to react before reaching the receptor sites. Reactive species such as propene, n-hexane, toluene, and $C_8$ aromatics (EX) have shown low concentration levels (Table 3 in the revised version of the manuscript, see response to referee #1 comment 15) and high variability, especially on short periods (Fig. S8 in the revised version of the supplement, see response to referee #1 comment 7), compared to long-lived VOCs.

Moreover, the reconstruction of measured concentrations by the PMF depends on VOC concentration levels, their variability as well as their uncertainties (related to their signal-to-noise ratio, Debevec et al., 2017 – see responses to referee #1 comments 7 and 8). More reactive species are prone to have higher relative uncertainties degrading their S/N ratios (Table A - see response to referee #1 comment 7) and hence have lower S/N ratios compared to VOCs with longer lifetime. As a result, the PMF model favoured in this study the reconstruction of the NMHCs with the longest lifetime, the highest concentration levels and the highest S/N ratios (i.e. ethane, propane, i-butane, n-butane and acetylene – see Table A) to maximize the reconstruction of the total variable (TVOC). The improvement of the capture of more reactive NMHCs may demand to increase the number factor of the PMF solution (as shown with the PMF solution composed of 6 factors in Sect. S2.3 - response to referee #1 comment 7). However, due to the limiting number of VOCs and the temporal coverage and the time resolution of the observations considered in this factorial analysis, a PMF solution with a higher number of factors will statistically improve the reconstruction of the more reactive species, but the additional factors may not have a physical meaning.

As a result, we decided to select and examine a PMF solution composed of 5 factors, to the detriment of the reconstruction of concentrations of more reactive species. The proposed PMF solution did not reconstruct well 5 NMHCs, which were propene, n-pentane, n-hexane, toluene, and EX, following their correlation between reconstructed and observed concentrations (Table S4 and Fig. S9 - see response to referee #1 comment 7), hence the remark "PMF model limitations to explain these species should be kept in mind when examining PMF results". However, the PMF model reconstructed well the variations of these species over long periods (Fig. S8) but not over short-periods, explaining their lower determination coefficients and slopes farther from 1. The aim of this study is to present and discuss ambient levels and variations of a selection of VOCs observed at the Ersa station over more than two years. We principally investigated seasonal and interannual variations in this study. The good reconstruction by the PMF model of concentration variations of reactive species over long-period was considered sufficiently adapted to the objectives of this study.

Moreover, as pinpointed by referee #1, these species corresponded to a large part of the factor 2 contributions (short-lived anthropogenic sources). Consequently, some precisions were brought into the revised Sect 3.5.2 ("Short-lived anthropogenic sources (factor 2)") to better introduce the PMF model limitations to the reader. Correction applied in the revised manuscript in Sect. 3.5.2 (Page 15 lines 21-27):

"Factor 2 is hence attributed to the grouping of several short-lived anthropogenic sources, partly related to gasoline combustion and/or evaporation and solvent use. Note that the PMF model did not reconstruct well 5 of the species composing this factor (propene, n-pentane, n-hexane, toluene, and $C_8$ aromatics – Sect. S2 in the Supplement), especially their concentration variations over short periods. As a result, factor 2 contributions over short periods may be underestimated. However, given the objectives of this study, the examination of factor 2 contribution variations will be limited to seasonal and interannual ones (Sect. 4). Factor 2 contribution variations over a short period was previously investigated in Michoud et al. (2017)."

Finally, as recommended by both referees, the Sect. 3.5.6 ("Towards the best experimental strategy to characterize variation in VOC concentrations observed at a remote background site") was fully revised to better highlight the relevance of the PMF solution examined in this study as well as its limitations. The revised Sect. 3.5.6 can be found in the response to referee #2 comment 49.


Our choice of the VOCs included in this study was based on two reasons: (i) we had planned to measure VOCs using off-line techniques since we could not be frequently present at the Ersa station (which would have been necessary to manage on-line measurements). (ii) We chose to include only VOC measurements realized with canisters in the PMF analysis to maximize the number of data observations.

We would like to inform referee #1 that the selection of the 35 VOCs selected in this study is now presented in the revised Supplement in Sect. S1 ("VOCs selected in this study"; see response to referee #2 comment 19). They comprised 17 NMHCs measured by steel canisters, 4 additional NMHCs measured by multi-sorbent cartridges, 4 carbonyl compounds measured using multi-sorbent cartridges and 10 additional carbonyl compounds measured using DNPH cartridges.

Among the 169 days of the 2-yr period when VOCs were measured at least by one of the three measurement techniques (Fig. S1 – response to referee #1 comment 3), DNPH and multi-sorbent cartridges were sampled only 73 and 52 days concurrently to the collection of steel canisters. Sampling days common to the three off-line techniques were only 36 days. As a result, the reconstruction of missing data points of each dataset would be arduous and would significantly affect the final dataset quality. Additionally, the restriction of the number of data points to those common to the three datasets would significantly impact the temporal representativity of the NMHC inputs of the study period and hence limiting the discussion on interannual and seasonal variations for statistical

robustness reasons. Given these reasons, we made the choice to limit the factorial analysis to the 17 NMHCs measured by steel canisters in this study. These pieces of information have been incorporated in the revised Sect. S2 (see Sect. S2.2 in the response to referee #1 comment 7).

We are aware that the VOC number (17) considered in the PMF analysis discussed in this study is limited compared to other studies (like those indicated by referee #1). The choice of the VOC range in the PMF analysis and its consequences were better discussed in the revised version (see responses to referee #1 comments 7 and 9 and referee #2 comment 49). Moreover, we stressed that our field campaign has taken place over more than two years whereas those of Yuan et al. (2012) and Abeleira et al. (2017) were much shorter (2 months and 16 weeks, respectively). Given the longer study period of the ChArMEx observation field campaign, we adapted the measurement strategy and only VOC off-line measurements were planned. The automatic analysers have been planned only for the purpose of the ChArMEx intensive field campaign of summer 2013, as reported by Michoud et al. (2017). Factorial analyses covering more than a year of VOC measurements at remote sites remain scarce in Europe (such as Sauvage et al. 2009 and Lo Vullo et al. 2016) and they are of high interest to better understand VOC long-term variations (as introduced in Sect. 1 of the manuscript; response to referee #1 comment 1). In this study, we pointed out PMF-derived factors controlling VOC concentration variations at remote sites may be controlled by the meteorological conditions that have occurred during the study period, when issued from short observation periods (i.e. up to two months). Moreover, from our experience gained with this field campaign, we provided some advises in the revised Sect. 3.5.6 ("Towards the best experimental strategy to characterize variation in VOC concentrations observed at a remote background site" – see response to referee #2 comment 49) and in the conclusion (see response to referee #2 comment 73) on the adapted strategy for long-term VOC observations at remote sites (in terms of the VOC number as well as the time resolution and the temporal coverage of VOC measurements).

Then, to check the relevance of the 17 NMHCs selected as inputs in the PMF analysis presented in this study, we benefited from the PMF analysis conducted with the summer 2013 VOC dataset. The PMF solution selected by Michoud et al. (2017) was obtained considering a larger number of VOCs (42), which is more comparable to VOC datasets of Abeleira et al. (2017) and Yuan et al. (2012). Comparisons in Sect. S5 ("Comparisons of VOC source apportionment with previous one performed at Ersa") highlighted a consistency between the two VOC source apportionments performed at Ersa. Furthermore, we checked a 4-factor PMF realized with the summer 2013 dataset restricted to the 17 NMHCs considered in this study. We selected a 4-factor PMF solution as 4 primary sources were identified in Michoud et al. (2017). The comparison results have been added in Sect. S6 of the revised Supplement, as following:

**"Section S6: Examination of a summer 2013 PMF solution realized considering the 17 NMHCs selected in this study**

To check the relevance of the 17 NMHCs selected as inputs in the PMF analysis presented in this study, we benefited from the PMF analysis previously conducted with the larger summer 2013 VOC dataset. The PMF solution selected by Michoud et al. (2017) was realized considering a larger number of VOCs (42). In this section, we selected a summer 2013 PMF solution composed of 4 factors, as 4 primary sources were identified in Michoud et al. (2017), and considering a VOC dataset of 13 variables (those selected for the 2-yr PMF solution, at the exception of propene which was not measured in summer 2013 at a 90-min time resolution). The two PMF solution comparison results are presented in Figs. S16 and S17. The same species dominantly composed the paired factors of the two PMF solutions (Fig. S16) suggesting that the 13 variables selected in this study comprised dominant tracers of the primary sources influencing VOC concentrations observed at Ersa in summer 2013. The primary biogenic source of the PMF solution with the VOC subset (factor 4 – Fig. S16) is composed of a lower proportion of anthropogenic NMHCs and a higher isoprene one. Species composing anthropogenic sources in low proportion tend to have been reduced with the 4-factor PMF solution (factors 1-3 – Fig. S16), suggesting a better deconvolution of the sources, at the exception of ethane proportion in the chemical profile of short-lived anthropogenic sources (factor 3) which increased. Concerning factor contribution variations (Fig. S17), medium-lived anthropogenic sources and the biogenic source showed the same variability in the two PMF solutions (determination coefficients: 0.85-0.89). Similar results were noticed for long-lived anthropogenic sources (determination coefficient: 0.72), at the

exception of the last days of the special observation period. However, short-lived anthropogenic sources have shown different inter-diel variations as a function of the PMF solution, even if factor contribution variations globally followed the same pattern (Fig. S17). This factor contribution variability seems to be not only influenced by the variations in concentrations of reactive selected NMHCs composing it (Fig. S16) but can be by other species such as ethane (for the one of the 4-factor PMF solution), $C_9$ aromatics, 2-methylfuran and OVOCs (carboxylic acids, acetone, isopropanol and n-hexanal; Michoud et al., 2017). Note that formic and acetic acids, and acetone concentrations corresponded to 42% of the total measured concentrations of the VOCs selected for the factorial analysis (Michoud et al., 2017).

[Figure]

Figure S16: Chemical profiles (percent of each species apportioned to each factor - %) of the 4-factor PMF solution (13 variables; blue bars) compared to a selection of VOCs composing chemical profiles of the 4 primary sources identified in Michoud et al. (2017) owing to a 6-factor PMF solution (42 VOC dataset; red bars).

[Figure]

**Figure S17: Times series (on the left) and scatter plots (on the right) of contributions (in ppt) of factors composing the 4-factor PMF solution (13 variables; blue lines) compared to the 4 primary sources identified in Michoud et al. (2017) owing to a 6-factor PMF solution (42 VOC dataset; red lines)."**

Furthermore, preliminary tests of exploring PMF solutions composed from 2 to 10 factors using an input dataset gathering VOC measurements using steel canisters and DNPH cartridges and composed of 73 observations and 23 variables were realized. The same methodology as presented in Sect. S2 (see response to referee #1 comment 7) was followed. An 8-factor PMF solution considering a Fpeak of 0.5 was further examined (see factor profiles and seasonal contributions in Fig. A). The reconstruction of the 14 variables related to HCNMs (Table B) is quite similar to the one with the steel canister dataset (Sect. S2; response to referee #1 comment 7) and the five same species were not well reconstructed (propene, n-pentane, n-hexane, toluene and EX). The reconstruction of ethylene concentrations by the selected 8-factor PMF solution is better compared to the one with the 5-factor PMF solution of 14 variables (Sect. S2). Two carbonyl compounds were not well reconstructed (glyoxal and benzaldehyde – Table B) by the 8-factor PMF solution of 23 variables. Moreover, the solution is composed of three factors mostly composed of OVOCs (cumulative relative contribution: 50%), 4 factors related to anthropogenic sources (cumulative relative contribution: 44%) and a factor related to the local biogenic source (relative contribution to the total variable of 6%). From a PMF solution of 8 factors, a factor was clearly related to the local biogenic source, notably improving the reconstruction of isoprene concentrations. Despite the different number of data observations, the 8-factor PMF

solution identified the same 5 primary sources as those presented in Sect. 3.5 ("Major NMHC sources"). A lower proportion of n-hexane and propene concentrations were attributed to the short-lived anthropogenic factor of the 8-factor PMF solution (Fig. A) compared to the one of the 5-factor PMF solution (see the revised Fig. 7 in response to referee #1 comment 18). OVOC concentrations were mostly apportioned to the three additional factors (named OVOC sources 1-3 which explained 88% of the total OVOC concentrations – Fig. A below) and could be considered mostly of secondary origins. Different interannual and seasonal contributions were observed for these three OVOC sources. The factor named 'OVOC sources 1' apportioned the larger part of OVOC concentrations (42%) mostly due to acetone contribution. This factor shows an increasing trend in the cold season as anthropogenic sources. Higher contributions of 'OVOC sources 1' and short-lived anthropogenic sources were noticed in fall 2013. Given the fact that the factor entitled 'OVOC sources 1' is both composed of short-to-long-lived OVOCs and given their interannual and seasonal variations, the sources related to this factor could be both of local and regional anthropogenic origins. Moreover, higher contributions of the factor 'OVOC sources 2' were noticed in summer 2012, consistently with a secondary biogenic origin (mostly related to the local biogenic source). Higher contributions of the factor 'OVOC sources 3' were noticed in summer 2013, consistently with secondary biogenic origins (both related to the local biogenic source explaining isoprene concentrations but also monoterpenes such as camphene and a-terpinene, which have shown higher concentrations in summer 2013 – see the revised Fig. 4; response to referee #2 comment 22). Links with anthropogenic sources are not discarded for 'OVOC sources 2 and 3' but they cannot be examined in this study. Moreover, concentrations of methylglyoxal, i,n-butanals, acetaldehyde and acetone were partly attributed to the local biogenic source (explained 6-32% of these OVOC concentrations, which represent 7% of the total OVOC concentrations), consistently with their seasonal variations (see response to referee #1 comment 16). Primary anthropogenic origins for MEK and glyoxal were supposed considering their seasonal concentration variations (see response to referee #1 comment 16) and were especially attributed to evaporative sources (explained 0-20% of these OVOC concentrations, which represent 5% of the total OVOC concentrations). A lower proportion of OVOC concentrations were apportioned to primary sources compared to Michoud et al. (2017), which could be linked to the measured OVOCs in the two studies, the study periods and the number and the resolution of the VOC measurements.

[Figure]

[Figure]

**Figure A: Factor chemical profiles (percent of each species apportioned to each factor - %; left) and seasonal contributions (µg.m⁻³; right) of the 8-factor PMF solution (23 variables). Factors are ordered as a function of their relative contribution to the total VOC mass (percentages into parentheses).**

5    **Table B: Evaluation of reconstructed VOC concentrations of the 8-factor PMF solution as a function of measured VOCs.**

| VOC | R² | Slope (µg.m⁻³) | Intercept (µg.m⁻³) |
|---|---|---|---|
| (0) TVOC | 0.993 | 1.003 | -0.013 |
| (1) Ethane | 0.996 | 1.002 | -0.005 |
| (2) Ethylene | 0.833 | 0.776 | 0.050 |
| (3) Propane | 0.967 | 0.974 | 0.011 |
| (4) Propene | 0.514 | 0.570 | 0.020 |
| (5) i-Butane | 0.929 | 0.847 | 0.031 |
| (6) n-Butane | 0.966 | 0.980 | 0.002 |
| (7) Acetylene | 0.995 | 0.979 | 0.004 |
| (8) i-Pentane | 0.764 | 0.723 | 0.058 |
| (9) n-Pentane | 0.294 | 0.416 | 0.101 |
| (10) Isoprene | 0.972 | 0.869 | 0.009 |
| (11) n-Hexane | 0.568 | 0.553 | 0.030 |
| (12) Benzene | 0.920 | 0.869 | 0.030 |
| (13) Toluene | 0.653 | 0.662 | 0.0 |
| (14) EX | 0.590 | 0.649 | 0.081 |
| (15) Formaldehyde | 0.999 | 1.007 | -0.009 |
| (16) Acetaldehyde | 0.778 | 0.872 | 0.084 |

| | | | |
|---|---|---|---|
| (17) Acetone | 0.995 | 0.973 | 0.092 |
| (18) MEK | 0.985 | 0.984 | 0.005 |
| (19) i,n-Butanals | 0.667 | 0.470 | 0.078 |
| (20) Benzaldehyde | 0.260 | 0.268 | 0.080 |
| (21) Glyoxal | 0.542 | 0.515 | 0.022 |
| (22) Methylglyoxal | 0.686 | 0.683 | 0.042 |
| (23) n-Hexanal | 0.752 | 0.471 | 0.052 |


Some corrections were applied to the revised Sect. 2.4.2 ("Identification of potential emission areas"; Page 8, lines 15-25) to better introduce the weighing function use and its non-consideration in the CF analyses discussed in this study:

"A better statistical significance of the CF results is commonly considered for grid cells with a higher number of crossing trajectory points. As a result, some studies applied an empirical weighing function so as to limit the possible influence of high concentrations which may be observed during occasional episodes with uncommon trajectories (e.g., Bressi et al., 2014; Waked et al., 2014, 2018) and hence could influence cells having a low number of trajectory points. We preliminary tried to apply this weighing function in this study. Exploratory tests revealed that CF results with the empirical weighing function only highlighted local contributions, given the number of air masses considered in this study. The farther a cell is from the Ersa station, the lower its corresponding $n_{ij}$ value (number of points of the total number of back-trajectories

contained in the ij[th] grid cell, Sect. S3 of the Supplement), and more the weighing function tended toward downweighting the low $n_{ij}$ value. Therefore, CF results discussed in this study were realized without weighing and these limitations should be taken into account when examining CF analyses, which are hence considered as indicative information."


**14/Sect. 3.2, Page 11**

Sect. 3.2 ("Air mass origins") and table 3: it is not clear why trajectories categorized as long have median transit time always shorter than the trajectories categorized as short.

An air mass trajectory classified as short has closer distance between two of its succeeding points compared to another one classified as long (see examples in Fig. C). Due to the location of Ersa in the Mediterranean Sea, the air mass with the shorter trajectory has spent more time to reach the Ersa site from French coastlines compared to an air mass trajectory classified as long (11h and 6h, respectively). This is illustrated by Figure C below.

[Figure]

**Figure C: Comparison of transit times between a short and long air mass trajectories both classified into cluster 2 (France).**

This was clarified in the revised Sect. 3.2 ("Air mass origins"; Page 10, lines 14-18):

"In particular, European and French air masses showed lower transit times over the sea (median values of 6 h and 8 h, respectively; Table 2) when their trajectories are categorized as long; compared to short ones (23 h and 19 h, respectively). These findings are based on the fact that an air mass trajectory classified as short has closer distance between two succeeding trajectory points compared to another one classified as long. Due to the Ersa location in the Mediterranean Sea, the air masses having trajectories characterized as long have spent more time to reach the Ersa site."

**15/Sect. 3.3 and 3.4.3**

Some of the VOCs listed (6 C6 - C11 n-aldehydes) are not discussed at all afterwards, why?

Considering referee #1's remark on the 6 n-aldehydes collected on multi-sorbent cartridges, they were incorporated to the revised Table 3, as well as additional carbonyl compounds sampled with DNPH cartridges, since OVOC measurements remain rather scarce. They were not discussed in the original manuscript since they were not incorporated in the PMF analysis, and we thought this would limit our examination of OVOC sources and origins.

We would like to inform referee #1 that the Sect. S1 ("VOCs selected in this study") was added in the revised Supplement in response to referee #2 comment 19. As a consequence, four additional carbonyl compounds sampled with multi-sorbent cartridges (n-octanal, n-nonanal, n-decanal, n-undecanal) and six ones sampled using DNPH cartridges (i,n-butanals, n-hexanal, benzaldehyde, glyoxal and methylglyoxal) were added to the 25 initial VOCs investigated in this study. Corrections applied to Table 3:

"Table 3: Statistics ($\mu$g.m$^{-3}$), standard deviations ($\sigma$ - $\mu$g m$^{-3}$), detection limits (DL - $\mu$g m$^{-3}$) and relative uncertainties U(X)/X (Unc. - %) of selected VOC concentrations measured at the site from June 2012 to June 2014.

| | Species | Min | 25 % | 50 % | Mean | 75 % | Max | $\sigma$ | DL | Unc. |
|---|---|---|---|---|---|---|---|---|---|---|
| BVOCs | Isoprene | 0.01 | 0.01 | 0.04 | 0.16 | 0.16 | 2.28 | 0.31 | 0.03 | 32 |
| | α-Pinene | <0.01 | 0.03 | 0.10 | 0.38 | 0.57 | 3.61 | 0.61 | 0.01 | 40 |
| | Camphene | <0.01 | 0.01 | 0.05 | 0.12 | 0.13 | 0.78 | 0.17 | 0.01 | 73 |
| | α-Terpinene | <0.01 | <0.01 | <0.01 | 0.06 | 0.05 | 0.88 | 0.15 | 0.01 | 47 |
| | Limonene | <0.01 | <0.01 | 0.03 | 0.19 | 0.36 | 1.73 | 0.30 | 0.01 | 45 |
| Anthropogenic NMHCs | Ethane | 0.57 | 1.13 | 1.85 | 1.86 | 2.46 | 4.28 | 0.81 | 0.01 | 7 |
| | Propane | 0.18 | 0.44 | 0.77 | 0.94 | 1.41 | 2.60 | 0.61 | 0.02 | 11 |
| | i-Butane | 0.01 | 0.09 | 0.17 | 0.24 | 0.35 | 1.02 | 0.19 | 0.02 | 22 |
| | n-Butane | 0.05 | 0.16 | 0.26 | 0.37 | 0.57 | 1.09 | 0.26 | 0.02 | 13 |
| | i-Pentane | 0.06 | 0.15 | 0.22 | 0.25 | 0.31 | 0.90 | 0.14 | 0.03 | 25 |
| | n-Pentane | 0.02 | 0.09 | 0.18 | 0.20 | 0.27 | 0.80 | 0.13 | 0.03 | 33 |
| | n-Hexane | 0.02 | 0.04 | 0.07 | 0.08 | 0.10 | 0.27 | 0.05 | 0.04 | 43 |
| | Ethylene | 0.09 | 0.19 | 0.28 | 0.32 | 0.39 | 0.87 | 0.17 | 0.01 | 14 |
| | Propene | 0.01 | 0.04 | 0.06 | 0.07 | 0.09 | 0.17 | 0.03 | 0.02 | 40 |
| | Acetylene | 0.03 | 0.09 | 0.18 | 0.26 | 0.36 | 1.23 | 0.23 | 0.01 | 12 |
| | Benzene | 0.07 | 0.16 | 0.26 | 0.31 | 0.39 | 1.11 | 0.19 | 0.03 | 25 |
| | Toluene | 0.04 | 0.15 | 0.23 | 0.28 | 0.34 | 0.84 | 0.17 | 0.04 | 26 |
| | Ethylbenzene | 0.02 | 0.02 | 0.02 | 0.04 | 0.05 | 0.15 | 0.03 | 0.04 | 50 |
| | m,p-Xylenes | 0.02 | 0.07 | 0.10 | 0.12 | 0.14 | 0.41 | 0.08 | 0.04 | 45 |
| | o-Xylene | 0.02 | 0.02 | 0.06 | 0.07 | 0.10 | 0.32 | 0.06 | 0.04 | 44 |
| OVOCs | Formaldehyde | 0.28 | 0.68 | 1.17 | 1.53 | 1.89 | 6.30 | 1.24 | 0.03 | 7 |
| | Acetaldehyde | 0.40 | 0.67 | 0.83 | 0.96 | 1.23 | 2.87 | 0.41 | 0.03 | 22 |
| | i,n-Butanals | <0.01 | 0.10 | 0.15 | 0.26 | 0.23 | 5.15 | 0.56 | 0.03 | 20 |
| | n-Hexanal | <0.01 | 0.08 | 0.13 | 0.22 | 0.24 | 1.83 | 0.27 | 0.03 | 12 |
| | Benzaldehyde | <0.01 | 0.06 | 0.13 | 0.15 | 0.22 | 0.60 | 0.12 | 0.04 | 21 |

| | | | | | | | | | |
|---|---|---|---|---|---|---|---|---|---|
| n-Octanal | <0.01 | 0.01 | 0.05 | 0.05 | 0.11 | 1.25 | 0.20 | 0.01 | 39 |
| n-Nonanal | <0.01 | 0.07 | 0.21 | 0.21 | 0.37 | 1.42 | 0.31 | 0.01 | 33 |
| n-Decanal | <0.01 | 0.04 | 0.16 | 0.16 | 0.31 | 1.19 | 0.26 | 0.01 | 33 |
| n-Undecanal | <0.01 | 0.04 | 0.05 | 0.05 | 0.08 | 0.33 | 0.06 | 0.01 | 39 |
| Glyoxal | <0.01 | 0.04 | 0.06 | 0.07 | 0.11 | 0.25 | 0.05 | 0.02 | 27 |
| Methylglyoxal | <0.01 | 0.07 | 0.11 | 0.16 | 0.19 | 0.95 | 0.15 | 0.04 | 23 |
| Acetone | 1.50 | 2.46 | 3.57 | 4.31 | 4.98 | 16.49 | 2.64 | 0.03 | 6 |
| MEK | 0.18 | 0.27 | 0.33 | 0.36 | 0.45 | 0.90 | 0.14 | 0.03 | 10 |

"

Some corrections were consequently applied to Sect. 3.3 ("VOC mixing ratios"; from Page 10, Line 23 to Page 11 Line 10):

"Statistical results on concentrations of 35 VOCs selected in this study (see Sect. S1 in the Supplement) are summarized in Table 3. Their average concentration levels as a function of the measurement sampling times (09:00-13:00 or 12:00-16:00) are indicated in Table S1. These VOCs were organized into three principal categories: biogenic, anthropogenic and oxygenated VOCs (5, 16 and 14 targeted species, respectively; Table 3). Isoprene and four monoterpenes were classified into BVOCs, while primary hydrocarbons (alkanes, alkenes, alkynes and aromatic compounds) were included into anthropogenic NMHCs, since their emissions are especially in connection with human activities. OVOCs have been presented separately, as these compounds come from both biogenic and anthropogenic (primary and secondary) sources. OVOCs were the most abundant, accounting for 65% of the total concentration of the 35 compounds selected in this study. They were mainly composed of acetone (contribution of 51% to the OVOC cumulated concentration). Anthropogenic NMHCs also contributed significantly (26%) to the total concentration of the 35 measured VOCs and principally consisted of ethane and propane (which represented 34 and 17% of the anthropogenic NMHC mass, respectively) as well as n-butane (7%). The high contribution of species with generally the longest lifetime in the atmosphere (see Sect. 3.4) is consistent with the remote location of the Ersa site and in agreement with Michoud et al. (2017). BVOCs only contributed little to the total VOC concentration on annual average (4%), reaching 13% in summer. They were mainly composed of isoprene and α-pinene (contribution of 44 and 32% to the BVOC mass, respectively). These compounds are among the major BVOCs in terms of emission intensity for the Mediterranean vegetation (Owen et al., 2001) and accounted for half of isoprenoid concentrations recorded during the intensive field campaign conducted in summer 2013 at Ersa (Debevec et al., 2018; Kalogridis, 2014). On the contrary, a larger α-terpinene contribution was noticed during the summer intensive campaign than the 2-yr observation period. Note that speciated monoterpenes were measured differently during the summer 2013 campaign, by means of an automatic analyser (see Sect. S4 in the Supplement)."

We added an additional table in the Sect. 3.4 ("VOC variability") of the revised manuscript which also includes seasonal average OVOC concentrations:

"Table 4: Seasonal average VOC concentrations (± 1 σ; µg m$^{-3}$)

| | Species | Winter | Spring | Summer | Fall |
|---|---|---|---|---|---|
| BVOCs | Isoprene | 0.1 ± 0.1 | 0.2 ± 0.5 | 0.3 ± 0.3 | 0.1 ± 0.1 |
| | α-Pinene | 0.1 ± 0.1 | 0.3 ± 0.9 | 0.7 ± 0.5 | 0.5 ± 0.5 |
| | Camphene | 0.1 ± 0.1 | 0.1 ± 0.1 | 0.1 ± 0.1 | 0.1 ± 0.1 |
| | α-Terpinene | 0.1 ± 0.1 | 0.1 ± 0.1 | 0.3 ± 0.3 | 0.1 ± 0.1 |
| | Limonene | 0.1 ± 0.1 | 0.1 ± 0.4 | 0.4 ± 0.2 | 0.3 ± 0.3 |
| | | | | | |
| Anthropogenic NMHCs | Ethane | 2.9 ± 0.5 | 1.8 ± 0.6 | 1.0 ± 0.2 | 1.9 ± 0.5 |
| | Propane | 1.7 ± 0.4 | 0.6 ± 0.2 | 0.4 ± 0.2 | 1.2 ± 0.5 |
| | i-Butane | 0.4 ± 0.1 | 0.1 ± 0.1 | 0.1 ± 0.1 | 0.4 ± 0.2 |
| | n-Butane | 0.7 ± 0.2 | 0.2 ± 0.1 | 0.2 ± 0.1 | 0.5 ± 0.2 |
| | i-Pentane | 0.3 ± 0.1 | 0.2 ± 0.1 | 0.2 ± 0.1 | 0.3 ± 0.1 |
| | n-Pentane | 0.2 ± 0.1 | 0.2 ± 0.2 | 0.2 ± 0.1 | 0.3 ± 0.1 |
| | n-Hexane | 0.1 ± 0.1 | 0.1 ± 0.1 | 0.1 ± 0.1 | 0.1 ± 0.1 |
| | Ethylene | 0.5 ± 0.2 | 0.2 ± 0.1 | 0.2 ± 0.1 | 0.4 ± 0.5 |
| | Propene | 0.1 ± 0.1 | 0.1 ± 0.1 | 0.1 ± 0.1 | 0.1 ± 0.1 |
| | Acetylene | 0.5 ± 0.3 | 0.2 ± 0.1 | 0.1 ± 0.1 | 0.3 ± 0.1 |
| | Benzene | 0.5 ± 0.2 | 0.2 ± 0.1 | 0.2 ± 0.1 | 0.4 ± 0.1 |
| | Toluene | 0.3 ± 0.2 | 0.2 ± 0.1 | 0.2 ± 0.1 | 0.3 ± 0.2 |

| | | | | | |
|---|---|---|---|---|---|
| | C8-aromatics | 0.2 ± 0.2 | 0.2 ± 0.2 | 0.2 ± 0.1 | 0.2 ± 0.2 |
| OVOCs | Formaldehyde | 0.8 ± 0.5 | 1.3 ± 0.8 | 2.3 ± 1.3 | 1.1 ± 0.4 |
| | Acetaldehyde | 0.8 ± 0.3 | 0.8 ± 0.3 | 1.3 ± 0.4 | 0.8 ± 0.3 |
| | i,n-Butanals | 0.1 ± 0.1 | 0.1 ± 0.1 | 0.5 ± 1.0 | 0.1 ± 0.1 |
| | n-Hexanal | 0.1 ± 0.1 | 0.2 ± 0.1 | 0.4 ± 0.4 | 0.2 ± 0.1 |
| | Benzaldehyde | 0.2 ± 0.1 | 0.1 ± 0.2 | 0.2 ± 0.1 | 0.1 ± 0.1 |
| | n-Octanal | 0.1 ± 0.1 | 0.1 ± 0.1 | 0.2 ± 0.4 | 0.1 ± 0.1 |
| | n-Nonanal | 0.3 ± 0.4 | 0.4 ± 0.4 | 0.1 ± 0.2 | 0.3 ± 0.2 |
| | n-Decanal | 0.3 ± 0.3 | 0.3 ± 0.3 | 0.1 ± 0.1 | 0.3 ± 0.2 |
| | n-Undecanal | 0.1 ± 0.1 | 0.1 ± 0.1 | 0.1 ± 0.1 | 0.1 ± 0.1 |
| | Glyoxal | 0.1 ± 0.1 | 0.1 ± 0.1 | 0.1 ± 0.1 | 0.1 ± 0.1 |
| | Methylglyoxal | 0.1 ± 0.1 | 0.2 ± 0.2 | 0.3 ± 0.2 | 0.1 ± 0.1 |
| | Acetone | 2.7 ± 1.2 | 3.8 ± 1.4 | 5.8 ± 1.8 | 3.7 ± 1.8 |
| | MEK | 0.4 ± 0.1 | 0.3 ± 0.1 | 0.4 ± 0.2 | 0.4 ± 0.1 |

"

Some of these additional OVOCs have also been added in the revised Fig. 6:

[Figure]

[Figure]

**Figure 6: (a) Monthly variations in a selection of oxygenated VOC concentrations (expressed in µg m$^{-3}$) represented by box plots; the blue solid line, the red marker, and the box represent the median, the mean, and the interquartile range of the values, respectively.** The bottom and top of the box depict the first and third quartiles (i.e. Q1 and Q3) and the ends of the whiskers correspond to the first and ninth deciles (i.e. D1 and D9). **(b) Their monthly average concentrations as a function of the year; full markers indicate months when VOC samples were collected from 12:00-16:00 and empty markers those when VOC samples were collected from 09:00-13:00."**

As a consequence, interpretations of OVOC concentration variations have been developed in the revised manuscript (in Sect. 3.4.3) following referee #1 suggestions (see response to referee #1 comment 16).

**16/Sect. 3.4.3, Page 14**

This section is hard to follow, line 14-26 are only general considerations with no direct link to the observations. Why not starting with the trends observed (end of line 26, "Formaldehyde and acetaldehyde concentrations..." and use some of the general information to support the discussion. Same comment for acetone and MEK.

We thank referee #1 for this feedback. As a consequence, the Sect. 3.4.3 ("Oxygenated VOCs") has been fully rewritten in the revised manuscript. More OVOCs were now considered in the revised Sect. 3.4.3 (in response to referee #1 comment 15) and OVOCs having similar variations were now presented together (in response to referee #2 general comment). Their seasonal variations were firstly presented and then selected indications concerning their potential sources are used to support the discussion. Interannual variations are used to confirm some statements or highlight additional contributions. Corrections applied in the revised manuscript in Sect. 3.4.3 (from Page 12 line 30 to Page 14 Line 8):

[revised manuscript text omitted]

**17/Sect. 3.4.3, Page 15, line 20**

The term "regional" is rather vague, can you indicate which geographical areas are included?

Some precisions were brought in this sentence (Page 13, lines 21-24 in the revised manuscript):

"Acetone can also be resulted of the oxidation of various VOCs (Goldstein and Schade, 2000; Jacob et al., 2002; Singh et al., 2004) and roughly half of its concentrations measured at diverse urban or rural sites have been assigned to regional background pollution by several studies (e.g., Debevec et al., 2017; de Gouw et al., 2005; Legreid et al., 2007) with regional contributions at a scale of hundreds of kilometres."

**18/Sect. 3.5.1, Page 17, line 1**

Biogenic source: "Local biogenic source" instead?

This factor was renamed in Figs. 7, 8, 10 and S3 in the revised manuscript and supplement as proposed by referee #1.

Note that following referee #2 general remark, we decided to move discussions on factor contribution variations into Sect. 4 ("Discussions on the seasonal variability of VOC concentrations") in the revised manuscript, inducing a change in figure numbers and a move of some figures to the Supplement (Fig. S3). Moreover, we decided to incorporate OVOC seasonal concentration levels in Fig. 10 following referee #1 remark 20. Revised Figs. 7, 8, 10 and S3:

"

[Figure]

Figure 7: Chemical profiles of the 5-factor PMF solution (14 variables). Factor contributions to each species (µg m⁻³) and the percent of each species apportioned to the factor are displayed as a grey bar and a color circle, respectively. Factor 1 - local biogenic source; factor 2 - short-lived anthropogenic sources; factor 3 – evaporative sources; factor 4 – long-lived combustion sources; factor 5 – regional background.

[Figure]

**Figure 8: (a)** Time series of NHMC factor contributions (µg m⁻³) and **(b)** accumulated relative NMHC contributions. Factor 1 - local biogenic source; factor 2 - short-lived anthropogenic sources; factor 3 – evaporative sources; factor 4 – long-lived combustion sources; factor 5 – regional background. Note that the NMHC dataset used for the PMF analysis included different sampling time hours (09:00-13:00 or 12:00-16:00) following shifts during the two-year period (see Table 1).

[Figure]

**Figure 10: Variations in seasonal averaged accumulated concentrations (expressed in µg m⁻³) of the 35 VOCs selected in this study. The 17 NHMCs selected for the factorial analysis were apportioned to the five modelled NMHC sources. NMHC seasonal measured concentrations which were not modelled by the PMF tool were lower than 0.09 µg m⁻³ and are not reported here. Winter: 01/01-31/03 periods – spring: 01/04-30/06 periods – summer: 01/07-30/09 periods – fall: 01/10-31/12 periods. Note that the VOC dataset included different sampling time hours (09:00-13:00 or 12:00-16:00) following shifts during the two-year period (see Table 1).**

[Figure]

**Figure S3: Seasonal (a) and interannual (b) variations in NMHC factor contributions (expressed in µg m⁻³) represented by box plots; the blue solid line, the red marker, and the box represent the median, the mean, and the interquartile range of the values, respectively. The bottom and top of the box depict the first and the third quartiles (i.e. Q1 and Q3) and the ends of the whiskers correspond to the first and the ninth deciles (i.e. D1 and D9). NMHC factors: factor 1 - local biogenic source; factor 2 - short-lived anthropogenic sources; factor 3 – evaporative sources; factor 4 – long-lived combustion sources; factor 5 – regional background. Winter: 01/01-31/03 periods – spring: 01/04-30/06 periods – summer: 01/07-30/09 periods – fall: 01/10-31/12 periods.**

Note that the NMHC dataset used for the PMF analysis included different sampling time hours (09:00-13:00 or 12:00-16:00) following shifts during the two-year period (see Table 1)."

**19/Sect. 3.5.1, Page 17, line 6**

"troposphere is" instead of "troposphere was"?

Past tense is used as we have reported lifetimes of isoprene measured during the 2-yr period, even if isoprene lifetime is generally low. We modified the sentence to avoid any confusion (Page 15, lines 3-5 of the revised manuscript):

"The estimated tropospheric lifetime of isoprene was quite short (winter: 5.6 h and summer: 1.1 h), indicating that this compound was emitted mostly by local vegetation."

**20/Sect. 3.5.1, Page 17, line 1**

"to the sum of measured VOC concentrations": do you include OVOC in this calculation? Anyway, because the list of the VOCs included in the PMF is not exhaustive, the average individual contribution of each factors to the sum of the measured VOCs should be considered with care.

We did not include OVOCs in the calculation as the revised Sect. 3.5 ("Major NMHC sources") presented NMHC sources only. However, we insisted more in the revised manuscript on the fact that the relative factor contributions are relative to the selected NMHCs for the PMF analysis.
        As a consequence some corrections were applied to the revised Sect. 3.5 (Page 14, lines 10-15):

"In the coming section, major NMHC primary sources which have impacted primary NMHC concentrations measured at Ersa were identified using a PMF 5-factor solution (from simulations presented in Sect. 2.3) and a dataset composed of 14 variables (selected NMHCs measured by steel canister, see Sect. S2). Figure 7 depicts factor contributions to the species chosen as inputs for the PMF tool along with NMHC contributions to the 5 factors defined by the factorial analysis. Figure 8 and Table 5 show PMF factor contribution time series and their relative contributions to the total concentrations of the selected NMHCs in this factorial analysis, respectively."

        Precisions were brought to the revised Sects. 3.5.1-3.5.5 to better specify concerned VOCs in the calculation. For instance "the sum of measured VOCs" was replaced by "the sum of selected measured NMHCs" or "the selected measured NMHCs" in the revised manuscript.

**21/Sect. 3.6, Page 20, line 27**

Page 20 "Towards the best experimental strategy to characterize variation in VOC concentrations" Larges parts of this section are copy/paste of the section S4 of the SI (page 20, line 28to page 21, line 2; similar to page 10 of the supporting information, lines 1-6 ; page 21,lines 8-19 similar to page 12 of the SI, lines 22-34).

Following referee #1's remark, information in the revised Sect. 3.5.6 ("Towards the best experimental strategy to characterize variation in VOC concentrations observed at a remote background site") have been clearly distinguished from those indicated in the Sect. S5 ("Comparisons of VOC source apportionment with previous one performed at Ersa"). The Sect. 3.5.6 now focuses on the relevance of the PMF solution considering the limited range of VOCs examined in this study (see response to referee #2 comment 49). The Sect. S5 has been limited to comparative results. As a result, the information initially repeated in the Sect. S5 was removed in the revised Supplement.

**22/Sect. 4.1 Page 21, line 25**

"are" instead of "were".

As proposed by referee #1, the tense was changed in the sentence (Page 20, lines 4-5):

"Firstly, the 21 NMHCs have shown low concentrations during summer and spring periods (average seasonal accumulated concentration of 4.6 ±0.1 μg m$^{-3}$; Fig. 10) [...]."

**23/ Sect. 4.1, Page 21, line 25**

Page 21 line 25 starting at "Note that.." to line 28 : I suggest to remove this information, it is not essential and distracts the reader from the topic of the section.

The precision on the non-taken into account of VOC concentrations during spring 2012 in Fig. 10 was removed in the revised manuscript.

**24/Sect. 4.1, Page 22, lines 6-8**

The authors attribute the high contribution of factor 2 (related to short lived species) in spring and summer to relatively nearby sources. Have the authors checked if the correlation with CO was improved in these specific conditions?

The correlation of factor 2 contributions with CO concentrations was not improved considering only spring and summer observations (Pearson coefficients: -0.05 during spring and summer and 0.22 during the 2-yr period). Note that CO measurements were only performed in spring and summer 2013 (see Fig. S1; response to referee #1 comment 3).

**25/Sect. 4.2, Page 24, line 24**

I don't understand the meaning of the last sentence. Please rephrase "As a consequence, this finding...may be reflected..."

The sentence was removed in the revised Sect. 4.2 ("VOC concentration variations in fall and winter"). However, this information was rewritten in the conclusion for clarity (from Page 26 line 33 to Page 27 line 3) as followed:

"They also pointed out that the PMF-derived factors controlling VOC concentration variations at remote sites may be mainly controlled by the meteorological conditions that occurred during the study period when issued from short observation periods (i.e. up to two months)."

**26/Sect. 5 and Sect. 2.2.4**

Page 7, section 2.2.4 ("Concurrent VOC measurements performed at other..") I would suggest shortening this section merge it with section 5. Both sections start with the same 4 lines.

As proposed by referee #1, we removed in the revised manuscript the section 2.2.4 and merged it in Sect. S7 ("Concurrent NMHC measurements performed at other European background stations"). As a consequence, the Fig. 2 of the initial version of the manuscript was also moved in Sect. S7 (as Fig. S18) and Sect. 5 ("VOC concentration variations in continental Europe") was modified. Correction applied in the revised manuscript in Sect. 5 (Page 23, lines 18-23):

"From June 2012 to June 2014, NMHC measurements were concurrently conducted at 17 other European background monitoring stations (described in Sect. S7), allowing us (i) to examine the representativeness of the Ersa station in terms of seasonal variations in NMHC concentrations impacting continental Europe and (ii) to provide some insights on dominant drivers for VOC concentration variations in Europe built on what we have learned from Ersa's VOC observations. Figure 13 depicts monthly concentration time series of a selection of NMHCs measured at the 18 considered European monitoring stations (including Ersa). "

Section S7 in the Supplement is the following:

"Section S7: Concurrent NMC measurements performed at other European background monitoring stations

From June 2012 to June 2014, NMHC measurements were concurrently conducted at 17 other European background monitoring stations. These European stations are part of EMEP and GAW networks. Figure S18 shows their geographical distribution. They cover a large part of western and central Europe from Corsica Island in the south to northern Scandinavia in the north, are located at different altitudes (up to 3580 m a.s.l.) and most of them are categorized as GAW 'regional stations for Europe'. More information on these stations can be found on EMEP (https://www.nilu.no/projects/ccc/sitedescriptions/index.html, last access: 03/04/2020) or GAW station information system (https://gawsis.meteoswiss.ch/GAWSIS//index.html#/, last access: 03/04/2020) sites. NMHC measurements were realized by different on-line (GC or proton-transfer-reaction mass spectrometer - PTR-MS) or off-line techniques (VOCs collected by steel canisters) and were reported in the EMEP EBAS database (http://ebas.nilu.no/Default.aspx, last access: 03/04/2020).

[Figure]

**Figure S18: Locations of 18 European monitoring stations that included NMHC measurements conducted from June 2012 to June 2014. These stations are part of EMEP/GAW networks. They are characterized by their GAW identification and their altitudes are given within brackets in reference to the standard sea level. AUC, BIR, ERS, KOS, NGL, PYE, RIG, SMR, SSL and TAD are categorized as GAW 'regional stations for Europe'. CMN, HPB, JFJ and PAL are categorized as GAW 'global stations'. AHRL, SMU, WAL and ZGT are considered as GAW 'other elements stations in Europe'. More precisely, ZGT is a 'coastal station' while HRL, SMU and WAL are 'rural stations'. Note that high-altitude stations such as CMN and HPB could be frequently in free-tropospheric conditions. More information on these stations can be found on EMEP (https://www.nilu.no/projects/ccc/sitedescriptions/index.html, last access: 03/04/2020) or GAW station information system (https://gawsis.meteoswiss.ch/GAWSIS//index.html#/, last access: 03/04/2020) sites.**

The Ersa site is underlined in red. Square markers indicate that VOCs were collected by steel canisters and analysed thereafter at laboratories (i.e. off-line measurements). Triangle and diamond markers indicate that VOC measurements were conducted in-situ using PTR-MS or GC systems, respectively. NMap provided by Google Earth Pro software (v.7.3.3 image Landsat/Copernicus – IBCAO; data SIO, NOOA, U.S, Navy, NGA, GEBCO; © Google Earth)."

**27/Sect. 5, Page 25, line 18**

Page 25 line 18 Please rephrase "were also be taken".

As recommended by referee #1, this sentence was modified in the revised manuscript (Page 24, lines 1-3):

"In addition, despite its shorter lifetime compared to other NMHCs of the selection, ethylene concentration variations were also examined in this study to investigate short-lived anthropogenic source importance and variability in continental Europe."

**acp-2020-607: "Seasonal variation and origins of volatile organic compounds observed during two years at a western Mediterranean remote background site (Ersa, Cape Corsica)"**

5    The manuscript presents observations of VOCs at the Ersa site in Cape Corsica over a two-year period and provides a comprehensive description and analysis of their behaviour during this time. PMF analysis of the data is presented along with comparison to other station across Europe. The plots are clear and generally well presented. The length of the manuscript, however, is something of a problem with lots of repetition throughout. I am sure there is an interesting story here, but it is difficult to assess
10   what that is from the current article. Careful consideration of each section, it's findings and their relevance is required in order to make the manuscript worthy of publication. I therefore recommend a major revision of the manuscript. I do however, urge the authors to continue to work on this as I feel it can be a very nice piece. The authors should consider writing the manuscript in terms of the features observed at the site and then use the data and plots to explain that behaviour. In its current format,
15   each compound or group is considered separately and methodically (which leads to a comprehensive, but repetitive narrative) whereas, the behaviour of these different compounds can often be explained by the same phenomena (e.g. a changing boundary layer or temperature difference). I have included suggested changes to individual sections, but these may not be relevant to the newly written article.

**Authors' responses to Referee #2**

20   We would like to thank the Referee #2 for her/his general feedback and each of her/his useful comments/questions for improving the quality of this manuscript. All comments addressed by both referees have been taken into account in the revised version of the manuscript.

As suggested by both referees, the revised manuscript was largely rewritten. The introduction was shortened. Complementary information on the VOCs selected in this study and on the PMF
25   analysis are now provided in the Supplement (Sects. S1 and S2, respectively). Sections on results have been reviewed in order to better separate information provided by them and hence removed repetitive ones. Section 3.1 ("Meteorological conditions") was shortened only keeping essential pieces of information to the explanation of seasonal and interannual VOC variations. Descriptive Sects. 3.3 and 3.4 ("VOC mixing ratios" and "VOC variability", respectively) have been limited to the presentation
30   of VOC concentration levels, their abundance and their variations and elements of interpretations linked to factors controlling them were removed or moved in Sect. 4 ("Discussions on the seasonal variability of VOC concentrations"). Sect. 3.4.1-3.4.3 were rewritten grouping results to emphasize similar or different VOC behaviours. Note that a larger number of OVOCs is now considered in the Sect. 3.4.3 ("Oxygenated VOCs") and comparisons with other VOC measurements performed at Ersa were
35   moved in the revised Supplement (Sect. S4). Section 3.5 ("Major NMHC sources") was limited to the presentation of the 5 NMHC factors identified in this study and results on their seasonal and interannual variations were removed or moved in Sect. 4. The other factorial analysis previously realized with the summer 2013 VOC dataset has been better used to support factor identification in this study. As a result, Sect. 3.5.6 ("Towards the best experimental strategy to characterize variation
40   in VOC concentrations observed at a remote background site") has been reviewed to better highlight (i) the relevance of the PMF solution to identify NMHC sources and (ii) its limitation to examine VOC concentration variations observed at Ersa. The Sect. 3.5.6 is supported by commentary results presented in the Supplement (Sects. S5 and S6). Section 4 has been restructured in order to distinguish factors controlling VOC concentration variations in spring and summer (Sect. 4.1) from those in fall and
45   winter (Sect. 4.2) and OVOC concentration variations have also been incorporated. The description of the 17 European sites, whose NMHC concentration variations are discussed in Sect. 5 ("VOC concentration variations in continental Europe"), was moved to the Supplement (Sect. S7). The conclusion has also been rewritten.

In this respect, several figures were notably modified, including in the supplement. Please note that figures numbers are now different in this new version. Additional sections have been added to the Supplement to present and discuss the selection of the VOCs in this study and the relevance of the PMF solution.

In the present document, authors' answers to the specific comments addressed by Referee #2 are mentioned in **blue**, while changes made to the revised manuscript are shown in **green**. The comments on the manuscript are listed as follows:

**1/Abstract, Page 1, line 17:**
"… The VOC speciation was largely dominated by oxygenated VOCs …"
Should this be the VOC abundance or mass? I'm not sure how speciation can be dominated.

The sentence was modified in the revised manuscript as (page 1, lines 17-18 in the revised manuscript):

"The VOC abundance was largely dominated by oxygenated VOCs (OVOCs) along with primary anthropogenic VOCs having a long lifetime in the atmosphere."

**2/Abstract, Page 1, line 18:**
"VOC temporal variations are then examined…"
Past tense, should be "were examined"

The sentence was modified in the revised manuscript as (page 1, line 18 in the revised manuscript):

"VOC temporal variations were then examined."

**3/Abstract, Page 1, line 19:**
"… and solar radiation ones."
Delete "ones"

We deleted "ones" in the revised manuscript (page 1, lines 18-19 in the revised manuscript):

"Primarily of local origin, biogenic VOCs exhibited notable seasonal and interannual variations, related to temperature and solar radiation."

**4/Abstract, Page 1, line 20:**
"Anthropogenic compounds have shown an increasing concentration trend in winter (JFM months) followed…"
This reads as though the concentrations increase between these months – ie March is bigger than February which is bigger than January – this doesn't appear to be the case from figure 6
"Anthropogenic compounds showed increased concentrations in winter (JFM months) followed…"

The sentence was modified in the revised manuscript as (page 1, lines 20-21 in the revised manuscript):

"Anthropogenic compounds showed increased concentrations in winter (JFM months) followed by a decrease in spring/summer (AMJ/JAS months), […]."

**5/Abstract, Page 1, line 21:**
"… and different concentration levels in winter periods of 2013 and 2014."
These are inevitably different, but the question is by how much are they different?
Suggest including "by up to *XX%* in the case of *compoundY*"

As suggested by referee #2, the different concentration levels between the two winter periods were clarified. However, we indicated absolute differences instead of relative ones. These latter can be

influenced by low concentrations inducing the highest relative differences. The sentence was modified in the revised manuscript as (page 1, lines 21-22 in the revised manuscript):

"[…], and higher concentration levels in winter 2013 than in winter 2014 by up to 0.3 µg m$^{-3}$ in the cases of propane, acetylene and benzene."

**6/Abstract, Page 1, line 21:**

"OVOC concentrations were generally higher in summertime, mainly due to secondary and biogenic sources, whereas their concentrations during fall and winter were potentially more influenced by anthropogenic primary/secondary sources."

This sentence seems a little confusing to me. I agree that the secondary sources of OVOCs will be increased during summertime and that the contribution from biogenic sources will also be greater during summer. As it is written though, it sounds like the anthropogenic secondary sources only contribute to the OVOC concentrations during the winter months.

Thanks to this referee #2 remark, the sentence was modified in the revised manuscript in order not to discard anthropogenic secondary source contributions to summer OVOC concentrations (page 1, lines 22-24 in the revised manuscript):

"OVOC concentrations were generally high in summertime, mainly due to secondary anthropogenic/biogenic and primary biogenic sources, whereas their lower concentrations during fall and winter were potentially more influenced by primary/secondary anthropogenic sources."

**7/Abstract, Page 1, line 26:**

When listing the PMF factors, I suggest that these be listed in order of significance in terms of relative contribution.

As proposed by referee #2, PMF factors are listed in the revised abstract in order of their relative contributions to total concentrations of the 17 selected VOCs for the factorial analysis (page 1, lines 26-30 in the revised manuscript):

"A PMF 5-factor solution was taken on. It includes an anthropogenic factor (which contributed 39% to the total concentrations of the selected VOCs in the PMF analysis) connected to the regional background pollution, three other anthropogenic factors (namely short-lived anthropogenic sources, evaporative sources, and long-lived combustion sources; which together accounted for 57%), originating from either nearby or more distant emission areas (such as Italy and south of France) and a local biogenic source (4%)."

**8/Abstract, Page 1, line 30:**

at the receptor site are also
Suggest changing "receptor site" to ERSA station or observatory

As proposed by referee #2, "receptor site" was replaced by "the Ersa station" in the following sentence (page 1, lines 30-31 in the revised manuscript):

"Variations in these main sources impacting VOC concentrations observed at the Ersa station are also investigated at seasonal and interannual scales."

**9/Abstract, Page 2, line 2:**

"… winter 2014 ones could …"
Delete "ones"

As proposed by referee #2, "ones" was removed in the following sentence (page 2, lines 3-5 in the revised manuscript):

"Higher VOC concentrations during winter 2013 compared to winter 2014  could be related to anthropogenic source contribution variations probably governed by the emission strength of the main anthropogenic sources identified in this study together with external parameters, i.e. weaker dispersion phenomena and the pollutant depletion."

**10/Sect. 2.2.1. VOC measurements:**

Where there any compounds measured by the multiple measurement techniques used in the study? If so, were there any comparison exercises performed to ensure consistency?

They were some VOCs measured by different measurement techniques. In Sect. S1 ("VOCs selected in this study"; see response to referee #2 comment 19), it was specified that:

"17 NMHCs were measured both by steel canisters and multi-sorbent cartridges (underlined species in Table S2) and n-hexanal was measured both by DNPH cartridges and multi-sorbent cartridges. Consistency between recovery species was checked during the intensive field campaign of summer 2013 (see Michoud et al., 2017) and was not checked a second time due to the low temporal recovery of the instruments in terms of data points. In this study, the concentrations of 17 NMHCs measured from steel canisters were retained given their higher number of observations and lower uncertainties compared to those measured with multi-sorbent cartridges. Concentrations of n-hexanal measured using DNPH cartridges were retained in this study for the same reason."

**11/Sect. 3.1, Page 10, line 14:**

"… and the lowest ones in winter …"
Delete "ones"

As proposed by referee #2, "ones" was removed in the following sentence (Page 9, lines 7-8 in the revised manuscript):

"Air temperature observed during the observation period showed typical seasonal variations, i.e. the highest temperatures recorded in summer (i.e. from July to September) and the lowest  in winter (i.e. from January to March)."

**12/Sect. 3.1, Page 10, line 22:**

Paragraph beginning "On one hand, western European …" and ending "…across the north Atlantic toward the British Isles (Kendon and McCarthy, 2015)." on P11, L7 seems excessively long. Of key relevance here (to VOC observations) is that the temperatures were different (lower in 2013) and a short statement/sentence to say the lower temperatures were observed across Europe, with relevant citations, would suffice.

As recommended by referee #2, Sect. 3.1 ("Meteorological conditions") has been shortened in the revised manuscript as follows (page 9, lines 15-21 in the revised manuscript):

"This finding could be explained by different climatic events which have occurred during these two winter periods and have concerned a large part of continental Europe. On one hand,  the stratospheric polar vortex  underwent a sudden stratospheric warming (SSW; Coy and Pawson, 2015) in early January 2013,  having repercussions on the tropospheric polar vortex which  collapsed several times towards Europe.  As a result, air flux orientation was modified from north to east, bringing cold air, and hence causing a particularly rigorous European winter 2013. On the other hand, most of the western European countries experienced a mild winter 2014 characterized by its lack of cold outbreaks and nights, caused by an anomalous atmospheric circulation (Rasmijn et al., 2016; Van Oldenborgh et al., 2015; Watson et al., 2016).

south. On the other side of the Atlantic Ocean, the eastern part of the USA and Canada were struck by cold polar air being advected southward due to the anomalously persistent deflection of the jet stream over the USA. The contrast between cold air advection south across the USA, and the warm tropical Atlantic was likely to have been partly responsible for the persistence and unusual strength of the north Atlantic jet stream. This situation created ideal conditions for active cyclogenesis leading to the generation of successive strong extratropical storms being carried downstream across the north Atlantic toward the British Isles (Kendon and McCarthy, 2015)."

**13/Sect. 3.1, Page 11, line 15:**

"Relative humidity globally followed opposite seasonal variations than temperature and solar radiation ones."
Should read:
"Globally, relative humidity followed opposite seasonal variation to temperature and solar radiation."

As proposed by referee #2, the sentence was modified in the revised manuscript as follows (page 9, line 28 in the revised manuscript):

"Globally, relative humidity followed opposite seasonal variations to temperature and solar radiation ones."

**14/Sect. 3.1, Page 11, line 15:**

"In June 2012, air was dryer compared to June 2013 and 2014 mean relative humidity values …"
Delete:
"mean relative humidity values …" they're not need ed here since these are described in the parentheses.

As proposed by referee #2, the sentence was modified in the revised manuscript as follows (page 9, lines 28-30 in the revised manuscript):

"In June 2012, air was dryer compared to in June 2013 and 2014 mean relative humidity values (mean relative humidity of 57 ±15%, 77 ±16% and 67 ±33% for June 2012, 2013 and 2014, respectively)."

**15/Sect. 3.1, Page 11, line 17:**

"The wind speed did not show a clear seasonal variation over the two years studied, except maybe higher wind speeds in April and May that could induce higher dispersion of air pollutants and could advect air pollutants from more distant sources to the receptor site."
Suggest changing to "The wind speed did not show a clear seasonal variation over the two years studied. Slightly higher wind speeds in April and May 2014 which could induce higher dispersion of air pollutants and advect air pollutants from more distant sources to the receptor site."

As proposed by referee #2, the sentence was split into two sentences in the revised manuscript as (page 9, lines 30-32 in the revised manuscript):

"The wind speed did not show a clear seasonal variation over the two years studied. Slightly higher wind speeds were noticed in April and May, which could induce higher dispersion of air pollutants and advect air pollutants from more distant sources to the Ersa station."

**16/Sect. 3.1, Page 11, line 19:**

"May 2014 encountered particularly windy conditions."
I don't think this sentence is warranted (only 1.5 m/s higher than April) and would suggest removing it this sentence, it is not needed

As proposed by referee #2, the sentence was removed in the revised manuscript.

**17/Sect. 3.2, Page 11, line 31:**

Air masses spending longer periods over the ocean will indeed have undergone more atmospheric processing, but they may also have more influence from oceanic sources of VOCs. While these are likely insignificant compared to the anthropogenic inputs from Continental Europe, I feel they should be mentioned here.

As proposed by referee #2, this piece of information was specified in the revised manuscript (page 10, lines 11-13 in the revised manuscript):

"These contrasting transit times may denote both distinctive atmospheric processing times for the air masses and different oceanic source influences on VOC concentrations observed at the Ersa station."

**18/Sect. 3.2, Page 12, line 4:**

"showed relatively close transport times"
Change to "… short transport times …"

As proposed by referee #2, the sentence was modified in the revised manuscript. However, "short" was not used in order not to create confusion regarding transit times associated with marine air masses which were long (up to 40 h). The sentence (page 10, lines 20-21 in the revised manuscript) was modified as follows:

"On the other hand, marine air masses having short and long trajectories have both shown long transit times (40-48 h – Table 2) and Corsican-Sardinian air masses only concerned long trajectories."

**19/Sect. 3.3 VOC mixing rations, P12, L8:**

The statement "Descriptive statistical results for a selection of 25 VOCs, which showed significant concentration levels during the 2-yr studied period, are summarized in Table 4" implies that more VOCs were measured during the period, but are not reported here because they were below some threshold value decided upon by the authors. If this is the case, there should be a statement describing the selection criteria used to define the "significant concentration levels".

Given referee #2 feedback, the selection of VOCs retained in this study is now described in the revised supplement in Sect. S1, as follows:

**"Section S1: VOCs selected in this study**

In this section, the selection of the VOCs retained for this study among those measured (see Table S2) is presented. Co-eluted VOCs, i.e. n-pentanal+o-tolualdehyde measured from DNPH cartridges and 2,3-dimethylbutane+cyclopentane measured from multi-sorbent cartridges, were not considered in this study. Concentrations of b-pinene resulting from multi-sorbent cartridges were also not considered in this study for analytical reasons.

17 NMHCs were measured from both steel canisters and multi-sorbent cartridges (underlined species in Table S2) and n-hexanal was measured from both DNPH cartridges and multi-sorbent cartridges. Consistency between recovery species was checked during the intensive field campaign of summer 2013 (see Michoud et al., 2017) and was not checked a second time due to the low temporal recovery of the instruments in terms of data points. In this study, the concentrations of 17 NMHCs measured from steel canisters were retained given their higher number of observations and lower uncertainties compared to those measured with multi-sorbent cartridges. Concentrations of n-hexanal measured using DNPH cartridges were retained in this study for the same reason.

Then, to select the VOCs examined in this study, their percentages of values below their detection limit (DL) were examined and VOCs having more than 50% of their concentrations below their DL were discarded. This criteria has concerned four NMHCs measured from steel canisters (2,2-dimethylbutane, i-octane, n-octane and 1,2,4-trimethylbenzene), one carbonyl compound from DNPH cartridges (acrolein) and seven VOCs measured from multi-sorbent cartridges (2-methylhexane, 2,2-dimethylpentane, 2,3-dimethylpentane, 2,4-dimethylpentane, 2,2,3-trimethylbutane, 2,3,4-trimethylpentane and 1,3,5-trimethylbenzene). Furthermore, VOC average signal-to-noise (S/N) ratios were examined. This parameter

determines the average relative difference between concentrations and their corresponding uncertainties, thus pondering the results in function of their quality (Norris et al, 2014). Species having a S/N ratio below 1.2 were discarded (see Debevec et al, 2017). This criteria has concerned three additional NMHCs measured from canisters (2-methylpentane, 3-methylpentane and n-heptane), two additional carbonyl compounds measured from DNPH cartridges (propanal and methacrolein) and 14 additional VOCs measured from multi-sorbent cartridges (cyclohexane, n-nonane, n-decane, n-undecane, n-dodecane, n-tridecane, n-tetradecane, n-pentadecane, n-hexanedecane, 1-hexene, cyclopentene, g-terpinene, styrene and n-heptanal).

Table S2: Listed VOCs as a function of family compounds and instruments. Underlined VOCs were measured by several instruments. Retained VOCs in this study are indicated in bold.

| Family compounds | Steel canisters | DNPH cartridges – Chemical desorption (acetonitrile) – HPLC-UV | Solid adsorbent – Adsorption/thermal desorption – GC-FID |
|---|---|---|---|
| **ALKANES** | **Ethane, propane, i-butane, n-butane, i-pentane, n-pentane,** 2,2-dimethylbutane, 2-methylpentane, 3-methylpentane, **n-hexane,** n-heptane, i-octane, n-octane | | i-Pentane, n-pentane, 2,2-dimethylbutane, 2,3-dimethylbutane+cyclopentane, 2-methylpentane, 3-methylpentane, n-hexane, cyclohexane, 2-methylhexane, 2,2,3-trimethylbutane, 2,2dimethylpentane, 2,4-dimethylpentane, 2,3-dimethylpentane, n-heptane, 2,3,4-trimethylpentane, i-octane, n-octane, n-nonane, n-decane, n-undecane, n-dodecane, n-tridecane, n-tetradecane, n-pentandecane, n-hexadecane |
| **ALKENES** | **Ethylene, propene** | | Cyclopentene, 1-hexene |
| **ALKYNE** | **Acetylene** | | |
| **DIENE** | **Isoprene** | | Isoprene |
| **TERPENES** | | | **a-pinene,** b-pinene, **camphene, limonene, a-terpinene,** g-terpinene |
| **AROMATICS** | **Benzene, toluene, ethylbenzene, m,p-xylenes, o-xylene,** 1,2,4-trimethylbenzene | | Benzene, toluene, ethylbenzene, m,p-xylenes, o-xylene, styrene, 1,3,5-trimethylbenzene, 1,2,4-trimethylbenzene |
| **CARBONYL COMPOUNDS** | | **Formaldehyde, acetaldehyde,** propanal, **i,n-butanals,** n-pentanal+o-tolualdehyde, **hexanal, benzaldehyde, acetone, MEK,** acrolein, methacrolein, **glyoxal, methylglyoxal** | Hexanal, n-heptanal, **n-octanal, n-nonanal, n-decanal, n-undecanal** |

"

The Section S1 is introduced at the end of the revised Sect. 2.2.1 ("VOC measurements", Page 6, lines 20-21):

"Among the 71 different VOCs monitored at Ersa during the observation period, 35 VOCs were finally selected in this study following the methodology described in Sect. S1 of the Supplement."

We would like to inform the referee #2 that the VOC number has increased from 25 to 35 since more carbonyl compound concentration levels and variations are discussed in the revised manuscript, following reviewer #1 suggestion (see referee #1 comment 15). Moreover, following the incorporation of Sect. S1, Table 1 was revised (see response to referee #1 comment 2).


"Note that speciated monoterpenes were measured differently during the summer 2013 campaign, by means of an automatic analyser (see Sect. S4 in the Supplement)."

**21/Sect. 3.4, P12, L28:**

"… dispersion, dilution processes …"

are these the same thing?

They are not the same thing and are partly linked. Horizontal dispersion processes are mostly driven by wind while vertical dilution processes are more related to variations in PBL height. However, due to the Ersa position at the northern tip of Corsica Island, air masses observed at the receptor site can be diluted with marine ones. Moreover, given referees #1 and #2 general feedbacks, the Sect. 3.4 ("VOC variability") has been refocused in the revised manuscript to the descriptive analysis of VOC variations and factors controlling their source variations are now only discussed in the revised Sect. 4 ("Discussions on the seasonal variability of VOC concentrations"), at the exception of environmental parameters influencing BVOC emission variations. As a result, the sentence was removed. Sect 3.4 was revised as follows (Page 11, Lines 12-16):

"~~The variability in VOC concentration levels is governed by an association of factors involving source strength (e.g., emissions), dispersion, dilution processes and transformation processes (photochemical reaction rates with atmospheric oxidants; Filella and Peñuelas, 2006). At this type of remote site, it is also important to consider the origin of air masses impacting the site as distant sources can play a significant role comparatively to local sources (see Sect. 3.5).selectedhence~~ discussed in this section. Seasonal VOC concentration levels are indicated in Table 4. In addition, the comparison between the VOC monitoring measurements investigated in this study with concurrent campaign measurements performed during the summers of 2012, 2013 and 2014 is presented in Sect. S4 of the Supplement, in order to check the representativeness of the 2-yr observation period with regard to summer concentration levels."

**22/Sect. 3.4.1, Page 13, lines 2-18:**

The authors state "Surprisingly, isoprene and α-pinene concentrations were drastically lower in July 2012…" and then go on to state that the temperature and solar radiation during July were lower, therefore, I fail to see the surprise here.

The bigger surprise here seems to be that the July 2013 isoprene falls below the July 2012 level despite the temperature and solar radiation being higher (increasing emissions) and the wind speed being lower (increasing dispersion) during that period. Perhaps including the wind direction in figure 4 may help to explain this?

Considering referee #2 remark, "Surprisingly" was removed in the sentence in revised manuscript and the sentence was rewritten to examine July and August variations in 2012 and 2013.

We do not understand referee #2 remark on isoprene concentrations in July 2012 and 2013 since isoprene concentrations in July 2012 (black line in Fig. 4) were lower than in July 2013 (green line in Fig. 4), consistently with temperature and light variations (Fig. 3).

Dispersion is rather favoured by high wind speeds while low wind speeds can enhance BVOC accumulation and degradation considering their short lifetimes. However, high wind speeds also increased dilution by air masses owing to the Ersa position on the northern tip of Corsica Island. The Ersa station is also surrounded by vegetation (maquis) and oak forests, that's why we did not consider 4h-average wind directions to explain BVOC variations. Corrections applied in Sect 3.4.1 ("Biogenic VOCs"; from Page 11 line 32 to Page 12 line 5 in the revised manuscript):

"Moreover , isoprene and α-pinene concentrations were  higher in July and August 2013 (0.5 ±0.3 and 1.1 ±0.4 µg m$^{-3}$, respectively) than in July and August 2012 (0.3 ±0.2 and 0.6 ±0.3 µg m$^{-3}$, respectively). High concentrations of camphene and α-terpinene were also noticed in August 2013 (0.2 ±0.1 and 0.3 ±0.3 µg m$^{-3}$, respectively; Fig. 4). Solar radiation was lower in July and August 2012, temperature was slightly lower in July 2012 and mean wind speed was slightly higher in July 2012 (Fig. 3), which could affect biogenic emissions and favour their dispersion and their dilution by marine air masses owing to the position of the Ersa station (Sect. 2.1)."

We would like to inform referee #2 that Fig. 4 was modified in the revised manuscript, following the incorporation of additional BVOCs into the discussion of their variabilities in Sect. 3.4.1 '"Biogenic VOCs"):

[Figure]

**Figure 4: (a)** Monthly variations in a selection of biogenic VOC concentrations (expressed in μg m⁻³) represented by box plots; the blue solid line, the red marker, and the box represent the median, the mean, and the interquartile range of the values, respectively. The bottom and top of the box depict the first and third quartiles (i.e. Q1 and Q3) and the ends of the whiskers correspond to the first and ninth deciles (i.e. D1 and D9). **(b)** Their monthly average concentrations as a function of the year; full markers indicate months when VOC samples were collected from 12:00-16:00 and empty markers those when VOC samples were collected from 09:00-13:00."

**23/Sect. 3.4.1, Page 13, line 10:**

"… which may be related to the fact that temperature and solar radiation were more favourable to enhance biogenic emissions in June 2012 compared to June 2013 and 2014 meteorological conditions …"

There is also the effect of relative humidity to consider here. Figure 4 shows the relative humidity was lower in June 2012 and 2014 compared to 2013, see the work of Ferraci et al. for the effect of drought conditions on the emissions of isoprene. Links to this research would be useful here.

We thank referee #2 for the indication of this interesting recent work (Ferracci et al., 2020) on controlling parameters of isoprene emissions. As proposed by referee #2, this research was integrated in the revised manuscript to support drought impact on BVOC emissions (Page 11, lines 24-32 in the revised manuscript):

"For instance, higher mean concentrations of isoprene and α-pinene were noticed in June 2012 (1.0 ±1.1 and 2.6 ±1.4 μg m⁻³ for isoprene and α-pinene, respectively) and June 2014 ones (0.7 ±0.5 and 0.2 μg m⁻³) compared to in June 2013 (0.2 ±0.2 and <0.1 μg m⁻³). Higher June concentrations of camphene (and α-terpinene; not shown) were also noticed in 2014 than in 2013 (Fig. 4). These concentration levels may be related to the fact that temperature and solar radiation were more favourable to enhance biogenic emissions in June 2012 and 2014 compared to in June 2013 meteorological conditions (Sect 3.1). Due to relative humidity values observed in June 2012 and 2014, which were lower than in June 2013, we cannot

rule out that an increase of BVOC concentrations may be related to a transient drought stress-induced modification of BVOC emissions (Ferracci et al., 2020; Loreto and Schnitzler, 2010; Niinemets et al., 2004)."


**24/Sect. 3.4.1, Page 13, line 16:**

"This finding could be the result of a weaker degradation of α-pinene due to lower ozone concentrations observed from October to December compared to summer …"

Emissions of isoprene are light and temperature dependant while monoterpenes are thought to be solely temperature dependant. I'd suggest that the difference in seasonal cycles of isoprene and alpha-pinene is due to the difference between the solar radiation and temperature profiles: solar radiation falls much quicker than temperature which may have the effect of "switching off" the isoprene emissions before the alpha-pinene emissions.

Monoterpenes emissions dependency is related to the emitting vegetation type (e.g., Owen et al., 2002). Emissions of monoterpenes are solely dependent on temperature when they are produced by plants disposing of a storage capability (Laothawornkitkul et al., 2009). Otherwise, their emissions are dependent on both temperature and solar radiation. During the intensive summer observation period, diel concentration variations of monoterpenes were examined (Kalogridis, 2014). They showed similar diel variations to isoprene. Hourly concentration variations of monoterpenes and isoprene were also well correlated, suggesting a common predominant source and similar dependence on the environmental parameters governing their emissions. Concentrations of monoterpenes were also correlated with both temperature and light (Kalogridis, 2014) as isoprene, confirming environmental parameters controlling their emissions in summer. As a result, monoterpenes measured at the Ersa station are not thought to be solely temperature dependent over the entire observation period.

However, the specific seasonal behavior of monoterpenes in fall could highlight additional sources from different emitting plants than those predominantly observed in summer. As indicated by referee #2, solar radiation fell much quicker than temperature in fall which may have the effect of "switching off" the biogenic emissions from plants not having storage capacity. Persistent concentration levels of monoterpenes in fall could be resulted from biogenic emissions, solely dependent on temperature, from plants having storage capability. Nevertheless, hourly BVOC measurements would have been helpful to confirm this statement.

Following this discussion, we slightly modified interpretations of fall BVOC concentration levels as follows (Page 12, lines 5-11):

"Additionally, significant concentrations of α-pinene were noticed from September to November (Fig. 4), while isoprene concentrations were close to the detection limit and temperature and solar radiation were decreasing. However, solar radiation decreased much quicker than temperature in fall (Fig. 3), which could suggest additional temperature-dependant emissions (Laothawornkitkul et al., 2009), contrarily to those prevailing in summer, have influenced α-pinene fall concentrations. Moreover, the lower ozone fall concentrations than in summer (see Fig. S2 of the Supplement) also pointed out a weaker degradation of α-pinene in fall."

"Despite lifetimes in the atmosphere ranging from a few hours to some days, all selected NMHCs were characterized by similar seasonal variation, […]."

**26/Sect. 3.4.2, Page 13, line 27:**

20  "… considering its low photochemical reaction rate with OH radicals …"
"… due to its low photochemical reaction rate with OH radicals …"

We thank the referee #2 for this proposition but the sentence was removed in the revised manuscript.

**27/Sect. 3.4.2, Page 13, line 27:**

25  "… with the highest atmospheric lifetime …"
Replace with "Longest": "… with the longest atmospheric lifetime …"

We thank the referee #2 for this proposition but the sentence was removed in the revised manuscript.

**28/Sect. 3.4.2, Page 13, line 28:**

30  "… It is typically emitted by natural gas use and can be also considered as a tracer of the most distant sources."
Transport and storage of natural gas are also important sources here.

We thank the referee #2 for this proposition but this sentence was removed in the revised manuscript, as the Sect. 3.4.2 was refocused on the description of NMHC concentration variations and hence
35  source information was removed. However, this precision was considered in Sect. 3.5.3 ("Evaporative sources (factor 3)") in the following sentence (Page 16, lines 11-12 in the revised manuscript):

"Additionally, propane can be viewed as a relevant profile signature of natural gas transport, storage and use (Leuchner et al., 2015)."

It was also considered in Sect. 3.5.5 ("Regional background (factor 5)") in the following
40  sentence (page 17, lines 7-9 in the revised manuscript):

"These compounds, with lifetimes of 21-93 days in winter and of 4-19 days in summer, typically result from the transport, storage and use of natural gas […]"

**29/Sect. 3.4.2, Page 14, line 2:**

"… four to ten times higher than ethane one (Atkinson, 1990; Atkinson and Arey, 2003)"
Remove "one", it's not required here.

We thank the referee #2 for this proposition but the sentence was removed in the revised manuscript.

**30/Sect. 3.4.2, Page 14, line 4:**

"… e.g., Leuncher et al., 2015)."
Should be "Leuchner"

We thank the referee #2 for this remark, but the sentence was removed in the revised manuscript since the Sect. 3.4.2 ("Anthropogenic VOCs") has been refocused on the description of NMHC concentration variations. However, this typo was corrected elsewhere in the revised manuscript (e.g., see response to referee #2 comment 28).

**31/Sect. 3.4.2, Page 14, line 6:**

"… winter 2014 ones."
Remove "ones", it's not required here.

We thank the referee #2 for this remark, but the sentence was removed in the revised manuscript and was replaced by the following sentence (page 12, lines 24-26 in the revised manuscript):

"Mean winter NMHC concentrations were higher in 2013 than in 2014 by up to 0.3 µg m$^{-3}$ in the cases of propane, acetylene and benzene (relative differences of 15%, 42% and 42%, respectively)."

**32/Sect. 3.4.2, Page 14, line 11:**

As a result, winter variations of concentration levels concerned at a time close sources and more distant ones and will be more investigated thereafter (Sect. 4.2)."
I don't think this sentence makes sense. Just a statement that the winter period will be investigated later in the manuscript would suffice.

We thank the referee #2 for this remark, but the sentence was removed in the revised manuscript since the Sect. 3.4.2 ("Anthropogenic VOCs") has been refocused on the description of NMHC concentration variations.

**33/Sect. 3.4.3, Page 14, line 21:**

"Nevertheless, acetaldehyde is only produced as a second or higher-generation oxidation product of isoprene for all its reaction pathways with atmospheric oxidants (Millet et al., 2010)."
I don't know what this means – further clarity in the text is needed here.

It means that acetaldehyde is produced by the oxidation of first-generation oxidation products of isoprene (those directly produced from the oxidation of isoprene). However, this sentence was removed in the revised manuscript as we did not precise acetaldehyde secondary sources in the revised Sect. 3.4.3 ("Oxygenated VOCs", following referee #2 general comment and referee #1 comment 16.

**34/Sect. 3.4.3, Page 14, line 30:**

"… with air temperature one,"
Remove "one", it's not required here.

**35/Sect. 3.4.3, Page 14, line 31:**

"… which can denotes that"

Replace with "denote": "… which can denote that"

We thank the referee #2 for this proposition but the sentence was removed in the revised manuscript.

**36/Sect. 3.4.3, Page 14, line 32:**

"These findings are in agreement with a large result on BVOC oxidation on the local photochemistry."
Remove this sentence, not required.

As suggested by referee #2, this sentence was removed in the revised manuscript.

**37/Sect. 3.4.3, Page 14, line 32:**

"… remained relatively significant during fall …"
significant to what?

We used "significant" in the sense of "high". However, in the revised manuscript, we only present their more important variations (those in summer). Information on minor contributions were hence removed, including this sentence.

**38/Sect. 3.4.3, Page 15, line 9:**

"and since meteorological conditions in August 2013 were more favorable to photochemical processes"
What "meteorological conditions" do the authors refer to here? If it is just the higher solar radiation, then state this in the text.

In the revised manuscript, meteorological conditions in August 2012 and 2013 are now discussed in the Sect. 3.4.1 ("Biogenic VOCs", see response to referee #2 comment 22). As a result, these meteorological conditions are not discussed a second time in Sect. 3.4.3 ("Oxygenated VOCs") and the sentence was removed and replaced in the revised manuscript by (Page 13, lines 11-14):

"For instance, the methylglyoxal highest concentrations were monitored in June 2012 (0.7 µg m$^{-3}$), similarly to isoprene (Sect. 3.4.1). Concentrations of n-hexanal peaked up at 0.7 µg m$^{-3}$ in August 2013, in agreement with monoterpenes, especially camphene and α-terpinene (Fig. 4). Formaldehyde showed high concentrations both in June 2012 and August 2013 (2.9 and 3.6 µg m$^{-3}$, respectively)."

**39/Sect. 3.4.3, Page 15, line 17:**

"Acetone is the OVOC of the selection with generally the highest atmospheric lifetime, considering its photochemical reaction rate …"
Unclear, suggest changing to: "Of the measured OVOCs, acetone has the longest atmospheric lifetime, considering its photochemical reaction rate …"

As proposed by referee #2, the sentence was modified in the revised manuscript (from Page 12 line 32 to Page 13 line 4):

"Formaldehyde, acetaldehyde, glyoxal, methylglyoxal and $C_6$-$C_{11}$ aldehydes have relatively short lifetime into the atmosphere (photochemical reaction rate with OH radicals of 9-30 $10^{-12}$ cm$^3$ molecule$^{-1}$s$^{-1}$) and hence they can result from relatively close sources. On the other hand, acetone and methyl ethyl ketone (MEK) have the longest atmospheric lifetime (0.17-1.22 $10^{-12}$ cm$^3$ molecule$^{-1}$s$^{-1}$) of the OVOCs selected in

this study, and hence they can also result from distant sources and/or be formed within polluted air masses before they reach the Ersa station."

**40/Sect. 3.4.3, Page 15, line 22:**

"Acetone showed similar seasonal variations than formaldehyde and acetaldehyde, …"
Replace with "to": "Acetone showed similar seasonal variations to formaldehyde and acetaldehyde, …"

We thank the referee #2 for this proposition but the sentence was removed in the revised manuscript. It was replaced by (page 13 lines 15-16 in the revised manuscript):

"Acetaldehyde and acetone have shown similar seasonal variations, with an increase of their concentrations more marked in summer than in winter (Fig. 6), [...]."

**41/Sect. 3.4.3, Page 15, line 25:**

"… remained significantly high during winter …"
Significant to what? Either include a parameter or remove "significantly" from the sentence.

We thank the referee #2 for this proposition but the sentence was removed in the revised manuscript.

**42/Sect. 3.4.3, Page 15, line 28:**

"… than summer 2013 one, …"
Remove "one", it's not required here.

We thank the referee #2 for this proposition but this sentence was removed in the revised manuscript.

**43/Sect. 3.4.3, Page 15, line 30:**

"… than winter 2014 one, …"
Remove "one", it's not required here.

We thank the referee #2 for this proposition but the sentence was removed in the revised manuscript. It was replaced by (page 13 lines 24-26 in the revised manuscript):

"Additionally, acetaldehyde and acetone concentration variations in winter (e.g., mean February concentrations higher by 0.5 and 2.4 $\mu g\,m^{-3}$ in 2013 than in 2014, respectively) also pinpointed primary/secondary anthropogenic origins (Sect. 3.4.2)."

**44/Sect. 3.4.3, Page 15, line 30:**

"… but admitted low enough to allow advection to the receptor site …"
Delete admitted", not needed here.

We thank the referee #2 for this proposition but the sentence was removed in the revised manuscript.

**45/Sect. 3.4.3, Page 16, line 2:**

"… from other OVOC ones …"
Remove "ones", it's not required here.

We thank the referee #2 for this proposition but the sentence was removed in the revised manuscript.

**46/Sect. 3.4.3, Page 16, line 2:**

"Indeed, MEK concentrations did not show seasonal variations except an increasing winter trend …"

Increased concentrations in winter sounds very much like a seasonal variation. From figure 7, it appears that there is a weak seasonal cycle droning 2013, but this is not replicated (or at least not so clear) in the other years.

MEK seasonal variations were presented differently in the revised manuscript (Page 13, lines 27-30):

"Glyoxal and MEK showed an increase of their concentrations both in summer and winter (Fig. 6 and Table 4), suggesting they were probably produced by several biogenic and anthropogenic sources. Those of glyoxal were in similar proportions (Fig. 6 and Table 4) while MEK increase in winter was more marked than in summer, which may indicate that primary/secondary anthropogenic sources primarily contributed to MEK concentrations."

In 2013, MEK showed an increase of its concentrations in winter, but it was not measured in summer 2013. As a result, potential biogenic contributions could have occurred in summer 2013 and could change MEK seasonal variations interpretations in 2013.

**47/Sect. 3.4.3, Page 16, line 6:**

"… in February 2013 was by 0.2 µg m-3 higher than …"
Remove "by", it's not required here.

As proposed by referee #2, the sentence was modified in the revised manuscript as follows (from Page 13, line 33 to Page 14 line 1):

"Glyoxal and MEK both exhibited different concentration levels during the two studied winter periods since their mean concentration in February 2013 was 65-75% higher than in February 2014, confirming their links with anthropogenic sources"

**48/Sect. 3.4.4, Comparisons with other VOC measurements performed at Ersa**

I don't think this section is needed as it doesn't say a lot. Perhaps the link to the supplementary material could be included in one of the earlier discussion sections.

As recommended by both referees #2 and #1 (comment 4), we removed Sects. 2.2.3 ("Additional high frequency VOC measurements performed at Ersa") and 3.4.4 ("Comparisons with other VOC measurements performed at Ersa") and merged all these results in the revised Sect. S4 ("Comparison of VOC measurements with other ones performed at Ersa"). In the revised manuscript, the Sect. S4 is now introduced in Sect. 3.4. ("VOC variability"). Correction applied in the revised manuscript in Sect. 3.4 (Page 11 lines 13-16):

"In addition, the comparison between the VOC monitoring measurements investigated in this study with concurrent campaign measurements performed during the summers 2012-2014 is investigated in Sect. S4 of the Supplement, in order to check the representativeness of the 2-yr observation period with regard to summer concentration levels."

**49/Sect. 3.5 VOC factorial analysis**

Perhaps some further explanation of the reasoning behind choosing a subset of the measured compounds is required here? Along with a discussion of whether limiting the number of species that are included in the PMF analyses may well affect the result and the number of factors. Perhaps a discussion of how the results here compare to the shorter, intensive campaign results published earlier would help here?

Firstly, as suggested by referee #2, the selection of the 35 VOCs retained in this study is now presented in the revised Supplement in Sect. S1 ("VOCs selected in this study"; response to referee #2 comment 19). Additionally, the methodology of the selection of the PMF solution is now presented in Sect. S2

("Identification and contribution of major sources of NMHCs by PMF 5.0 approach"; in response to referee #1 comment 7). It is specified that (in S2.2 "VOC dataset and data preparation"):

"In order to have sufficient completeness (in terms of observation number), only primary HCNM measurements from bi-weekly ambient air samples collected into steel canisters from 04 June 2012 to 27 June 2014 were retained in this factorial analysis. The NMHC dataset encompassed 152 atmospheric data points having a time resolution of 4 hours. VOC observations resulting from DNPH and multi-sorbent cartridges were not considered in the PMF analysis since they were sampled only 73 and 52 days concurrently to the collection of steel canisters (Fig. S1). Reconstruction of missing data points would significantly affect the dataset quality. Additionally, the restriction of the number of data points to those common to the three datasets (36 data points) would significantly impact the temporal representativeness of the VOC inputs of the study period and hence limiting the discussion on interannual and seasonal variations for statistical robustness reasons. Note that no outlier was removed from the dataset."

Note that, a developed discussion on the limited number of VOCs retained for the PMF analysis is proposed in response to referee #1 comment 10. Moreover, the consideration of the PMF solution of the shorter campaign to consolidate our results has been specified in the revised Sect. 3.5 ("Major NMHC sources", page 14 lines 26-29):

"Since the low number of NMHCs considered in the factorial analysis in this study, PMF result relevance was checked, benefiting from previous PMF analysis performed with the Ersa VOC summer 2013 dataset (42 variables; Michoud et al., 2017) and experimental strategies to characterize VOC concentration variations are discussed in Sect. 3.5.6."

We referred to the summer 2013 PMF solution when we encountered limitations with the 2-yr PMF solution (in Sect. 3.5.1 "Local biogenic source (factor 1)" and Sect. 3.5.2 "Short-lived anthropogenic sources (factor 2)"; see responses to referee #1 comments 6 and 9).

Finally, a discussion on the limited number of species to PMF results has been incorporated in the Sect. 3.5.6 ("Towards the best experimental strategy to characterize variation in VOC concentrations observed at a remote background site") and supported by comparisons of PMF results presented in Sect. S5 ("Comparisons of VOC source apportionment with previous one performed at Ersa") and Sect. S6 ("Examination of a summer 2013 PMF solution realized considering the 17 NMHCs selected in this study", see response to referee #1 comment 10). Section 3.5.6 has hence been reviewed to better highlight the relevance of the PMF solution to identify primary NMHC sources and its limitation to examine VOC concentration variations observed at Ersa. Corrections applied to Sect. 3.5.6 (from Page 17, line 26 to Page 19, line 23 in the revised manuscript):

[revised manuscript text omitted]

We thank the referee #2 for this proposition. The sentence was modified in the revised manuscript (page 14, lines 20-22 in the revised manuscript):

"A part of them may not be precisely associated with emission profiles but should rather be explained as aged profiles originating from several source regions comprising several source categories (Sauvage et al., 2009)."

**51/Sect. 3.5.1, Page 17, line 6:**

"The relative load of this VOC for the factor 1 is 70%."
Clarify what is meant by this statement.

It means that isoprene concentrations corresponded to 70% of the factor 1 mass. The sentence was modified in the revised manuscript (Page 15, line 3) as follows:

"The isoprene relative contribution to the factor 1 is 70%."

**52/Sect. 3.5.2 Short-lived anthropogenic sources (factor 2)**

The description of Factor 2 and its influences is rather vague and contains a number of potential contributing sources. This is a result of this type of analysis, but the authors need to be wary of making contradicting statements, for example describing "slightly higher contributions during fall" (P17, L27), then "factor 2 contributions were also significant in spring and summer" (P17, L32) and then "mean monthly factor 2 contributions (Fig. 10b2) pointed out no clear seasonal variation over the study period" I think this is due to the differences observed between different years and so care should be taken not to generalise here.

Given the location and the site typology of the Ersa station, we think that factor 2 encompasses different source categories (combustion processes, solvent use and gasoline evaporation) of various origins. Following the referee #2 remark, we insisted on this statement and the following sentence was modified (Page 15, lines 21-23):

"Factor 2 is hence attributed to the grouping of several short-lived anthropogenic sources, partly related to gasoline combustion and/or evaporation and solvent use."

We would like to inform referee #2 that factor 2 contribution variations were no longer presented in Sect. 3.5.2 ("short-lived anthropogenic sources (factor 2)"), since Sects. 3.5.1-3.5.5 have been refocused on the identified primary sources associated with the 5 factors of the selected PMF solution. Seasonal and interannual variations in factor contributions are now used only in Sect. 4 ("Discussions on the seasonal variability of VOC concentrations") to explain VOC concentration variations observed at the Ersa station during the 2-yr period.

In Sect. 4, we checked our statements on factor 2 contribution variations. Given the fact that factor 2 regrouped several anthropogenic sources of various origins and limitations of the PMF model to reconstruct some VOCs composing factor 2 (see responses to referee #1 comment 9 and referee #2 comment 49), we decided to limit the discussion on factor 2 contribution variation in the revised manuscript.

**53/Sect. 3.5.2, Page 17, line 16:**

"This latter is mainly consisted of primary anthropogenic ..."
What is meant by "This latter"?

"This latter" referred to factor 2. It was used to avoid the repetition. Given referee #2 remark, it was modified in the revised manuscript (Page 15, lines 13-14):

"This factor is mainly consisted of primary anthropogenic compounds, [...]"

**54/Sect. 3.5.2, Page 17, line 19:**

"… with an average contribution to the sum of measured VOC concentrations from this factor of 66%."
Is this correct? Looking at figure 9(b), factor 2 does not appear to ever be 66% of the total.

Here, we wanted to say that concentrations of toluene, EX, ethylene and propene contributed to factor 2 by 66%, on average (i.e. the factor 2 explained 66% on average of their concentrations). Factor 2 contribution to the total measured concentrations of the 17 NMHCs selected for the PMF analysis was 19%. Given referee #2 remark, this statement was modified in the revised manuscript (Page 15, line 16) as follows:

"The relative contribution of these VOCs to factor 2 is 66%."

**55/Sect. 3.5.2, Page 17, line 31:**

"… winter, conducting to less dilution of emissions, …"
suggest changing to "leading to"

We thank the referee #2 for this proposition but the sentence was removed in the revised manuscript.

**56/Sect. 3.5.2, Page 17, line 32:**

"However, factor 2 contributions were also significant in spring and summer …"
This only appears to be the case in 2013.

We thank the referee #2 for this proposition this sentence was removed in the revised manuscript, since Sects. 3.5.1-3.5.5 have been refocused on the identified sources associated with the 5 factors of the selected PMF solution. Seasonal and interannual variations in factor contributions are now used only in Sect. 4 ("Discussions on the seasonal variability of VOC concentrations") to explain VOC concentration variations observed at the Ersa station during the 2-yr period. To explain VOC concentration variations in spring and summer, it is indicated that (page 21, lines 1-4):

"Summer and spring contributions of short-lived anthropogenic sources seemed to be more variable as a function of the year (0.7-1.1 µg m$^{-3}$; Figs. 10 and S4). This finding suggests that these sources were largely influenced by origins of air masses, which advected to Ersa numerous emissions, potentially of variable strength and from various locations relatively close to Ersa."

As noticed by referee #2, an increase of factor 2 contributions occurred in spring 2013 and another one in summer 2012. However, compared to other factor contribution variations, occurring especially in winter, changes in factor 2 concentration levels in spring and summer appeared to be less significant. As a result, we decided not to stress on this point in the revised manuscript, in agreement with our decision to limit the discussion on factor 2 contribution variation in the revised manuscript (see referee #2 comment 52).

**57/Sect. 3.5.2, Page 18, line 1:**

"… which could illustrate an enhanced evaporation of gasoline, solvent inks, paints and additional applications during these months as a result of higher temperatures."
This is contradicted by the temperature data shown in figure 4(b1) which shows lower temperatures in June 2013 compared to 2012 and 2014 which have smaller factor 2 contributions shown in figure 10(b2). The authors go on to give explanation of these differences, but I feel it's important to highlight this anomaly here.

We thank the referee #2 for this feedback. For different reasons stated in referee #2 comment 52, we decided to limit the discussion on factor 2 contribution variation and hence this statement was removed in the revised manuscript. Given the numerous parameters potentially having an effect on sources associated with factor 2, the control of the temperature on some potential sources composing

factor 2 cannot be examined distinctly to other influences, that is why we suggested only its effect on factor 2. Factor 2 contribution variations were supposed to be mainly driven by air mass origins, which could partly explain why lower factor 2 contributions were observed in June 2013 than in June 2012 and 2014.

**58/Sect. 3.5.5, Page 20, line 17:**

"… probably related to photochemical decay and dilution processes."

Earlier in this section the authors state that natural gas may be an important source for factor 5 so presumably a summer decrease in emissions may also contribute to the observed seasonal variation?

We thank the referee #2 for this suggestion. As the discussion on factor 5 contribution variations was removed from the revised Sect. 3.5.5 ("Regional background (factor 5)"). This information was added in Sect. 4.1 ("VOC concentration variations in spring and summer"; Page 20 lines 11-15):

"Moreover, regional background explained in spring and summer from 24 to 53% of the total Ersa concentration of the NMHCs selected in this study. As natural gas sources were attributed to the regional background (Sect. 3.5.5), a decrease in their emissions can presumably occurred in the hot season, enhancing the decline in its contributions (Fig. S4). These regional background contributions also suggest that aged emissions advected by air masses to the Ersa station significantly influenced VOC concentrations observed during these seasons."

**59/Sect. 3.5.5, Page 20, line 21:**

"Mean factor 5 contributions in function of air mass origins were in the same range, except that more elevated contributions were noticed under the influence of European air masses (especially those potentially connected to distant contributions; Fig. 11) compared to the ones related to others continental origins."

This is a confusing sentence; can it be re-written for improved clarity?

As proposed by referee #2, this sentence was rewritten in the revised manuscript (Page 17, lines 19-20):

"Factor 5 showed slightly higher contributions when the Ersa station was under the influence of European air masses (especially those having long trajectories and hence potentially connected to distant emission areas; Fig. 9)."

**60/Sect. 4.1 The controlling factors**

This whole section appears to re-cap the information given in section 3.5. In order to reduce the size of the manuscript, I would suggest these sections be combined to give a more concise explanation of the observations at the site. This could be by either including extra information in section 3.5 (and removing section 4

We thank referee #2 for this feedback. As a result, we decided to remove information in Sect. 3.5 ("Major NMHC sources") concerning factor contribution variations and to keep Sect. 4 ("Discussions on the seasonal variability of VOC concentrations") in the revised manuscript. As anthropogenic NMHC sources have shown similar seasonal variations, we thought that it was more relevant to discuss their variations conjunctly instead of individually. This decision was also motivated by referee #2 general comment. Note that monoterpenes and OVOC seasonal concentrations have been incorporated in Sect. 4 to support discussing changes in biogenic contributions and VOC depletion and to further discuss OVOC sources. We would like to inform referee #2 that we decided to reorganize Sects. 4.1 and 4.2 in the revised manuscript: on one hand (Sect. 4.1) the factors explaining the VOC concentration levels and their variations in spring and summer and on the other hand (Sect. 4.2) the factors controlling those in winter and fall. Sect. 4 has been revised as follows (from Page 19 line 24 to Page 23 line 16 in the revised manuscript):

[revised manuscript text omitted]

**61/Sect. 4.1, Page 22, lines 13-14**

"… favouring phenomena of vertical dispersion."
Delete "phenomena of", not required here: "… favouring vertical dispersion."

As proposed by referee #2, the sentence was modified in the revised manuscript (from Page 20, lines 32 to Page 21 line 1):

"These findings can suggest that these anthropogenic sources originating from distant emission areas were largely influenced by pollutant depletion and vertical/horizontal dispersion during these seasons."

**62/Sect. 4.2, the particular case of winter**

Figure 13, referred to in the text needs further explanation and a legend describing the colour scheme and the meaning of C1 – C5. These are described elsewhere, but should be included again here in the figure.

As suggested by referee #2, the figure was modified in the revised manuscript:
"

[Figure]

**Figure 11:** Accumulated average contributions (expressed in µg m⁻³) of the NMHC anthropogenic sources (factors 2-5 which explained measured concentrations of the 16 selected NMHCs in the PMF analysis – Sect. 3.5) per season as a function of air mass origins (Sect. 3.2). Winter: 01/01-31/03 periods – fall: 01/10-31/12 periods. "

**63/Sect. 4.2, Page 24, line 2**

"… compared to winter 2013 ones …"
Remove "ones", it's not required here.

As suggested by referee #2, the sentence was modified in the revised manuscript (Page 22, lines 7-9):

"Even though winter contributions of long-lived combustion sources, short-lived anthropogenic sources and evaporative sources were significantly reduced in 2014 compared to in 2013 (absolute difference from 0.3 to 0.9 µg m⁻³; Fig. 10), the seasonal pattern of their variations were similar in 2013 and 2014, as depicted in Fig. S4. "

**64/Sect. 4.2, Page 24, line 34**

"As a consequence, this finding also point out that shorter observation periods (i.e., up to two months) may be reflected the variability of the identified parameters under the specific meteorological conditions of the studied period."
Sentence is poorly written and doesn't make sense, needs to be re-written for clarity.

We thank referee #2 for this suggestion, but the sentence was removed in the revised Sect. 4.2 ("VOC concentration variations in fall and winter"). However, this information was rewritten in the conclusion for clarity (from Page 26 line 33 to Page 27 line 3) as follows:

"They also pointed out that the PMF-derived factors controlling VOC concentration variations at remote sites may be mainly controlled by the meteorological conditions that occurred during the study period when issued from short observation periods (i.e. up to two months)."

**65/Sect. 5, VOC concentration variations in continental Europe**

Figure 15 is referred to in the text. The ERSA site should be highlighted in the caption to identify the station under study here.

As suggested by referee #2, the Ersa site was highlighted in the caption of the revised Figure 13 as follows:
"

[Figure]

[Figure]

**Figure 13: Monthly concentration time series of a selection of NMHCs (expressed in µg m⁻³) measured at Ersa and 17 other European monitoring stations. Stations are indicated according to their GAW identification (Sect S7). "ERS" referred to the study site."**

**66/Sect. 5, Page 25, line 17**

"… observed in most continental Europe …"

"… observed in most of continental Europe …"

As suggested by referee #2, the sentence was modified in the revised manuscript (From Page 23 line 31 to Page 24 line 1):

"As a result, the study of concentration variations of these source tracers may help to highlight temporal and spatial variations in source contributions to NMHC concentrations observed in most of continental Europe."

**67/Sect. 5, Page 25, line 20**

"… were globally lower and …"
not globally, but European wide

As suggested by referee #2, the sentence was modified in the revised manuscript (Page 24, lines 4-5):

"Monthly NMHC concentrations were European wide lower and relatively homogeneous from June to August whatever the location and the typology of the station […]."

**68/Sect. 5, Page 25, line 23**

"… suggesting a high importance of photochemistry processes and vertical dispersion phenomena in regulating concentration levels."
I would suggest that temperature (linked to boundary layer height) is the main driver here. As the authors state earlier in the manuscript, the majority of these compounds (with the exception of ethylene) have relatively long lifetimes and so photochemistry will likely be limited.

As suggested by referee #2, the sentence was modified in the revised manuscript (Page 24, lines 7-8):

"It suggests that the temperature was the main driver in regulating summer concentration levels, linked to photochemistry processes and vertical dispersion."

**69/Sect. 5, Page 26, line 5**

"Then, at stations located …"
Delete "then"

As suggested by referee #2, the sentence was modified in the revised manuscript (Page 24, lines 21-22):

"Then, At stations located in central Europe (i.e. stations located in Switzerland, Germany and Czech Republic - see Sect. S7), […]."

**70/Sect. 5, Page 26, line 22**

to normal values
How do the authors conclude which is "normal"?

These normal values were obtained from the CPC (Climate Prediction Center) of NOOA (Fig. S5). The CPC produces monthly and three month maps of total precipitation and percent of normal plus average temperature and departures from normal (https://www.cpc.ncep.noaa.gov/products/monitoring_and_data/restworld.php, last access: 13/10/2020). As indicated in Fig. S5, normal values are calculated using the average of monthly (or quarterly) values for 1981-2010 period. The belonging of these normal values to NOOA is now indicated in Fig. S5 and in Sect. 5 ("VOC concentration variations in continental Europe"; Page 24, lines 27-29):

"Furthermore, precipitations in central Europe were less frequent and/or intense both in winters 2013 and 2014 compared to normal values for the 1981-2010 period (average values calculated by the NOAA – see Fig. S5 of the Supplement) […]."

**71/Sect. 5, Page 27, line 6**

"Then, VOC concentrations …"
Delete "then"

As suggested by referee #2, the sentence was modified in the revised manuscript (Page 25, lines 24-25):

" NMHC winter concentrations monitored in Scandinavia (represented by PAL station results on Fig. 14) were higher in 2014 than in 2013 […]."

**72/Sect. 5, Page 27, line 11**

Sentence containing "… was not as warmer-than-average as …" is poorly written, please re-write for clarity

As suggested by referee #2, the sentence was modified in the revised manuscript (Page 25, lines 29-31):

"Even though these regions experienced a cold winter in 2013 (Fig. S5 of the Supplement), early winter 2014 in northern Europe was also colder than normal values for the 1981-2010 period, since an intense cold wave occurred in January 2014 and was associated with a strong anticyclone centred on western Russia and extending from Finland to Crimea."

**73/Sect. 6 Conclusions**

This section is far too long and needs to be re-written more concisely.

As recommended by referee #2, the conclusions have been rewritten more concisely in the revised manuscript (from Page 26, line 2 to Page 27, line 11):

[revised manuscript text omitted]